# A Systematic Assessment of Weak-to-Strong Confidence Prediction in Large Language Models

**Tracy Yixin Zhu**[*]                                                    *yz5880@nyu.edu*
*New York University*

**Yukai Yang**[*]                                                         *yy2949@nyu.edu*
*New York University*

**Marco Morucci**                                                        *moruccim@msu.edu*
*Michigan State University*

**Tim G. J. Rudner**                                                     *tim.rudner@utoronto.ca*
*University of Toronto*

**Reviewed on OpenReview:** *https://openreview.net/forum?id=xYSzkg5qPD*

## Abstract

As large language models (LLMs) are deployed in increasingly diverse applications, understanding their capacity through uncertainty quantification (UQ) is crucial for ensuring safe and reliable behavior. Reliable uncertainty estimates that accompany the text generated by an LLM can signal when a response is likely to be incorrect and thus serve as an effective fail-safe mechanism against hallucinations. We study the extent to which a smaller and weaker open-access model, using only question embeddings and a lightweight probe, can predict the probability that a stronger black-box generator answers a query correctly. Across six benchmarks, two generators, and fifteen open-access predictors, we find that this simple approach provides useful confidence estimates: embeddings from models as small as Llama3-8b achieve 83.4% AUROC on TriviaQA and 64.3% on MMLU, and improve selective generator accuracy by up to 17.9%. Our analysis shows that performance is not determined by predictor size alone, but depends more strongly on representational compatibility between weak model embeddings and strong model correctness. The signal is robust to decoding configurations, label imbalance, and embedding aggregation choices, but is weaker on reasoning-heavy benchmarks such as SuperGPQA and transfers poorly across datasets. These findings suggest that weak-to-strong probes are best viewed as lightweight in-distribution confidence estimators: after generator-based labels are collected for training, they provide efficient deployment-time uncertainty estimates without repeated generator sampling. Overall, our results provide a systematic baseline for studying scalable oversight of black-box LLMs. Our code and data are available at: `https://github.com/YukaiYang0803/w2s-confidence-prediction`.

## 1 Introduction

To what extent can a weaker language model predict a stronger language model's ability to answer a given question correctly? In this paper, we explore this question and find that a frozen open-access backbone paired with a two-layer probe on question-only embeddings can predict correctness of frontier models such as GPT-4o (OpenAI, 2024a) across diverse question-answering benchmarks.

Prior work on uncertainty quantification in large language models (LLMs) has largely focused on estimating a model's uncertainty from the model itself via self-reported probabilities (Kadavath et al., 2022), LLM-as-a-

---

[*]Equal contribution.

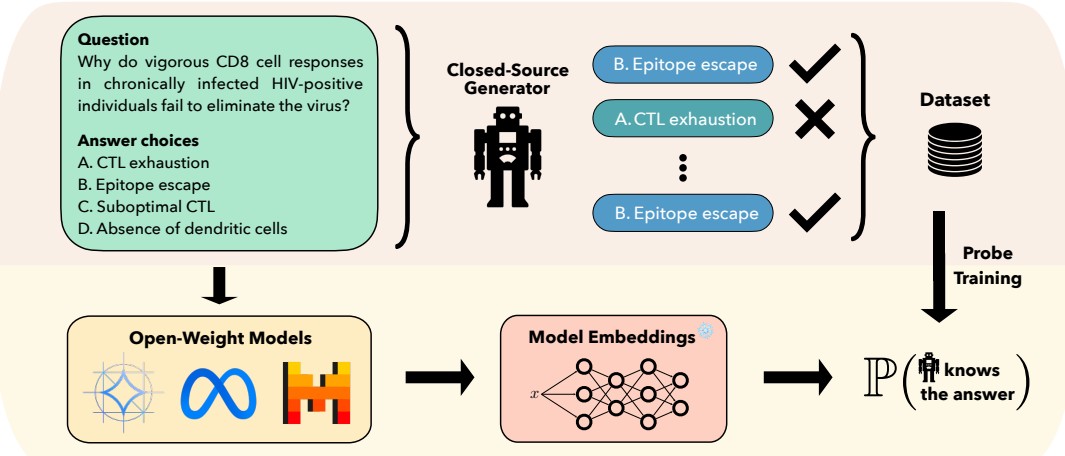

Figure 1: A schematic representation of weak-to-strong confidence prediction. First, given a dataset of questions, answers from a "strong" generator model are collected and labeled to indicate whether the generated answer is correct or incorrect. The question-answer pairs are aggregated into a weak-predictor training dataset. Using this dataset, a probe is then trained on top of a "weak," frozen open-weight or open-access model, with the goal of predicting whether the generator model is able to answer a given question correctly. After training is completed, the weak predictor model can then be used to produce confidence scores that predict whether the strong generator model will be able to answer a new, previously unseen question correctly.

judge scoring (Liu et al., 2023; Zheng et al., 2023), or sampling-based disagreement (Wang et al., 2023), often without external evidence. These approaches are fundamentally introspective, probing a model's *internal* latent beliefs and decision process rather than using an *external* signal. As a result, they do not directly apply to strong, closed-weight models such as GPT, where we only observe the final answers. Moreover, recent work shows that such introspective signals can be highly sensitive to task design and prompt formats, leading to brittle calibration even when internal access is available (Dorner et al., 2025; Gu et al., 2024).

By contrast, it remains unclear when a *weaker* model, using for example question-only embeddings, can reliably predict a stronger black-box model's correctness across diverse tasks and generators (Burns et al., 2023). We address this gap with a systematic analysis of weak-to-strong confidence prediction in language models across six benchmarks and five groups of open-access models. Concretely, we study: given a question posed to one LLM (the *generator*), can a *weaker* LLM (the *predictor*) estimate the likelihood that the generator's answer is correct? We show that this is possible with a minimal evaluation setup: we apply a small probe on top of frozen embeddings from an open-access predictor to estimate the generator answer's correctness. The predictor can be orders of magnitude smaller than the generator, and only the simple probe is trained. This design makes it easy to analyze when such oversight works and what signals it relies on. We illustrate our settings in Figure 1. This yields an *external*, data-driven confidence signal that does not rely on the black-box model's own probabilities or hidden states and is computationally efficient to deploy after training, making the framework practical for stable target distributions where generator responses can be collected.

In our evaluation, we treat GPT-3.5-turbo and GPT-4o as black-box generators and use 15 open-access predictors: 13 open-weight LLM backbones drawn from the Llama (Meta, 2024), Mistral (Jiang et al., 2023), Gemma (Gemma Team, 2024; Gemma Team and Google DeepMind, 2025), and GPT-oss(OpenAI et al., 2025) models, plus two embedding models from OpenAI (OpenAI, 2024b) and DeepMind (Google DeepMind, 2025), with sizes ranging from 7B to 120B. We train a shallow MLP probe on the predictor's question-only embeddings to predict a binary correctness label, outputting the prediction via a softmax over logits. Notably, our setup does **not** require the *generators*' log-probabilities: the generators are used solely to produce answers. Our evaluations span 6 benchmarks, and we report both per-dataset and macro-aggregated results.

We position this setup as a pragmatic baseline for uncertainty quantification in LLMs, providing a standardized point of comparison for future work. The approach requires only question embeddings and generator responses.

Although we find limited zero-shot transfer across datasets, this does not preclude practical utility, since the probe is lightweight and inexpensive to train per target distribution once generator answers can be sampled, and we envision this simple approach can be particularly useful in narrowly defined settings such as medical question answering. In addition, our analysis raises several open questions: *Can predictor predictions enhance reliability of generator outputs? Does predictor performance improve with predictor model size? Are predictors just learning the label distributions? Do different embedding aggregations impact predictor performance?* To facilitate further research, we release all datasets.

To summarize, our key contributions are:

1. We examined the framework of weak-to-strong confidence prediction for LLMs and empirically showed that embeddings from "weaker" open-access models can predict the correctness of a "stronger" language model, thereby improving generator reliability in selective prediction (See Table 1).

2. We provide a systematic empirical study across six benchmarks, two generators and fifteen weak predictors, showing that predictor performance depends more on representational compatibility between weak-model embeddings and strong-model correctness than on model scale alone.

3. Through extensive ablations, we show that predictor performance is robust to label distributions, embedding aggregation strategies, and sampling and decoding choices, but it is weaker on reasoning heavy benchmarks and exhibits limited cross dataset transfer. These findings suggest that the observed weak-to-strong signal is not an artifact of any single evaluation choice.

4. We release benchmark datasets to facilitate systematic study of weak-to-strong confidence prediction across future LLM generations.

## 2 Related Work

**Weak-to-strong generalization.** As large frontier models are outperforming humans on various benchmarks (Wei et al., 2022; Kaplan et al., 2020; Brown et al., 2020), we might be concerned that human intelligence is not strong enough to reliably supervise future models. But can we use a source that is weaker to supervise the training of a stronger model? This is the motivation of weak-to-strong generalization (Burns et al., 2023). Researchers have developed different frameworks to study this question. Hase et al. (2024) find that large pretrained models can generalize well from easy to hard data using in-context learning, linear probe heads, and other methods on various datasets. Liu & Alahi (2024) improves such generalization with a hierarchical mixture of experts (MoE) and Du et al. (2023) asks multiple LLM agents to debate with each other to achieve better generalization. weak-to-strong generalization has also been studied theoretically: Lang et al. (2024) shows that the student model can achieve better performance on the actual task than a weak teacher model under certain robustness conditions. Our work focuses on showing the weak-to-strong trend on model uncertainty prediction, an important step towards building more interpretable frontier models.

**Uncertainty quantification in LLMs.** Much research into uncertainty quantification in LLMs has had the goal of mitigating problems stemming from hallucination (Yadkori et al., 2024). Many works develop novel external metrics to compute model uncertainty or confidence in an answer generated to a given question (Kuhn et al., 2023; Duan et al., 2024). Another popular approach is to prompt LLMs to explicitly express their uncertainty (Lin et al., 2022; Tanneru et al., 2023), and other works explicitly train LLMs to identify which questions they do not know the answer to(Amayuelas et al., 2024; Yin et al., 2023). In this work, we will instead directly *learn* the probability that a model may be correct or incorrect about a specific question it is asked using an external, other model, which may be different from the question-answering one.

**Selective prediction.** Given the nature of our problem of interest (predicting how likely the answer of an LLM to a given question is to be incorrect) it is crucial that our evaluation models are not only accurate when their predicted probabilities are binarized into yes/no prediction labels, but also that they are well-calibrated, i.e., that their predicted probabilities match the *frequency* with which a given generator LLM is correct or incorrect in answering a question. This is crucial as this predicted probability is likely of more interest to an end-user than a simple yes/no judgment would be. This directly links our work to existing literature on model calibration. To compute model calibration, we introduce a rejection class (El-Yaniv & Wiener, 2010a): if the predictor's uncertainty is beyond a given threshold, the model abstains from making a prediction. By

Table 1: Accuracy and selective accuracy of two strong generator models, GPT-3.5-turbo and GPT-4o, across six datasets. For each rejection rate, we report the mean and standard deviation of selective generator accuracy across five Llama predictor backbones. The results show that weak predictor models improve generator reliability under selective prediction, with larger gains on knowledge retrieval benchmarks and more limited gains on reasoning-heavy benchmarks such as SuperGPQA.

| | GPT-3.5-turbo | | | | GPT-4o | | | |
|---|---|---|---|---|---|---|---|---|
| | Accuracy | Selective Accuracy (with different rejection rates) | | | Accuracy | Selective Accuracy (with different rejection rates) | | |
| Dataset | | 10% | 30% | 50% | | 10% | 30% | 50% |
| TriviaQA | 74.33 | 79.27±1.40 | 85.19±2.99 | 88.77±3.68 | 81.47 | 85.55±1.15 | 89.26±2.14 | 91.41±2.73 |
| CounterFact | 94.42 | 96.28±0.36 | 97.33±0.70 | 97.95±0.84 | 95.68 | 97.16±0.31 | 97.90±0.49 | 98.24±0.69 |
| Winogrande | 72.70 | 73.95±0.73 | 75.24±1.73 | 77.06±3.56 | 90.00 | 90.57±0.36 | 90.99±0.91 | 91.24±1.44 |
| MMLU | 69.78 | 71.96±0.56 | 74.40±1.55 | 75.87±2.63 | 85.93 | 87.24±0.53 | 88.35±0.95 | 89.00±1.90 |
| MedQA | 59.95 | 61.21±0.43 | 63.82±1.62 | 66.29±2.33 | 88.30 | 89.24±0.45 | 89.94±0.85 | 90.48±0.76 |
| SuperGPQA | 32.37 | 33.20±0.42 | 35.58±0.63 | 38.71±1.36 | 44.98 | 46.60±0.28 | 50.46±0.89 | 54.98±1.94 |

evaluating model outputs on a variety of thresholds, we can compute selective metrics like selective (balanced) accuracy (Fisch et al., 2024). By comparing the selective metrics with the non-selective ones, we can better understand if our model is well-calibrated (Rudner et al., 2024; Varshney et al., 2022).

## 3 Experimental Design

We employ a large and more capable pretrained language model, denoted as **the generator**, $f_G$, to generate responses to questions given in our evaluation datasets. Then we use a smaller and weaker open-access model (an open-weight LLM or an embedding model via API) with a linear probe as **the predictor**, $f_P$. Our goal is to train $f_P$ to learn the uncertainty of $f_G$ from distributions of correctness labels $y$ of the responses and the embeddings $X$ from the dataset $\mathcal{D}$.

### 3.1 Dataset Construction

For a given dataset $\mathcal{D}$, we collect $n$ questions from $\mathcal{D}$ and query $f_G$. For each question $x_i$, we sample $k$ (default 10) answers from $f_G$. We then compare each generated answer to the true answer provided in $\mathcal{D}$ and obtain $Y = \{1, 0\}$, indicating whether $f_G$ answers correctly. By averaging the $k$ samples for each question, we compute a probability label $y$, representing the likelihood that an answer from $f_G$ is correct.

$$y_{\text{true},i} = \frac{1}{k} \sum_{j=1}^{k} \mathbb{1}\{f_G(x_i)_j = \text{ground truth}(x_i)\} \tag{1}$$

$$y_{\text{pred},i} = f_P(x_i). \tag{2}$$

This probability serves as the prediction target for the predictor, that is, $P(f_G$ knows the answer$)$. We use a cross entropy loss function:

$$-\sum_i \left[ y_{\text{true},i} \log\left(y_{\text{pred},i}\right) + (1 - y_{\text{true},i}) \log\left(1 - y_{\text{pred},i}\right) \right]. \tag{3}$$

We evaluate with questions from six datasets, three knowledge recall datasets: TriviaQA (Joshi et al., 2017), CounterFact (Meng et al., 2023), and Winogrande (Sakaguchi et al., 2019), and three logical reasoning datasets: MMLU (Hendrycks et al., 2021), MedQA (Jin et al., 2020), and SuperGPQA (M-A-P Team, 2025). For open-ended TriviaQA, only the original question is given to $f_G$, whereas for the multiple-choice datasets we provide the question along with up to four choices. We select the first 10,000 questions from chosen MMLU categories and the first 10,000 from the other datasets. More details on how our evaluation data was selected can be found in Appendix A. We provide the generator accuracy in answering each dataset in Table 1.

## 3.2 Evaluation Setup

To train the predictor $f_P$, we leverage the latent representations of the input questions produced by a "backbone" model, which may be either a generator LLM or an embedding-only model. These representations are inputs to a simple probe that predicts the correctness labels. We freeze the representations and only train the probe.

**Probe head classifier setup.** We implement a probe head as a two-layer neural network that takes as input the high-dimensional text representations generated by an open-access fixed "backbone" model. Following Kadavath et al. (2022), we use only the representation of the last non-padding token of each prompt, resulting in an input $X \in \mathbb{R}^{n \times d}$, where $d$ is the output dimension of the last layer of the "backbone" model, before the final linear head. We normalize the output $y \in \mathbb{R}^2$ with a softmax function, so that $y$ becomes two probabilities that sum to 1. We train the probe head using cross-entropy loss.

For $f_G$, we collected answers from GPT-3.5-turbo and GPT-4o. For $f_P$, we use a total of 15 open-access models: five Llama models, three Mistral models, three Gemma models, two GPT-oss models, and two embedding-only models (Text-Embedding-3-Small and Gemini-Embedding-001) via API. We use these models as the backbone to obtain embeddings. Our results show that open-access models as small as Llama3-8b can be used to predict uncertainty of much larger black-box models like GPT-4o.

**Evaluation metrics.** We evaluate the performance of our predictors using a range of metrics. Since $y$ represents a probability, we discretize both $y_{\text{true}}$ and $y_{\text{pred}}$ with threshold $= 0.5$ to compute metrics including accuracy and F1 score. For metrics like AUROC and AUPRC, we keep $y_{\text{pred}}$ as a probability, discretizing only $y_{\text{true}}$ to treat it as a classification task. To address data imbalance, we subsample the data to create a class-balanced test set, which we use to compute balanced accuracy. Additionally, we compute selective metrics as an area under the prediction curve with different thresholds of binary categorical entropy. Unless stated otherwise, all metrics in the plots and tables are presented in percentages.

**Datasets.** We evaluate the predictive performance on across six benchmarks spanning both knowledge retrieval and logical reasoning. TriviaQA (Joshi et al., 2017) is an open-domain question-answering dataset that covers factual knowledge in various fields. CounterFact (Meng et al., 2023) aims at factual consistency, which requires models to identify the true statements over the counterfactual one. Winogrande (Sakaguchi et al., 2019) asks models to fill in pronoun blanks and requires commonsense knowledge. MMLU (Hendrycks et al., 2021) contains questions from various professional and academic domains. MedQA (Jin et al., 2020) consists of medical licensing exam questions and assesses models' medical reasoning ability. Lastly, SuperGPQA (M-A-P Team, 2025) is a challenging benchmark with graduate-level scientific questions.

## 3.3 Selective Prediction

We use selective metrics to further assess the models' predictive uncertainty. Selective prediction modifies the standard prediction pipeline by introducing a rejection class (El-Yaniv & Wiener, 2010b). Given a classifier $f$ where $f(x) = \text{argmax}_{k \in \mathcal{Y}} f(x|k)$, the selective prediction model introduces a selection function $s$ which determines whether a prediction should be made. This selection function can be based on the outputs of $f$, such as $s(x) = \max_{k \in \mathcal{Y}} f(x|k)$. The selective prediction model $f_s$ is defined as

$$f(x; \tau) = \begin{cases} f(x) & \text{if } s(x) \geq \tau \\ \bot & \text{otherwise } s(x) < \tau, \end{cases} \tag{4}$$

where $\tau$ represents the rejection threshold.

Predictive uncertainty is a natural choice for a selection function. If a model's predictive uncertainty is above a certain threshold, the selection function will decline to make a prediction. In many real-world applications, especially those where the cost of an incorrect prediction is high (e.g. medical LLM making a hallucinated diagnosis), it is valuable for a model to abstain from making a prediction when it is uncertain.

Formally, given predicted probabilities $p \in [0, 1]^N$ and true labels $Y \in \{0, 1\}^N$, let $\pi$ be the permutation that sorts $p$ in an ascending order such that $p_{\pi(1)} \leq p_{\pi(2)} \leq \cdots \leq p_{\pi(N)}$. For a rejection rate $\tau$ (where $0 \leq \tau \leq 1$),

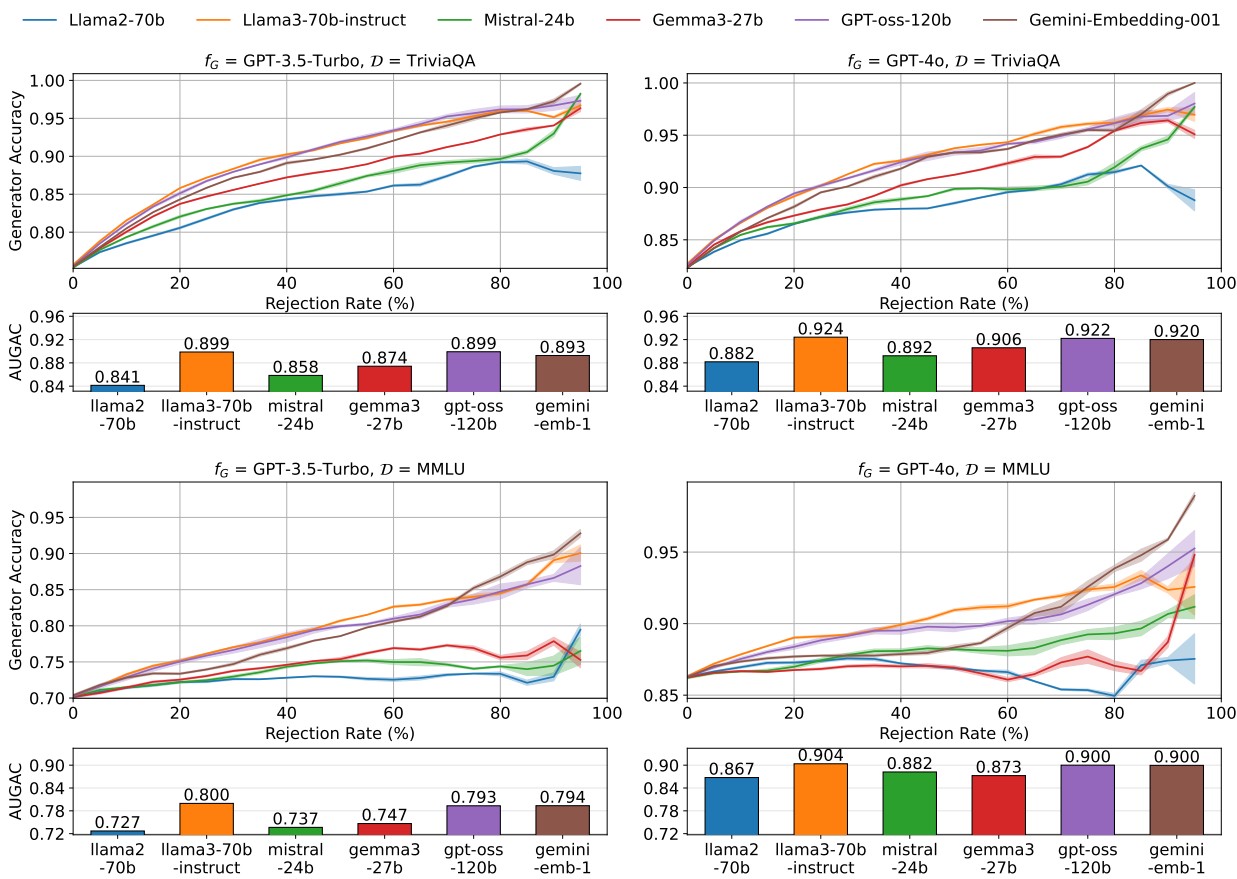

Figure 2: Upper: Selective generator accuracy on TriviaQA and MMLU for GPT-3.5-turbo and GPT-4o under different rejection thresholds. Each curve corresponds to a single representative "best" predictor, chosen as the largest or most recent model within each predictor family or generation. Generators can often achieve about 90% accuracy with a rejection rate less than 40%. Lower: Area under the selective generator accuracy curve (AUGAC) for the same set of predictors. The largest gains come from Llama3-70b-Instruct compared to Llama2-70b, while differences across the other groups of models are more modest. Generator accuracy and AUGAC are reported in proportion units. Full results for all 15 predictors are shown in Appendix B.

we reject the lowest $\lfloor \tau N \rfloor$ predictions and define the selective generator accuracy as

$$S(\tau) = \frac{1}{N - \lfloor \tau N \rfloor} \sum_{i=\lfloor \tau N \rfloor+1}^{N} \mathbb{1}\{Y_{\pi(i)} = 1\}, \tag{5}$$

where $\mathbb{1}\{\cdot\}$ is the indicator function. We compute $S(\tau)$ from $\tau = 0\%$ to $99\%$, and then use the area under $S(\tau)$ to yield the average selective generator accuracy.

## 4 When Do Weak Predictor Models Provide Useful Confidence Scores?

In this section, we empirically analyze when weaker predictor models can produce confidence scores that help identify likely errors from stronger generator models. We consider a selective prediction setting, in which a "weak" predictor model is used to decide when not to trust a "strong" generator model. This setting is practically appealing because weak-to-strong confidence prediction can improve generator reliability when deployed on a stable target distribution.

Table 2: AUROC comparison between weak-to-strong probe predictors and black-box multi-sample baselines, with ($f_G$ = GPT-4o). Stronger probe backbones are competitive with the baselines on knowledge retrieval datasets, but they are not uniformly better, particularly on reasoning datasets. SC and DA performs unusually strong on TriviaQA benchmark due to its free response format, as naï string matching can split many incorrect open-ended answers into small separate clusters, artificially inflating the results.

| Baseline Method / $f_p$ | TriviaQA | CounterFact | Winogrande | MMLU | MedQA | SuperGPQA |
|---|---|---|---|---|---|---|
| Self-consistency (SC) | 92.42 | 83.50 | 60.29 | **71.95** | 62.54 | 67.52 |
| Disagreement (DA) | 92.37 | 83.46 | 61.48 | 71.76 | 64.51 | **68.62** |
| Confidence Prompting (CP) | 56.12 | 68.03 | 64.41 | 57.95 | 55.89 | 55.16 |
| Llama3-70b-instruct | **81.28** | **85.20** | **65.46** | 65.86 | **66.15** | 64.85 |

First, we investigate whether a simple probe can provide useful confidence scores for predicting strong-generator correctness. We then examine which factors influence a predictor model's effectiveness in providing accurate confidence scores and improving generator selective prediction performance. Specifically, we investigate the role of model size (Section 4.2), the effect of label distribution (Section 4.3), and the contribution of different embedding aggregations (Section 4.4).

## 4.1 Does Weak-to-Strong Confidence Prediction Improve Generator Reliability?

To assess predictor performance, we use its predictions to filter out generator answers it identifies as likely incorrect. We assess performance in terms of the generator model's selective accuracy, where a certain threshold of low-confidence answers are rejected based on predictor confidence scores. We plot the selective generator accuracy curve of TriviaQA and MMLU under different thresholds in Figure 2. The curves show that selective generator accuracy increases with higher rejection rates.

To obtain a single evaluation metric that reflects the generator models' selective prediction performance using weak-predictor confidence scores, we compute the area under the selective generator accuracy curve (AUGAC). While absolute AUGAC values vary depending on each task's baseline accuracy, AUGAC is consistently higher than standard generator accuracy for all representative weak predictors shown in Figure 2. The best improvements over the initial generator accuracy can be found in Table 1, and full AUGAC for all predictor backbones are reported in Tables 6 and 7 in Appendix B.

We present results on additional datasets and all 15 models in Appendix B. These results demonstrate substantial gains from weak predictors in selective prediction on both factual retrieval and logical reasoning tasks, with larger improvements on the former. The largest improvements are observed on TriviaQA, where the strongest predictors achieve AUGAC exceeding 90% with an initial generator accuracy of only 70%. In contrast, weaker backbones like Llama2-70b yield more limited gains, resulting in noticeable gaps between the selective accuracy curves across predictors.

Next, we compare our method, in terms of AUROC, against simple black-box baselines based on multiple sampled outputs. Self-consistency (SC) (Wang et al., 2023) is the fraction of samples assigned to the majority answer, while disagreement (DA) (Kuhn et al., 2023) is the negative entropy of the resulting empirical answer distribution. For these two baselines, we first normalize the sampled responses and group identical normalized strings as the same answer. We additionally consider confidence prompting (CP), following Kadavath et al. (2022), in which the generator performs a post-hoc true or false self-evaluation of a proposed answer and this judgment is used as a confidence score. Unlike our probes, all these baselines require $k$ generator calls at inference time. Results for GPT-4o are shown in Table 2, with full results for both generators in Appendix H.

Overall, the trained probe is competitive with the multi-sample black-box baselines, but it does not uniformly outperform them across datasets. Excluding TriviaQA, the probe performs better on the knowledge retrieval datasets, whereas SC or DA can be higher on reasoning datasets such as MMLU and SuperGPQA. TriviaQA is a special case: SC and DA are unusually strong because naive string matching groups many incorrect free-response answers into small separate clusters, which can artificially inflate both results. Thus, while the probe is not uniformly superior in AUROC, it remains competitive and has a practical deployment advantage:

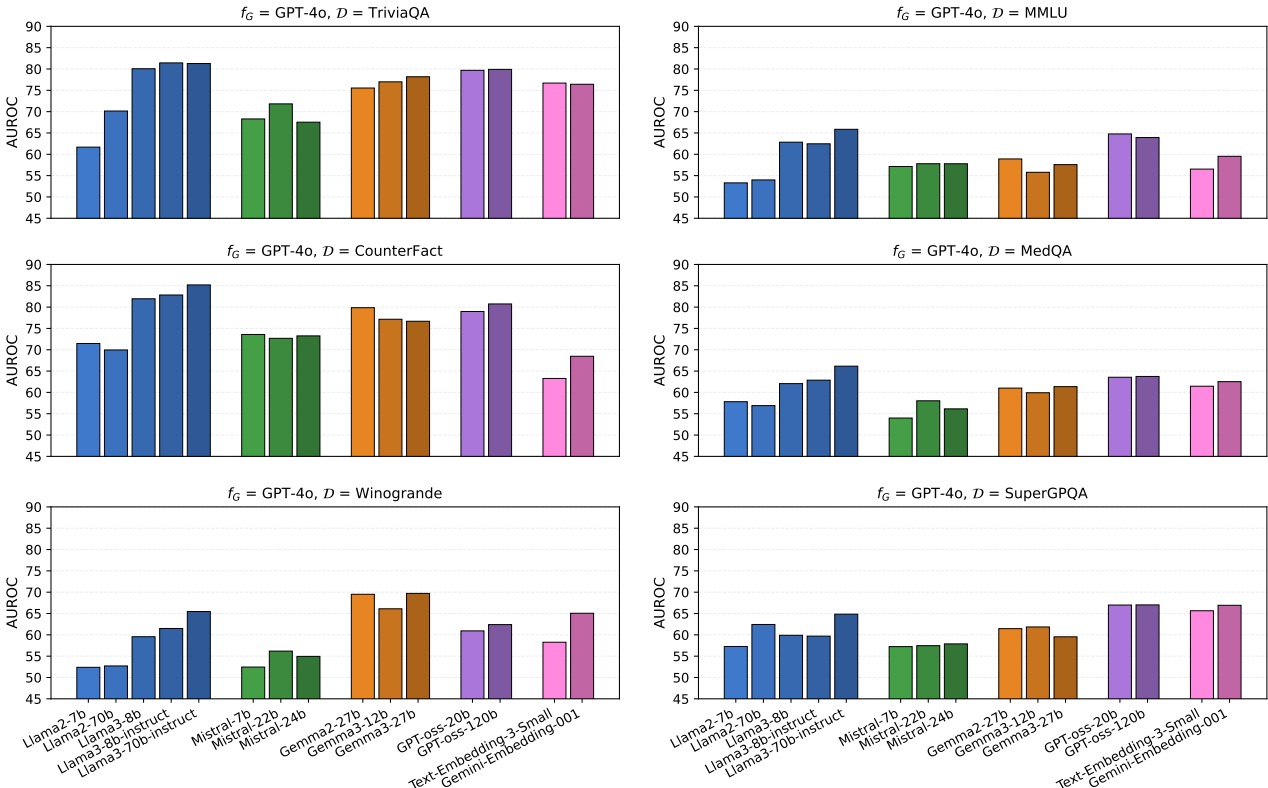

Figure 3: Evaluations across all six datasets with different predictors. For Llama models (blue), as the model size of the weak predictor model, $f_P$, increases, a significant improvement in AUROC can be observed. However, this is generally not true for Mistral and Gemma models. It is also hard to find a consistent scaling trend between different model groups within each task. Similar results can be observed on other metrics like balanced accuracy. For more visualized results, please refer to Figures 13 to 18 in Appendix C.

repeated generator samples are used only upfront to construct training labels, after which confidence for new questions from the target distribution can be predicted from weak-model embeddings without repeated generator sampling.

**Takeaway:** Weak-to-strong confidence prediction can improve strong generator reliability in selective prediction by providing useful in-distribution confidence scores.

## 4.2 Does Weak-to-Strong Confidence Prediction Improve with Predictor Model Size?

Figure 3 shows that our predictors perform better on factual retrieval datasets than on logical reasoning datasets. We believe there are several plausible reasons for this. Question-only embeddings may capture topical content and coarse difficulty, but they do not condition on latent intermediate steps that determine correctness in multi step reasoning. In addition, there may be multiple failure modes in the latent reasoning path (Song et al., 2026; Yang et al., 2024; Li et al., 2024), increasing difficulty in separability for probes. Finally, latent reasoning chains may lead to more stochastic generations, producing noisier generator answers.

We examined the stochasticity hypothesis and quantified the effect using empirical answer disagreement entropy. The results, included in Appendix D, demonstrate that generator answers on reasoning datasets are more stochastic and less identifiable from question-only features.

A central question in our analysis is whether increasing the size of the predictor backbone improves performance. Within the Llama family, more capable backbones result in higher AUROC. Notably, Llama 3 embeddings provide a clear boost over Llama 2, even though Llama 3's generation accuracy improves only modestly (Meta, 2024). Other metrics, such as class-balanced accuracy, exhibit similar scaling trends as the predictor backbone becomes larger and more capable (see Figure 13 to Figure 15 for more details.)

To test whether this scaling effect appears specific to Llama or universal, we experiment with the Mistral, Gemma, and GPT-oss families, and additionally the two embedding-only backbones. We select Mistral and Gemma models with ascending model size and different generations, where their generation performance also improves with size (Gemma Team and Google DeepMind, 2025), but we do not observe the same consistent improvements (see Figures 3 and 13 to Figure 18). In fact, scaling trends vary across tasks; for example, the larger and more recent Gemma sometimes underperforms its smaller counterparts, while in other cases it even surpasses Llama3-70b.

This raises a broader concern: while the LLMs are trained to be increasingly capable at answering the questions, the representations they learn to *map questions to answers* are not necessarily aligned with the task of *mapping questions to probabilities of correctness*. This may explain why the scaling behavior observed in generation ability does not consistently transfer to probes built on model embeddings. To induce a predictor scaling behavior that mirrors LLM scaling, one would likely need to train probes on richer, model-internal signals that underlie an LLM's own reasoning, rather than relying solely on binary correctness labels.

Moreover, even within the Llama family, the benefits of scaling are modest. *Upgrading from sub-10B to 70B-parameter Llama backbones yields only incremental predictor improvements.* This suggests that, for the evaluation task that is central to our paper, smaller models may be nearly as effective in the setting. We hypothesize that less capable Llama backbones produce representation distributions well aligned with GPT outputs, leaving limited room for further gains.

Finally, the training method of the backbone LLM contributes only marginally to predictor improvements. For instance, Llama3-8b-instruct, which is finetuned via instruction tuning, brings more than 3% improvement than Llama3-8b in AUROC and balanced accuracy on the Winogrande dataset, but is only slightly better on the TriviaQA dataset, where AUROC already exceeds 80%. Similar patterns hold across other metrics, as shown in Table 8 - Table 13, underscoring that predictor performance is not strongly determined by the same factors that drive generation performance.

> **Takeaway:** Unlike in generation tasks, model size alone is not a reliable determinant of predictor performance across architectures.

### 4.3 Are Weak-to-Strong Confidence Predictors Just Learning Label Distributions?

We next test whether predictors truly learn generator- and question-specific features, or whether they simply learn to mimic the overall distribution of correctness labels. To disentangle these effects, we designed two ablation experiments described below.

**Controlling "Difficulty" Levels.** Imbalanced datasets pose a greater challenge for predictors. Predictors may struggle because of the dataset intrinsic difficulty or the skewed label distribution, which often arises because generation quality is influenced by that same intrinsic difficulty.

To isolate the effect of dataset intrinsic difficulty, we collected additional GPT-3.5-turbo responses to construct three TriviaQA subsets with controlled generator accuracies: 60% (*Easy*), 75% (*Medium*), and 90% (*Hard*). By fixing the proportion of correct answers, we disentangle label imbalance effects from dataset-specific features and can better assess their impact on training.

Results are shown in Figure 4. The three gray diamonds mark predictor performance on TriviaQA-*Easy*, *Medium*, and *Hard*. Notably, the *Hard* baseline – where incorrect responses are scarce – remains the most challenging. This setting enables comparisons across datasets with matched "difficulty" level, as each marker can be compared to the nearest TriviaQA-*Easy/Medium/Hard* baseline in terms of generator accuracy. For instance, MedQA with GPT-3.5-turbo achieves 58.91% generator accuracy, close to TriviaQA-*Easy*, yet predictors perform worse on MedQA. This suggests that embeddings from TriviaQA questions are more

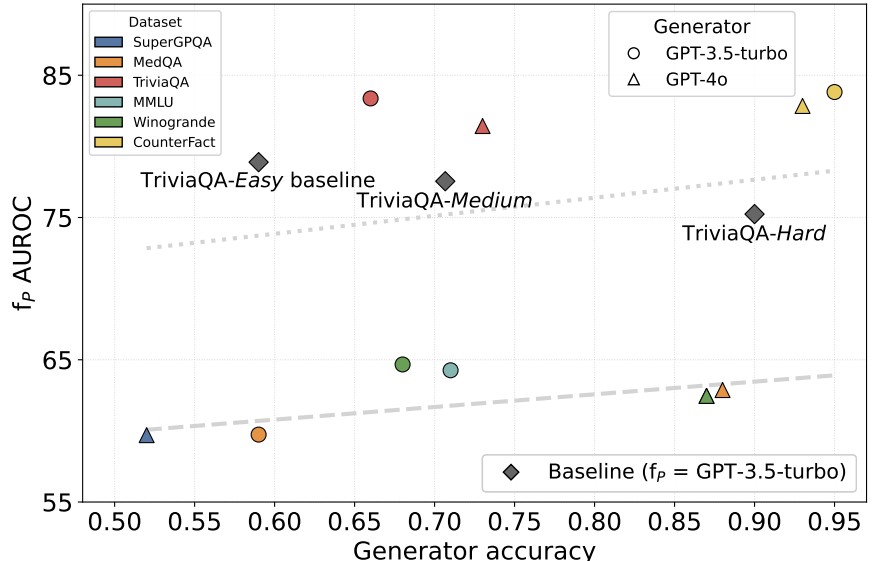

Figure 4: Predictor AUROC (%) using Llama-3-8B-Instruct backbone on six datasets. Generator accuracy is displayed in proportion units. Circle markers denote GPT-3.5-turbo, and triangles denote GPT-4o. Each marker is compared to the nearest TriviaQA-*Easy/Medium/Hard* baseline in terms of generator accuracy. Dotted and dashed lines fit knowledge recall and reasoning datasets. No statistically significant trend is observed. Complete baseline results are provided in Appendix F.

predictive of correctness than those from MedQA. A similar pattern appears in the *Hard* cluster: datasets such as MMLU, MedQA, and Winogrande (all near 90% generator accuracy) yield predictor AUROCs of only around 65%, far below CounterFact or TriviaQA-*Hard*.

**Controlling Exact Label Ratios.** We also tested three variants of TriviaQA and MMLU: a balanced 50/50 subset (*50/50*), one matching the training label ratio (*Exact ratio*), and the full test set (*Original*). The original training set had 8,000 samples, and the test set 2,000. This removes natural imbalance while preserving dataset content.

This experiment probes a different aspect than the difficulty study. While the earlier experiment varied dataset difficulty across tasks, here we fix the dataset (TriviaQA or MMLU) and vary only the input-output mapping via generator labels (GPT-3.5-turbo vs GPT-4o). Thus, inputs are drawn from the same distribution, and differences reflect whether the predictor learns meaningful associations between embeddings and correctness.

Results (Tables 21 and 23) show that predictors behave differently for GPT-4o versus GPT-3.5-turbo labels, confirming that they exploit input features rather than only memorizing label priors. Moreover, performance on the *Original* and *Exact ratio* test sets is comparable, with the *Original* sometimes slightly worse. This indicates that predictors trained on balanced data still generalize well to natural, imbalanced distributions. We refer readers to Table 21–Table 27 for full results and label distribution details.

> **Takeaway:** Predictors learn generator- and question-specific features rather than simply imitating the label distributions.

## 4.4 Does Weak-to-Strong Confidence Prediction Depend on Embedding Choice?

We have been using the embedding of the last non-padding token of each question as inputs for the probe layers. While this is common practice in autoregressive models (Kadavath et al., 2022), when the context becomes long it may not always capture the most relevant information. Ahdritz et al. (2024) studies how embeddings from different layers may impact the performance of downstream layers on different tasks.

Following this example, we used Llama models to experiment and compare middle layer embeddings with last layer embeddings as inputs to our predictors.

We also experimented with different pooling strategies to aggregate the embeddings, including average pooling and max pooling. We find that only average pooling and intermediate embeddings shows a minor improvement, and results are given in Appendix G. Overall, the choice of aggregation strategy makes only a small difference. Crucially, the same scaling trends observed earlier persist across all aggregation methods, suggesting that backbone model quality remains a more important factor.

> **Takeaway:** Different embedding aggregations produce broadly consistent results, suggesting that predictor performance is robust to the choice of aggregation method.

### 4.5 Does Predictor Performance Depend on Sampling and Decoding Choices?

Because $y_{\text{true}}$ is constructed from multiple sampled generator outputs, its value can vary with the decoding strategy and the sampling budget. To address this concern, we conduct a comprehensive sensitivity analysis over the sampling temperature $\in \{0.2, 0.8, 1.0, 1.2\}$, the nucleus sampling threshold (Holtzman et al., 2020) $top\_p \in \{0.9, 1.0\}$, and the sampling budget $k \in \{5, 10, 20\}$. Across all datasets, generators and predictors, the main qualitative findings remain consistent, indicating that our conclusions are broadly stable and do not rely on specific sampling hyperparameter choices. The complete results are provided in Appendix I.

Overall, we find that smaller predictor backbones such as Llama3-8B still provide effective weak-to-strong confidence signals, and that knowledge retrieval tasks are consistently easier to predict than reasoning benchmarks. Reducing the sampling budget from 10 to 5 typically leads to a small decrease in AUROC, while increasing it from 10 to 20 often produces modest improvements. However, neither change affects the overall benefits of selective prediction or alters the main conclusions regarding representational compatibility.

> **Takeaway:** Predictor performance is driven primarily by the quality of the underlying representations, rather than by any particular choice of decoding hyperparameters or sampling budget.

Table 3: Cross dataset transfer matrix for out-of-distribution (OOD) probe evaluation. Rows denote the probe training dataset and columns denote the evaluation dataset. Entries are AUROC on the target dataset without further fine-tuning. Diagonal entries correspond to in-distribution evaluation; off-diagonal entries correspond to zero-shot transfer. For illustration, here $f_G$ = GPT-4o, and $f_P$ = Llama3-70b-Instruct.

| Source \| Target | TriviaQA | CounterFact | Winogrande | MMLU | MedQA | SuperGPQA |
|---|---|---|---|---|---|---|
| TriviaQA | 81.3 | 47.0 | 52.0 | 57.0 | 56.1 | 50.8 |
| CounterFact | 50.6 | 85.2 | 47.8 | 58.0 | 51.7 | 50.2 |
| Winogrande | 48.9 | 45.8 | 65.5 | 50.4 | 50.1 | 51.3 |
| MMLU | 46.0 | 51.8 | 51.7 | 65.9 | 55.2 | 62.8 |
| MedQA | 53.5 | 53.0 | 49.9 | 55.8 | 66.2 | 61.0 |
| SuperGPQA | 59.7 | 44.2 | 51.4 | 53.7 | 52.9 | 64.8 |

### 4.6 Does Weak-to-Strong Confidence Prediction Transfer Across Datasets?

The preceding experiments evaluate probes using train and test splits from the same benchmark. We next ask whether the learned confidence signal transfers across benchmarks. Specifically, we used probes trained on one dataset and evaluated their AUROC performance on all target datasets without any further fine-tuning. Table 3 reports AUROC for Llama3-70B-Instruct as the predictor backbone and GPT-4o as the generator.

The results show limited cross dataset transfer. The diagonal entries recover the in-distribution performance reported earlier, while the off-diagonal entries are substantially lower and often only modestly above chance. This suggests that the uncertainty signal learned by the probe is not strongly transferable across benchmarks,

likely because the probe only sees question embeddings (Section 3.2) and relies on benchmark-specific regularities in question style, domain, format, and difficulty (Section 3.1). At the same time, the strong diagonal performance suggests that weak model representations do contain useful signals for uncertainty prediction, and that the probe can recover these signals within an in-domain target distribution.

These results suggest that the probe is best understood as an in-distribution confidence estimator rather than a benchmark-independent OOD uncertainty estimator. We also note that the probe is lightweight and inexpensive to train per target distribution once generator answers can be sampled. In practice, we recommend training a separate probe in the current setting for a stable target distribution after collecting generator responses from that distribution, rather than training once and deploying across unrelated domains.

> **Takeaway:** Weak-to-strong confidence prediction has limited cross-dataset transfer. It is useful for in-distribution confidence estimation, not benchmark-independent OOD uncertainty prediction.

## 5 Discussion and Limitations

First, because the predictor only sees the question, it often underperforms on reasoning tasks where crucial evidence arises in intermediate solution steps. This limitation is more visible on difficult reasoning benchmarks such as SuperGPQA. Second, the learned confidence signal transfers poorly across datasets: cross-dataset evaluations are often close to chance, suggesting that the probes rely substantially on benchmark-specific regularities and are best viewed as in-distribution confidence estimators rather than benchmark-independent OOD uncertainty estimators. Third, while we observe a slight scaling trend with Llama models, this trend does not generalize across broader model families. Finally, no single factor clearly determines predictor performance: model size, label balance, embedding aggregation, and hyperparameter selection each have only minor effects.

A main limitation of our analysis is that we study only frozen open-access backbones with a lightweight probe classifier. This design makes the setting simple and computationally accessible, but it also limits what information the predictor can use: the probe cannot adapt the backbone representation, access the generator's internal states, or condition on the generated reasoning trace. As a result, our conclusions should be interpreted as evidence about what can be recovered from frozen question embeddings, leaving open whether richer predictors could capture additional confidence signals.

These considerations suggest several future research directions. On the architecture side, the choice of probe can be refined. Replacing the MLP with more complicated structure has shown promise in recent works (CH-Wang et al., 2024; Psomas et al., 2025), but with the increased parameter count, it is unclear whether our dataset is large enough to fit such models. Additionally, answer-aware weak-to-strong confidence prediction may be a promising direction for improving transfer across different tasks (e.g., TriviaQA to CounterFact) and categories (TriviaQA to SuperGPQA). Given the limited cross-dataset transfer observed in our current setting, demonstrating generalization to unseen benchmarks would be practically valuable. We release our datasets to support these investigations and to provide a clean baseline for research into weak-to-strong confidence prediction.

## 6 Conclusion

In this paper, we studied weak-to-strong confidence prediction–the task of using a weak predictor model to produce confidence scores that can be used to predict whether a stronger generator model will answer a given question correctly. In our empirical evaluation, we found that weak-to-strong confidence prediction can improve strong generator reliability in in-distribution selective prediction settings. We hope this work will encourage future research into weak-to-strong confidence prediction as a step towards building safer and more reliable frontier models.

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

# Appendix

## Table of Contents

## A Reproducibility

We have made a significant effort to ensure the reproducibility of our results. The implementation of our method is provided at GitHub, which includes training, evaluation, and analysis scripts. The experimental setup—including hyperparameters, model configurations, and compute requirements are described in Section 3 and appendix A.1. All datasets used in our experiments are also provided in the repository, and we additionally provide instructions for reproducing predictor experiments. Our experiments rely on open-access models and certain API-only models; instructions for reproducing results with both are included in the `README.md` file in the released code. Finally, the datasets constructed for this study are released with the code repository, enabling direct benchmarking of our results.

### A.1 Details on Dataset Construction and Experimental Setup

We collect questions from HuggingFace datasets and obtain answers from GPT models using OpenAI APIs. For MMLU, we prioritize questions from specific categories (see: Tables 4 and 5) and continue sampling from *auxiliary_train* until reaching 10k questions in total. For other datasets, we simply take the first 10k questions. SuperGPQA includes between 4 to 10 answer choices per question; we standardize this by randomly selecting three incorrect answers and shuffling them with the correct one. All other datasets contain four or fewer choices, which we retain as-is. We split the data into training and test sets with an 8:2 ratio, where the exact partitioning is determined by the random seed.

All predictor implementations are based on PyTorch, and all question representations are collected from pretrained models on Huggingface. We train the probes with a learning rate of $1 \times 10^{-5}$, dropout rate of 0.2,

weight decay of 0.01, cosine learning-rate scheduler, and batch size of 32. Each configuration is trained with three random seeds, and we report the mean performance across runs. The best-performing model is selected based on AUROC.

We use a cross entropy objective with class-weighted loss to address label imbalance. The class-0 weight is tuned within [0.5, 10.0]; while a suitable choice slightly improves performance, it does not substantially affect our conclusions.

All open-weight predictor models are loaded in 4-bit precision with float16 computation.

Table 4: Selected MMLU Subsets (Part 1)

| Subset |
| --- |
| anatomy |
| astronomy |
| business ethics |
| clinical knowledge |
| college biology |
| college chemistry |
| college medicine |
| econometrics |
| global facts |
| high school biology |
| high school chemistry |
| high school european history |
| high school geography |
| high school government and politics |
| high school macroeconomics |
| high school microeconomics |
| high school psychology |
| high school us history |
| high school world history |
| human aging |

Table 5: Selected MMLU Subsets (Part 2)

| Subset |
| --- |
| human sexuality |
| international law |
| jurisprudence |
| management |
| marketing |
| medical genetics |
| miscellaneous |
| nutrition |
| philosophy |
| prehistory |
| professional law |
| professional medicine |
| public relations |
| sociology |
| us foreign policy |
| virology |
| world religions |
| conceptual physics |

# B Selective Generator Accuracy

We report full experimental results and visualizations for the evaluation in Section 4.1. For cross-family comparison, Mistral and Gemma results are presented together in Figures 9 and 10, and GPT-oss and the two embedding models are presented together in Figures 11 and 12.

Table 6: Selective prediction performance using the strong generator model $f_G$ = GPT-3.5-turbo, measured by the area under the generator accuracy curve (AUGAC), across six datasets and fifteen open-access backbones.

| $f_P$ | TriviaQA | CounterFact | Winogrande | MMLU | MedQA | SuperGPQA |
|---|---|---|---|---|---|---|
| Llama2-7b | 82.11 | 96.65 | 74.06 | 73.65 | 62.94 | 36.34 |
| Llama2-70b | 84.14 | 96.77 | 74.05 | 72.70 | 64.00 | 40.71 |
| Llama3-8b | 89.32 | 98.09 | 78.44 | 76.57 | 65.88 | 39.37 |
| Llama3-8b-instruct | 90.08 | 98.02 | 79.05 | 76.86 | 66.57 | 41.69 |
| Llama3-70b-instruct | 89.87 | 98.54 | 83.15 | 79.95 | 70.69 | 40.45 |
| Mistral-7b | 82.54 | 96.66 | 74.50 | 72.51 | 64.13 | 39.19 |
| Mistral-22b | 83.49 | 95.94 | 73.94 | 72.60 | 63.85 | 39.58 |
| Mistral-24b | 85.85 | 96.93 | 74.05 | 73.65 | 60.77 | 41.32 |
| Gemma2-27b | 87.16 | 96.97 | 73.08 | 78.08 | 68.76 | 41.49 |
| Gemma3-12b | 86.15 | 97.31 | 75.85 | 75.36 | 66.76 | 41.01 |
| Gemma3-27b | 87.42 | 96.89 | 73.23 | 74.66 | 66.45 | 40.51 |
| GPT-oss-20b | 89.28 | 97.80 | 78.85 | 78.65 | 68.15 | 41.93 |
| GPT-oss-120b | 89.91 | 98.07 | 79.16 | 79.33 | 68.95 | 42.87 |
| Text-Embedding-3-Small | 89.06 | 97.22 | 77.47 | 78.51 | 67.39 | 42.38 |
| Gemini-Embedding-001 | 89.26 | 97.97 | 80.66 | 79.35 | 68.36 | 43.19 |

Table 7: Selective prediction performance using the strong generator model $f_G$ = GPT-4o, measured by the area under the generator accuracy curve (AUGAC), across six datasets and fifteen open-access backbones.

| $f_P$ | TriviaQA | CounterFact | Winogrande | MMLU | MedQA | SuperGPQA |
|---|---|---|---|---|---|---|
| Llama2-7b | 86.07 | 97.80 | 89.75 | 86.61 | 90.40 | 53.11 |
| Llama2-70b | 88.18 | 97.36 | 89.65 | 86.75 | 90.38 | 55.06 |
| Llama3-8b | 92.10 | 98.27 | 91.57 | 89.93 | 90.51 | 57.14 |
| Llama3-8b-instruct | 92.83 | 98.47 | 91.00 | 90.49 | 91.01 | 56.33 |
| Llama3-70b-instruct | 92.42 | 98.76 | 93.58 | 90.37 | 91.10 | 56.36 |
| Mistral-7b | 87.04 | 97.49 | 90.45 | 87.11 | 89.81 | 55.29 |
| Mistral-22b | 88.37 | 96.87 | 90.67 | 86.75 | 89.98 | 47.03 |
| Mistral-24b | 89.22 | 97.69 | 90.55 | 88.21 | 89.33 | 29.20 |
| Gemma2-27b | 90.04 | 97.28 | 92.57 | 89.91 | 91.81 | 58.22 |
| Gemma3-12b | 89.98 | 97.39 | 91.71 | 87.88 | 91.18 | 57.05 |
| Gemma3-27b | 90.59 | 97.70 | 91.80 | 87.26 | 91.27 | 56.42 |
| GPT-oss-20b | 91.56 | 98.46 | 92.70 | 90.58 | 91.85 | 57.71 |
| GPT-oss-120b | 92.20 | 98.72 | 93.05 | 90.00 | 92.03 | 57.13 |
| Text-Embedding-3-Small | 91.79 | 96.89 | 91.91 | 88.01 | 91.39 | 57.43 |
| Gemini-Embedding-001 | 92.00 | 97.65 | 94.02 | 89.95 | 91.79 | 58.34 |

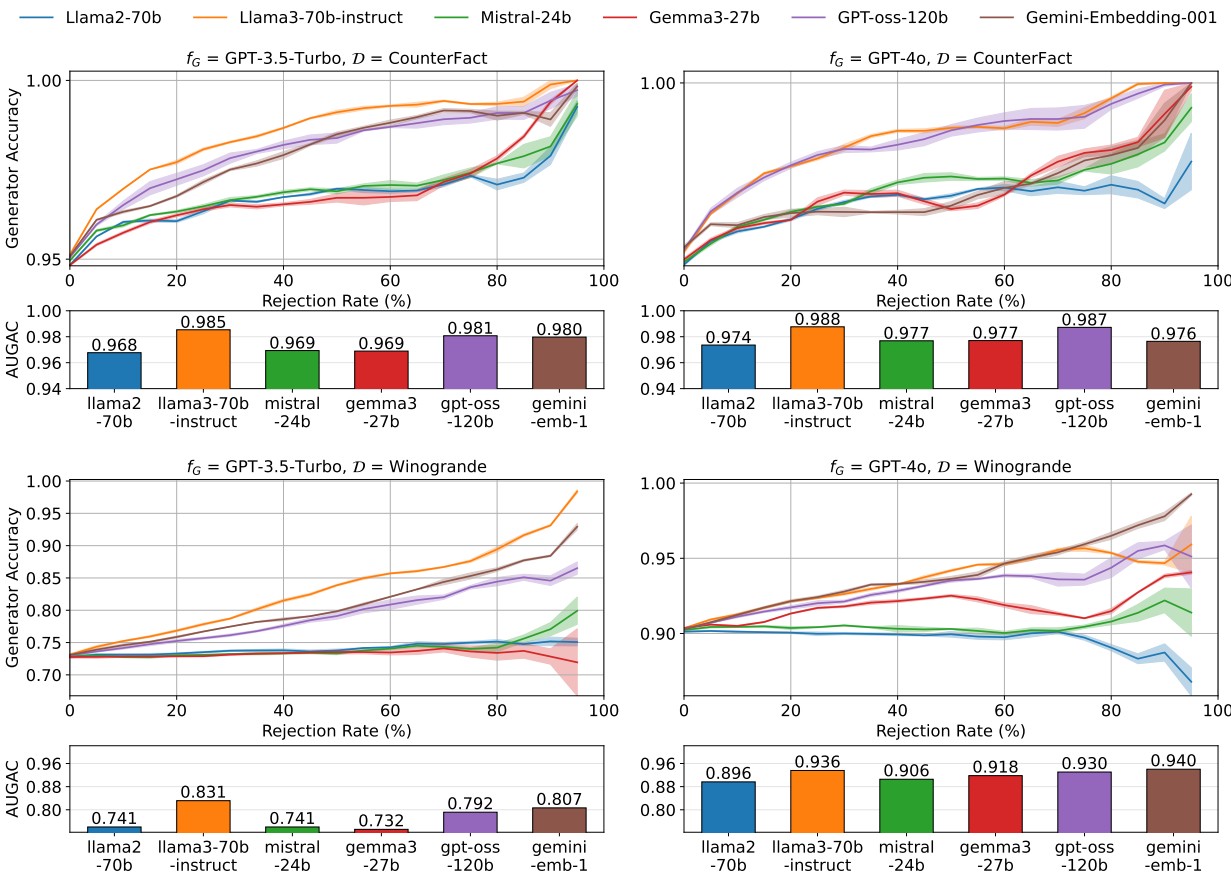

Figure 5: Upper: Selective generator accuracy on CounterFact and Winogrande. Each curve corresponds to one representative "best" predictor: the largest backbone within each predictor family or generation. Lower: Area under the selective generator accuracy curve (AUGAC) for the same predictors. Generator accuracy generally increases as the rejection rate rises, consistent with the pattern observed in Figure 2. Llama3-70b-Instruct provides the strongest improvements, with other families showing smaller but consistent gains. All quantities are shown in proportion units for readability.

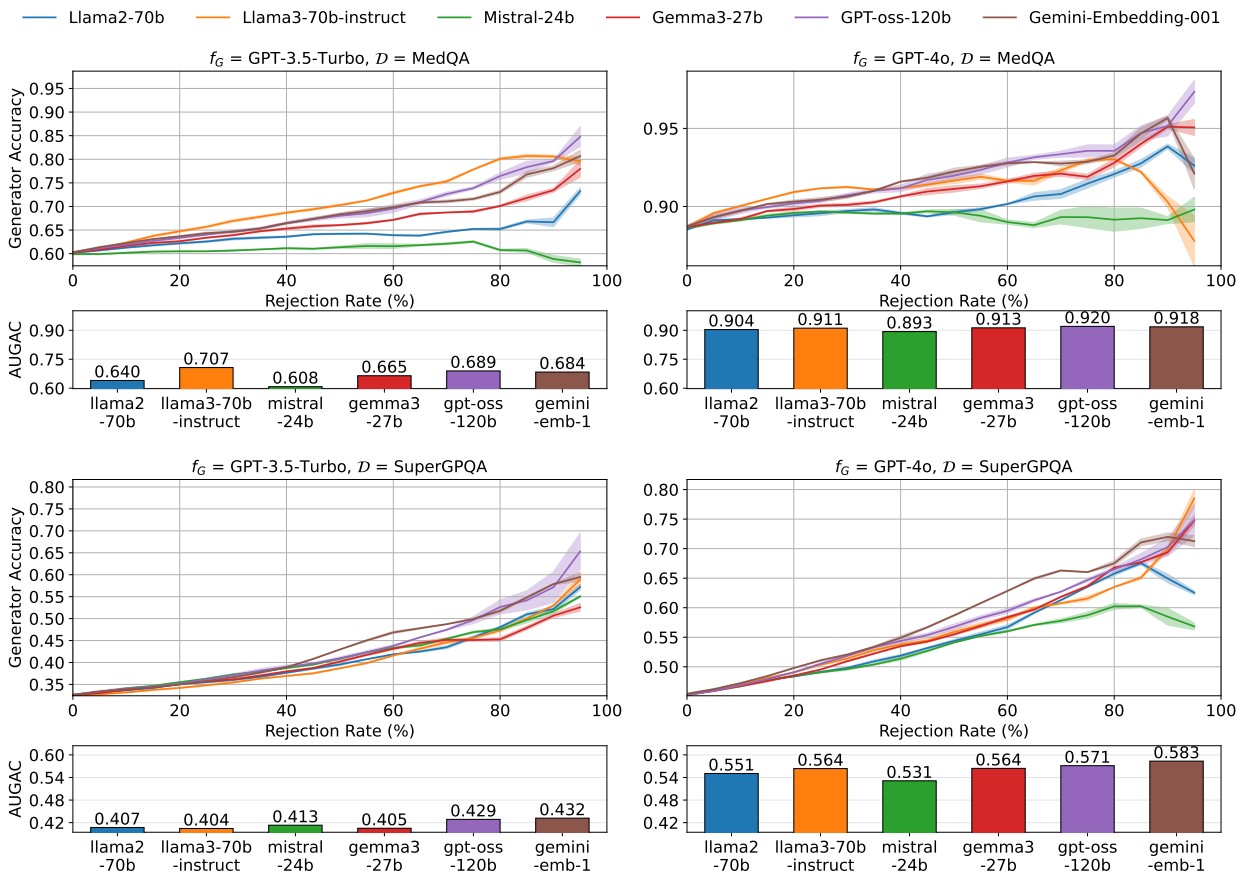

Figure 6: Upper: Selective generator accuracy on MedQA and SuperGPQA. Each curve corresponds to one representative "best" predictor: the largest backbone within each predictor family or generation. Lower: Area under the selective generator accuracy curve (AUGAC) for the same predictors. Generator accuracy generally increases as the rejection rate rises, consistent with the pattern observed in Figure 2. Llama3-70b-Instruct provides the strongest improvements, with other families showing smaller but consistent gains. All quantities are shown in proportion units for readability.

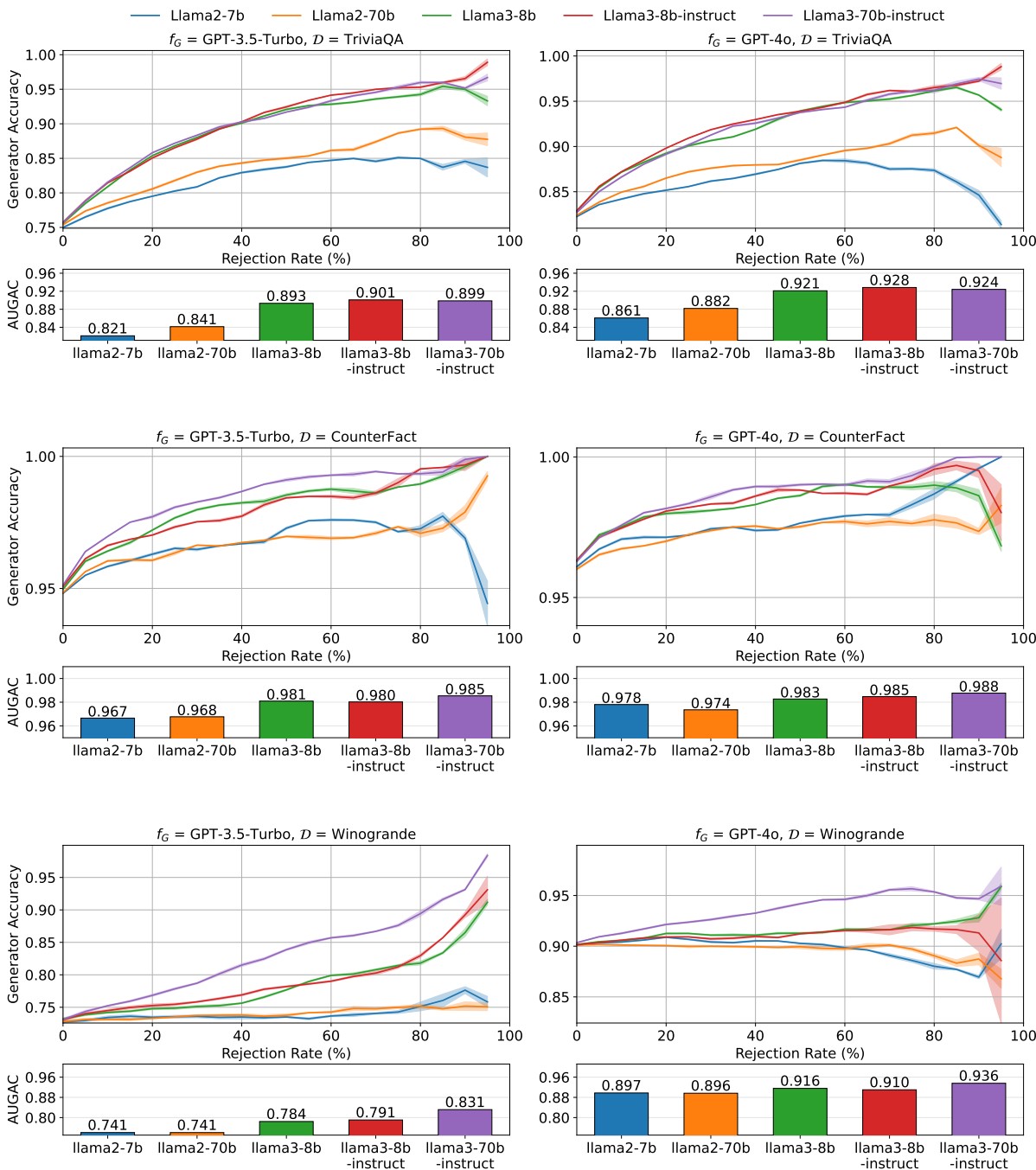

Figure 7: Upper: Generator Accuracy on TriviaQA, CounterFact, and Winogrande. Lower: Area under the generator accuracy curve (AUGAC) for each Llama backbone. The predictor family is Llama. Generator accuracy generally increases as the rejection rate rises, consistent with the pattern observed in Figure 2. The improvement is the strongest between Llama3 and Llama2 backbones. Within the same generation of models, the differences are smaller. Generator accuracy and AUGAC are displayed in proportion units for readability.

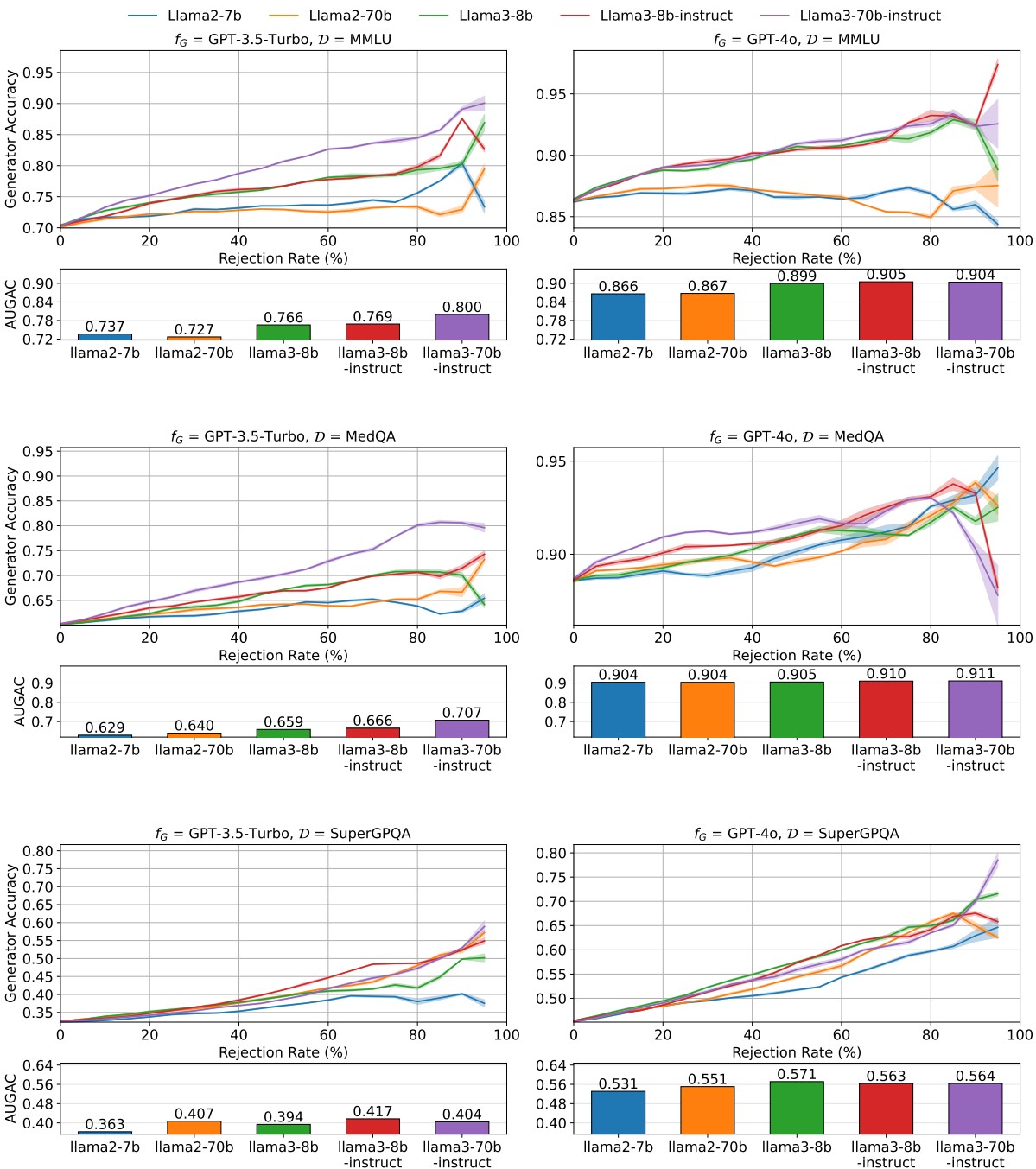

Figure 8: Upper: Generator Accuracy on MMLU, MedQA, and SuperGPQA. Lower: Area under the generator accuracy curve (AUGAC) for each Llama backbone. The predictor family is Llama. Generator accuracy generally increases as the rejection rate rises, consistent with the pattern observed in Figure 2. The improvement is the strongest between Llama3 and Llama2 backbones. Within the same generation of models, the differences are smaller. Generator accuracy and AUGAC are displayed in proportion units for readability.

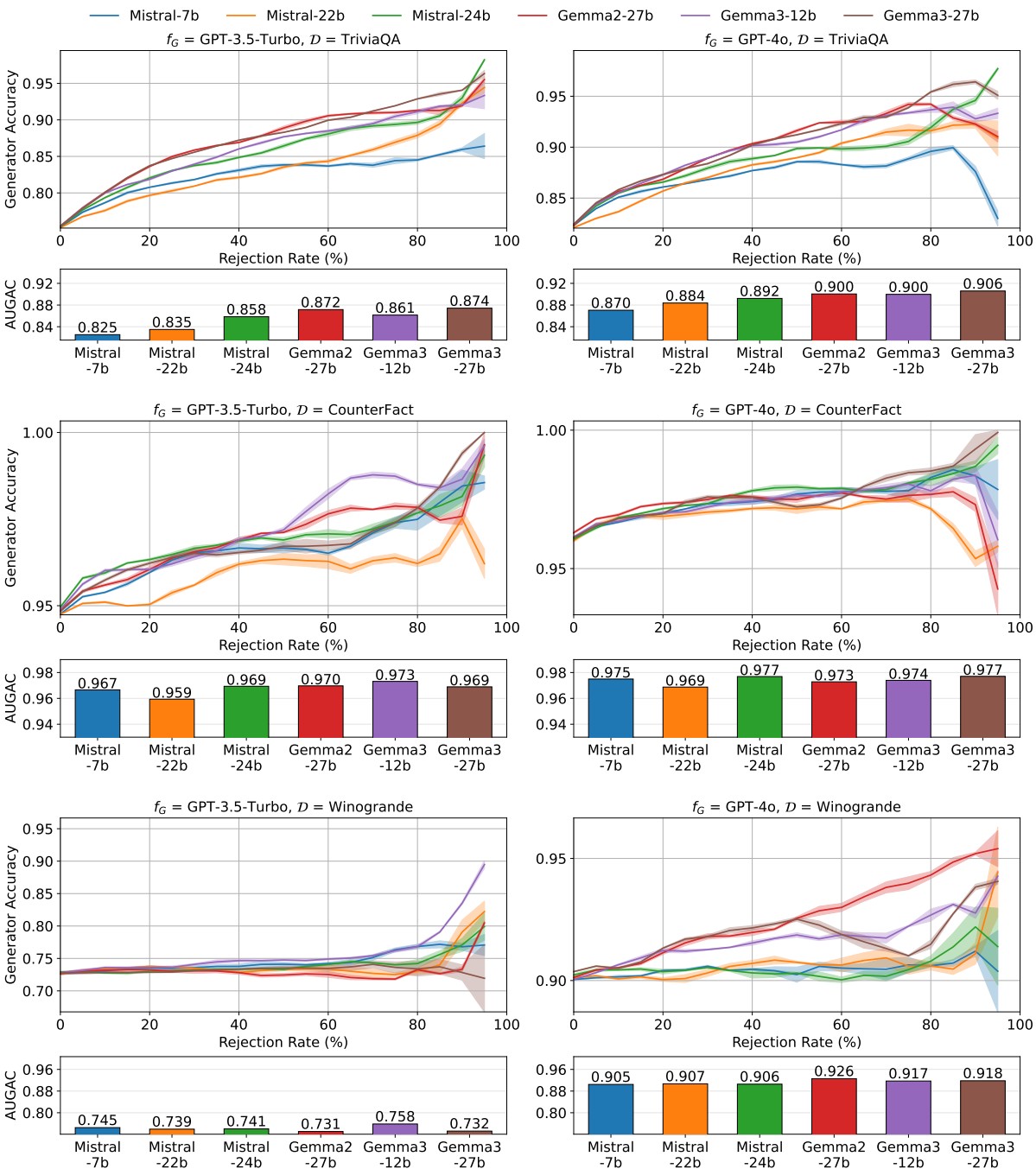

Figure 9: Upper: Generator Accuracy on TriviaQA, CounterFact, and Winogrande. Lower: Area under the generator accuracy curve (AUGAC) for each Mistral and Gemma backbone. The predictors are from the Mistral and Gemma families. Generator accuracy generally increases as the rejection rate rises, consistent with the pattern observed in Figure 2. Cross-generation gains are present but smaller than those observed for Llama (see Figures 2, 7 and 8), and differences within the same generation remain modest. Generator accuracy and AUGAC are displayed in proportion units for readability.

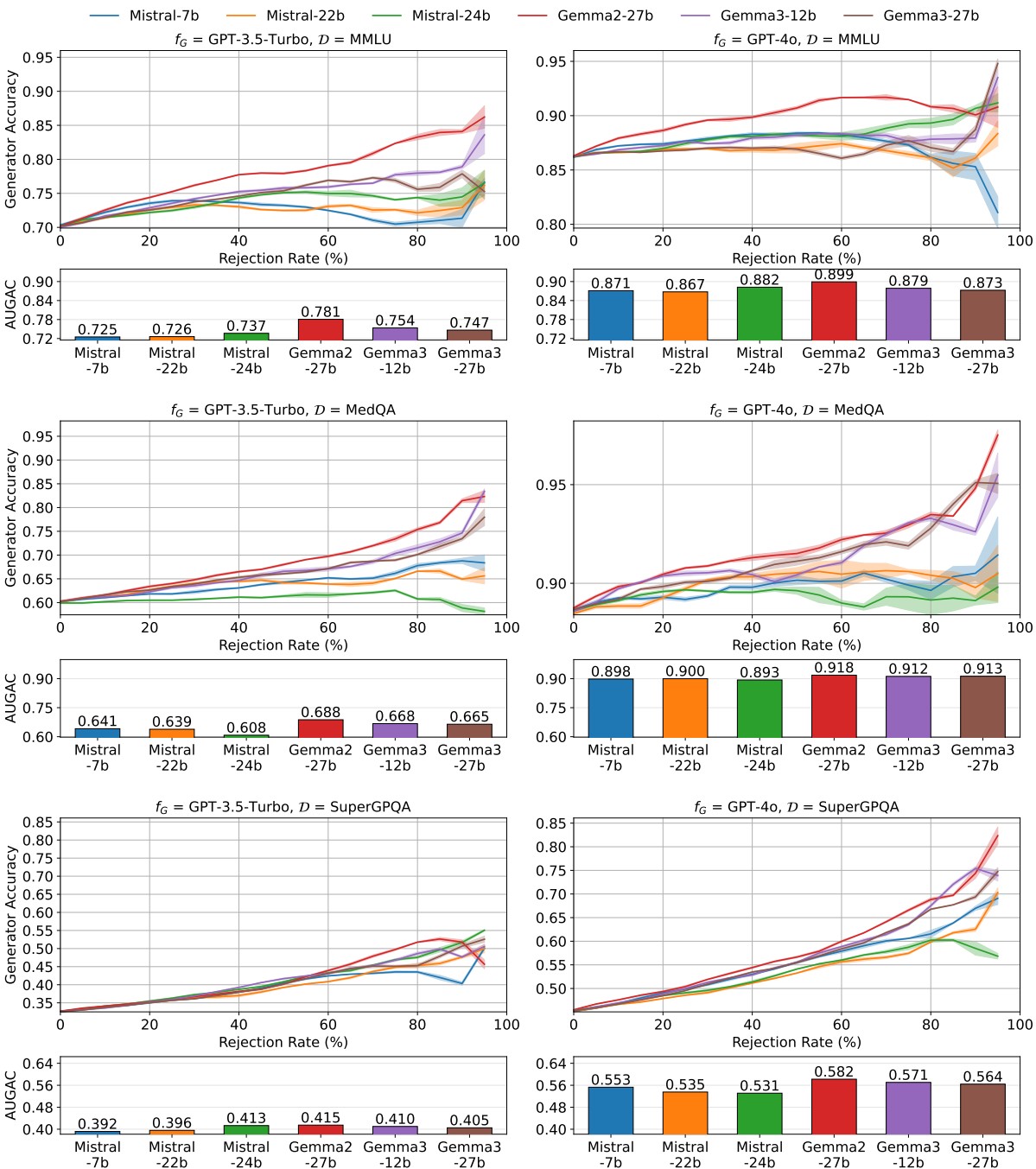

Figure 10: Upper: Generator Accuracy on MMLU, MedQA, and SuperGPQA. Lower: Area under the generator accuracy curve (AUGAC) for each Mistral and Gemma backbone. The predictors are from the Mistral and Gemma families. Generator accuracy generally increases as the rejection rate rises, consistent with the pattern observed in Figure 2. Cross-generation gains are present but smaller than those observed for Llama (see Figures 2, 7 and 8), and differences within the same generation remain modest. Generator accuracy and AUGAC are displayed in proportion units for readability.

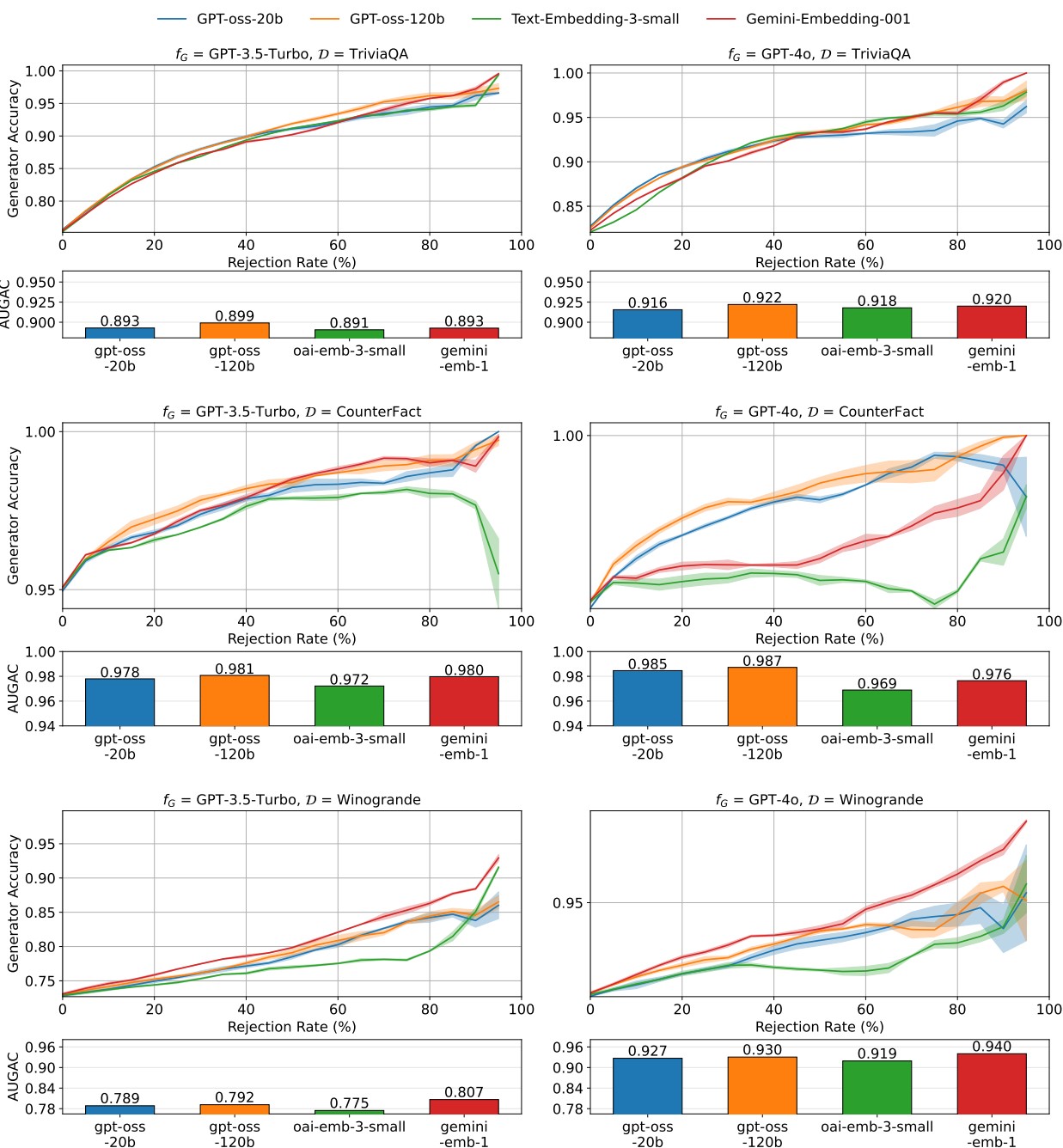

Figure 11: Upper: Generator Accuracy on TriviaQA, CounterFact, and Winogrande. Lower: Area under the generator accuracy curve (AUGAC) for the two GPT-oss backbones and the two embedding models. Generator accuracy generally increases as the rejection rate rises, consistent with the pattern observed in Figure 2. The two embedding models show a modest difference, while the gains for the GPT-oss models are present but smaller than those observed for Llama (see Figures 2, 7 and 8). Generator accuracy and AUGAC are displayed in proportion units for readability.

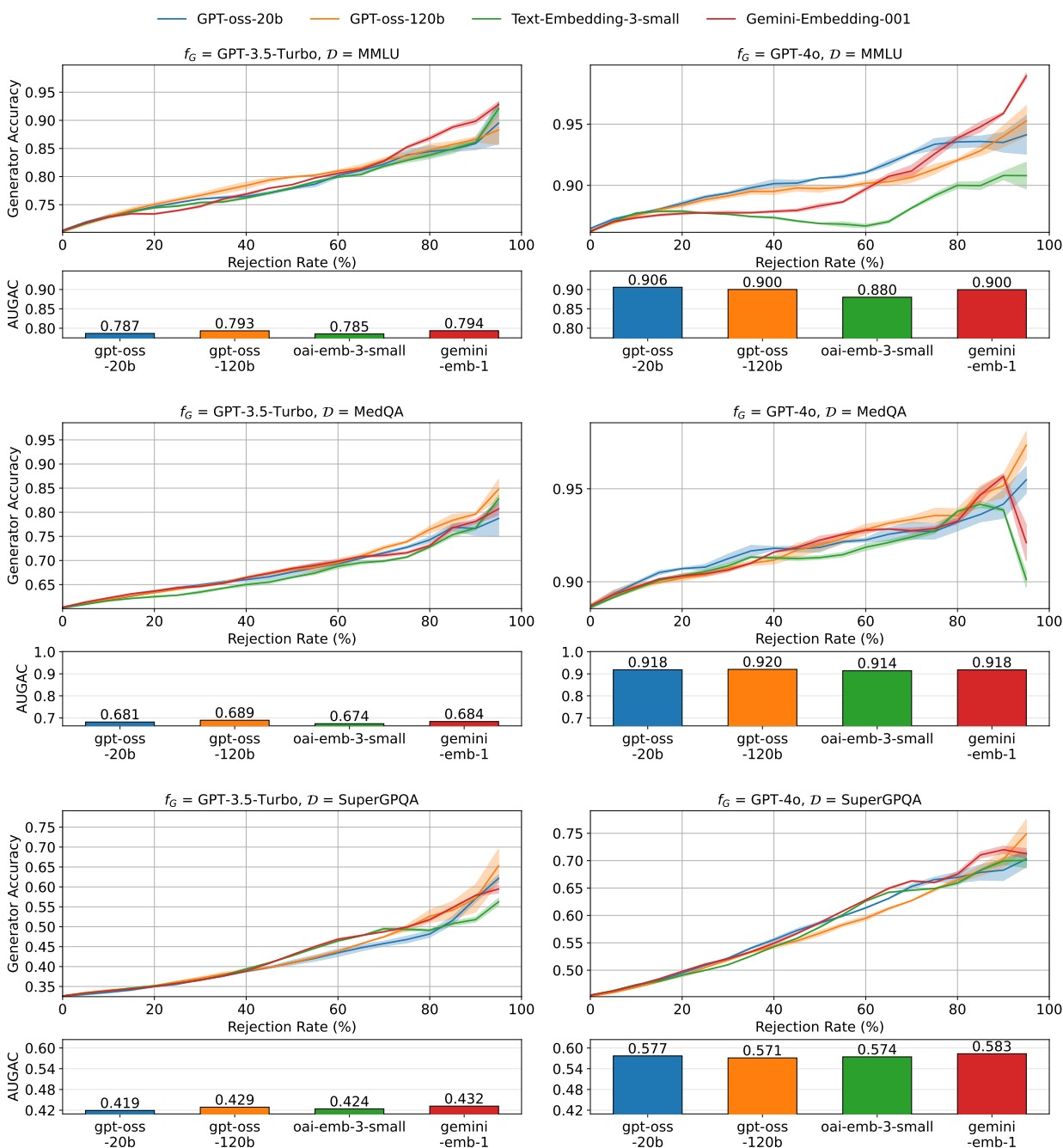

Figure 12: Upper: Generator Accuracy on MMLU, MedQA, and SuperGPQA. Lower: Area under the generator accuracy curve (AUGAC) for the two GPT-oss backbones and the two embedding models. Generator accuracy generally increases as the rejection rate rises, consistent with the pattern observed in Figure 2. The two embedding models show a modest difference, while the gains for the GPT-oss models are present but smaller than those observed for Llama (see Figures 2, 7 and 8). Generator accuracy and AUGAC are displayed in proportion units for readability.

## C    Evaluation of Predictor Performance on Six Datasets

We provide full experimental results and visualizations for our evaluation in Section 4.2 as follows.

Table 8: Predictor performance on TriviaQA-constructed dataset evaluated across nine metrics using the strong generator model $f_G$ = GPT-3.5-turbo. Dataset construction is described in Appendix A.1.

| $f_P$ | AUROC | Accuracy | Balanced Acc. | AUPRC | F1 Score | Sel. Balanced Acc. | Sel. AUROC | Sel. F1 | ECE |
|---|---|---|---|---|---|---|---|---|---|
| Llama3-70b-instruct | 84.15 | 80.77 | 75.34 | 92.80 | 87.09 | 85.64 | 90.12 | 93.62 | 0.5082 |
| Llama3-8b-instruct | 83.37 | 79.88 | 75.05 | 92.67 | 86.39 | 85.18 | 89.39 | 91.33 | 0.4834 |
| Llama3-8b | 82.47 | 78.65 | 74.02 | 91.89 | 85.31 | 84.93 | 87.99 | 92.17 | 0.4774 |
| Llama2-70b | 71.08 | 72.58 | 65.39 | 85.59 | 81.39 | 72.32 | 72.20 | 88.29 | 0.4519 |
| Llama2-7b | 63.34 | 71.75 | 59.60 | 81.30 | 82.85 | 65.01 | 65.58 | 84.21 | 0.3579 |
| Mistral-7b | 67.62 | 74.82 | 62.08 | 84.23 | 84.91 | 69.08 | 72.78 | 89.06 | 0.0000 |
| Mistral-22b | 67.99 | 73.47 | 63.96 | 84.69 | 83.03 | 70.16 | 73.63 | 88.35 | 0.0027 |
| Mistral-24b | 74.42 | 76.97 | 68.90 | 87.78 | 85.47 | 76.80 | 78.73 | 90.94 | 0.0000 |
| Gemma2-27b | 77.55 | 78.48 | 70.84 | 89.42 | 85.83 | 80.54 | 84.11 | 91.52 | 0.0103 |
| Gemma3-12b | 79.36 | 79.72 | 73.05 | 90.33 | 86.63 | 82.16 | 85.13 | 92.27 | 0.0109 |
| Gemma3-27b | 79.82 | 79.82 | 73.87 | 90.63 | 86.54 | 82.85 | 86.43 | 92.12 | 0.0122 |
| GPT-oss-20b | 81.83 | 80.98 | 75.10 | 91.69 | 87.63 | 85.05 | 87.51 | 93.05 | 0.5087 |
| GPT-oss-120b | 83.00 | 81.71 | 75.88 | 92.31 | 88.17 | 84.63 | 88.11 | 93.38 | 0.4803 |
| Text-Embedding-3-Small | 80.04 | 78.22 | 72.26 | 90.82 | 86.28 | 81.97 | 66.90 | 92.49 | 0.0488 |
| Gemini-Embedding-001 | 79.78 | 78.90 | 72.01 | 91.16 | 87.08 | 82.59 | 85.13 | 92.97 | 0.0428 |

Table 9: Predictor performance on TriviaQA-constructed dataset evaluated across nine metrics using the strong generator model $f_G$ = GPT-4o. Dataset construction is described in Appendix A.1.

| $f_P$ | AUROC | Accuracy | Balanced Acc. | AUPRC | F1 Score | Sel. Balanced Acc. | Sel. AUROC | Sel. F1 | ECE |
|---|---|---|---|---|---|---|---|---|---|
| Llama3-70b-instruct | 81.28 | 83.63 | 73.68 | 94.20 | 90.07 | 81.54 | 86.41 | 95.06 | 0.6151 |
| Llama3-8b-instruct | 81.43 | 83.72 | 72.58 | 90.12 | 90.12 | 81.60 | 83.54 | 95.03 | 0.6276 |
| Llama3-8b | 80.06 | 82.30 | 71.41 | 93.47 | 89.22 | 79.44 | 77.87 | 94.60 | 0.6330 |
| Llama2-70b | 70.15 | 80.30 | 64.70 | 89.67 | 88.52 | 69.52 | 65.67 | 92.64 | 0.6059 |
| Llama2-7b | 61.68 | 81.57 | 56.42 | 85.75 | 89.85 | 58.36 | 54.74 | 90.85 | 0.7058 |
| Mistral-7b | 68.29 | 81.40 | 62.86 | 89.30 | 89.63 | 68.99 | 71.80 | 92.37 | 0.0013 |
| Mistral-22b | 67.53 | 80.97 | 63.35 | 89.21 | 89.21 | 69.71 | 73.54 | 92.25 | 0.0324 |
| Mistral-24b | 71.82 | 82.32 | 66.82 | 90.70 | 90.03 | 72.79 | 76.33 | 93.66 | 0.0094 |
| Gemma2-27b | 75.54 | 82.05 | 68.88 | 91.61 | 89.26 | 76.57 | 79.89 | 93.95 | 0.0386 |
| Gemma3-12b | 76.99 | 82.92 | 69.11 | 92.77 | 89.82 | 78.31 | 82.42 | 94.53 | 0.0210 |
| Gemma3-27b | 78.18 | 83.05 | 71.42 | 92.96 | 89.86 | 79.36 | 83.50 | 94.64 | 0.0347 |
| GPT-oss-20b | 79.69 | 84.01 | 71.49 | 93.01 | 94.06 | 79.63 | 81.24 | 94.85 | 0.6103 |
| GPT-oss-120b | 79.91 | 84.45 | 71.45 | 93.46 | 90.74 | 79.30 | 81.59 | 94.93 | 0.6037 |
| Text-Embedding-3-Small | 76.70 | 71.15 | 71.80 | 93.12 | 79.96 | 74.98 | 75.23 | 72.46 | 0.0138 |
| Gemini-Embedding-001 | 76.42 | 66.98 | 69.69 | 92.84 | 76.43 | 76.40 | 77.65 | 72.22 | 0.0035 |

Table 10: Predictor performance on CounterFact-constructed dataset evaluated across nine metrics using the strong generator model $f_G$ = GPT-3.5-turbo. Dataset construction is described in Appendix A.1.

| $f_P$ | AUROC | Accuracy | Balanced Acc. | AUPRC | F1 Score | Sel. Balanced Acc. | Sel. AUROC | Sel. F1 | ECE |
|---|---|---|---|---|---|---|---|---|---|
| Llama3-70b-instruct | 85.28 | 94.12 | 75.35 | 98.88 | 96.97 | 85.90 | 91.95 | 98.04 | 0.8008 |
| Llama3-8b-instruct | 83.82 | 93.98 | 73.90 | 98.66 | 96.89 | 83.35 | 79.76 | 97.77 | 0.8314 |
| Llama3-8b | 81.07 | 94.00 | 71.08 | 98.38 | 96.89 | 80.21 | 75.88 | 97.60 | 0.8314 |
| Llama2-70b | 73.60 | 94.15 | 67.09 | 97.63 | 96.97 | 77.17 | 82.08 | 97.20 | 0.7293 |
| Llama2-7b | 70.10 | 94.12 | 63.36 | 96.31 | 96.97 | 65.79 | 57.89 | 96.97 | 0.7277 |
| Mistral-7b | 74.88 | 94.08 | 67.97 | 97.46 | 96.95 | 71.75 | 65.30 | 97.31 | 0.0834 |
| Mistral-22b | 72.61 | 94.07 | 64.04 | 97.28 | 96.94 | 70.55 | 67.57 | 97.36 | 0.0885 |
| Mistral-24b | 75.97 | 94.07 | 69.09 | 97.68 | 96.94 | 74.53 | 69.90 | 97.30 | 0.0822 |
| Gemma2-27b | 78.59 | 94.07 | 68.82 | 98.06 | 96.94 | 77.25 | 73.85 | 97.66 | 0.0868 |
| Gemma3-12b | 80.09 | 94.07 | 66.56 | 98.30 | 96.94 | 78.33 | 82.57 | 97.72 | 0.0859 |
| Gemma3-27b | 77.70 | 94.07 | 64.48 | 97.87 | 96.94 | 72.87 | 65.58 | 97.69 | 0.0888 |
| GPT-oss-20b | 76.67 | 94.42 | 69.24 | 97.90 | 97.13 | 77.09 | 78.14 | 97.56 | 0.6441 |
| GPT-oss-120b | 79.30 | 94.42 | 70.44 | 98.23 | 97.13 | 79.83 | 80.62 | 97.80 | 0.5968 |
| Text-Embedding-3-Small | 73.65 | 94.45 | 63.58 | 97.35 | 97.13 | 69.22 | 58.24 | 97.46 | 0.8250 |
| Gemini-Embedding-001 | 77.34 | 94.41 | 65.22 | 98.06 | 97.13 | 75.45 | 74.18 | 97.75 | 0.8101 |

Table 11: Predictor performance on CounterFact-constructed dataset evaluated across nine metrics using the strong generator model $f_G$ = GPT-4o. Dataset construction is described in Appendix A.1.

| $f_P$ | AUROC | Accuracy | Balanced Acc. | AUPRC | F1 Score | Sel. Balanced Acc. | Sel. AUROC | Sel. F1 | ECE |
|---|---|---|---|---|---|---|---|---|---|
| Llama3-70b-instruct | 85.20 | 94.36 | 74.65 | 98.82 | 97.10 | 85.30 | 83.20 | 97.90 | 0.7601 |
| Llama3-8b-instruct | 82.84 | 94.14 | 75.95 | 98.51 | 96.98 | 82.86 | 81.23 | 97.60 | 0.6910 |
| Llama3-8b | 81.95 | 94.28 | 70.77 | 98.43 | 97.05 | 79.77 | 72.95 | 97.83 | 0.7962 |
| Llama2-70b | 69.95 | 94.27 | 63.49 | 97.26 | 97.05 | 72.14 | 75.18 | 97.26 | 0.7322 |
| Llama2-7b | 71.47 | 94.13 | 64.69 | 96.86 | 96.97 | 65.90 | 59.07 | 97.37 | 0.7075 |
| Mistral-7b | 73.58 | 94.38 | 65.70 | 97.56 | 97.11 | 71.06 | 63.52 | 97.53 | 0.0836 |
| Mistral-22b | 72.68 | 94.32 | 59.84 | 97.66 | 97.08 | 70.23 | 64.32 | 97.58 | 0.0884 |
| Mistral-24b | 73.25 | 94.40 | 67.31 | 97.56 | 97.12 | 71.80 | 65.01 | 97.30 | 0.0804 |
| Gemma2-27b | 79.86 | 94.33 | 67.17 | 98.22 | 97.08 | 77.73 | 74.16 | 97.86 | 0.0873 |
| Gemma3-12b | 77.15 | 94.32 | 66.00 | 97.94 | 97.08 | 75.88 | 73.96 | 97.65 | 0.0859 |
| Gemma3-27b | 76.68 | 94.32 | 65.98 | 97.97 | 97.08 | 74.38 | 75.71 | 97.61 | 0.0881 |
| GPT-oss-20b | 78.97 | 95.68 | 66.41 | 98.62 | 97.82 | 77.03 | 73.03 | 98.08 | 0.8080 |
| GPT-oss-120b | 80.74 | 95.68 | 71.44 | 98.82 | 97.78 | 80.37 | 77.89 | 98.19 | 0.7685 |
| Text-Embedding-3-Small | 63.27 | 95.80 | 62.15 | 96.91 | 97.85 | 59.34 | 53.45 | 97.33 | 0.9460 |
| Gemini-Embedding-001 | 68.47 | 95.68 | 62.35 | 97.66 | 97.79 | 69.59 | 67.13 | 97.59 | 0.9244 |

Table 12: Predictor performance on Winogrande-constructed dataset evaluated across nine metrics using the strong generator model $f_G$ = GPT-3.5-turbo. Dataset construction is described in Appendix A.1.

| $f_P$ | AUROC | Accuracy | Balanced Acc. | AUPRC | F1 Score | Sel. Balanced Acc. | Sel. AUROC | Sel. F1 | ECE |
|---|---|---|---|---|---|---|---|---|---|
| Llama3-70b-instruct | 68.62 | 82.22 | 63.48 | 91.00 | 90.22 | 75.08 | 71.19 | 93.20 | 0.3480 |
| Llama3-8b-instruct | 64.67 | 82.28 | 60.66 | 89.43 | 90.28 | 70.57 | 65.94 | 92.60 | 0.2985 |
| Llama3-8b | 61.68 | 82.33 | 59.47 | 87.80 | 90.31 | 62.51 | 65.36 | 91.99 | 0.2648 |
| Llama2-70b | 53.03 | 82.30 | 51.82 | 84.53 | 90.29 | 55.02 | 53.48 | 90.41 | 0.2827 |
| Llama2-7b | 51.75 | 82.34 | 51.27 | 82.84 | 90.31 | 52.30 | 51.55 | 89.39 | 0.0386 |
| Mistral-7b | 54.70 | 82.32 | 53.81 | 84.33 | 90.30 | 55.28 | 55.67 | 89.94 | 0.0165 |
| Mistral-22b | 55.25 | 82.22 | 55.19 | 84.61 | 90.24 | 56.34 | 56.20 | 90.18 | 0.0255 |
| Mistral-24b | 55.80 | 82.32 | 55.41 | 84.95 | 90.30 | 55.95 | 56.39 | 90.43 | 0.0114 |
| Gemma2-27b | 67.59 | 82.32 | 63.24 | 90.76 | 90.30 | 69.67 | 73.82 | 93.25 | 0.0150 |
| Gemma3-12b | 65.43 | 82.33 | 61.88 | 89.63 | 90.31 | 66.91 | 70.60 | 92.74 | 0.0152 |
| Gemma3-27b | 68.81 | 82.28 | 65.22 | 90.79 | 90.28 | 71.01 | 72.51 | 93.14 | 0.0123 |
| GPT-oss-20b | 60.02 | 72.68 | 57.66 | 79.36 | 84.18 | 60.40 | 61.09 | 85.76 | 0.0001 |
| GPT-oss-120b | 60.37 | 72.50 | 58.08 | 79.66 | 84.15 | 61.05 | 63.12 | 86.15 | 0.0001 |
| Text-Embedding-3-Small | 56.69 | 72.70 | 53.28 | 77.79 | 84.19 | 55.62 | 55.07 | 85.09 | 0.0000 |
| Gemini-Embedding-001 | 62.59 | 72.70 | 56.75 | 81.38 | 84.19 | 60.56 | 59.27 | 87.96 | 0.0000 |

Table 13: Predictor performance on Winogrande-constructed dataset evaluated across nine metrics using the strong generator model $f_G$ = GPT-4o. Dataset construction is described in Appendix A.1.

| $f_P$ | AUROC | Accuracy | Balanced Acc. | AUPRC | F1 Score | Sel. Balanced Acc. | Sel. AUROC | Sel. F1 | ECE |
|---|---|---|---|---|---|---|---|---|---|
| Llama3-70b-instruct | 65.46 | 92.38 | 60.75 | 95.42 | 96.04 | 68.39 | 66.44 | 96.32 | 0.7520 |
| Llama3-8b-instruct | 61.49 | 92.38 | 58.14 | 94.63 | 96.04 | 64.13 | 61.57 | 96.17 | 0.2740 |
| Llama3-8b | 59.56 | 92.45 | 58.35 | 94.45 | 96.08 | 59.48 | 61.37 | 96.11 | 0.7328 |
| Llama2-70b | 52.71 | 92.39 | 51.97 | 93.00 | 91.04 | 52.58 | 51.51 | 95.27 | 0.3152 |
| Llama2-7b | 52.39 | 92.38 | 51.78 | 92.76 | 96.03 | 51.29 | 51.40 | 95.14 | 0.3020 |
| Mistral-7b | 52.45 | 92.42 | 53.19 | 93.00 | 96.06 | 52.73 | 52.11 | 95.31 | 0.0744 |
| Mistral-22b | 56.19 | 92.42 | 55.18 | 93.75 | 96.06 | 54.97 | 53.44 | 95.72 | 0.0822 |
| Mistral-24b | 54.95 | 92.43 | 54.62 | 93.37 | 96.07 | 56.31 | 52.46 | 95.38 | 0.0777 |
| Gemma2-27b | 69.51 | 92.42 | 64.40 | 96.40 | 96.06 | 71.84 | 74.63 | 96.97 | 0.0685 |
| Gemma3-12b | 66.11 | 92.42 | 62.09 | 95.64 | 96.06 | 65.76 | 66.57 | 96.60 | 0.0791 |
| Gemma3-27b | 69.72 | 92.42 | 66.59 | 96.30 | 96.06 | 71.97 | 68.61 | 96.88 | 0.0773 |
| GPT-oss-20b | 60.93 | 90.00 | 57.92 | 92.95 | 94.75 | 62.30 | 63.04 | 94.89 | 0.0254 |
| GPT-oss-120b | 62.40 | 90.00 | 60.41 | 93.20 | 94.74 | 63.67 | 66.28 | 94.93 | 0.0288 |
| Text-Embedding-3-Small | 58.27 | 90.00 | 54.49 | 82.08 | 94.75 | 59.70 | 87.08 | 94.91 | 0.0028 |
| Gemini-Embedding-001 | 65.06 | 90.00 | 60.33 | 94.17 | 94.74 | 67.72 | 69.31 | 95.91 | 0.0026 |

Table 14: Predictor performance on MMLU-constructed dataset evaluated across nine metrics using the strong generator model $f_G$ = GPT-3.5-turbo. Dataset construction is described in Appendix A.1.

| $f_P$ | AUROC | Accuracy | Balanced Acc. | AUPRC | F1 Score | Sel. Balanced Acc. | Sel. AUROC | Sel. F1 | ECE |
|---|---|---|---|---|---|---|---|---|---|
| Llama3-70b-instruct | 68.91 | 66.45 | 63.80 | 82.88 | 75.23 | 70.48 | 73.90 | 81.64 | 0.4818 |
| Llama3-8b-instruct | 64.26 | 58.08 | 60.10 | 78.00 | 64.57 | 65.00 | 66.71 | 66.35 | 0.4043 |
| Llama3-8b | 63.77 | 62.17 | 59.46 | 78.82 | 71.11 | 65.41 | 68.21 | 74.37 | 0.3938 |
| Llama2-70b | 57.01 | 59.30 | 55.58 | 73.78 | 69.44 | 58.69 | 60.44 | 74.67 | 0.4237 |
| Llama2-7b | 58.39 | 55.57 | 55.17 | 74.78 | 64.10 | 61.22 | 63.45 | 58.44 | 0.3874 |
| Mistral-7b | 56.28 | 69.65 | 54.63 | 73.12 | 81.61 | 57.23 | 57.45 | 83.28 | 0.0260 |
| Mistral-22b | 58.14 | 70.02 | 57.00 | 73.91 | 81.69 | 59.55 | 59.31 | 83.77 | 0.0343 |
| Mistral-24b | 57.11 | 69.72 | 55.68 | 74.13 | 81.54 | 57.52 | 59.83 | 83.82 | 0.0169 |
| Gemma2-27b | 62.52 | 69.60 | 59.07 | 78.74 | 80.82 | 64.57 | 66.48 | 86.14 | 0.0261 |
| Gemma3-12b | 62.77 | 70.28 | 58.80 | 77.89 | 81.59 | 64.36 | 66.36 | 85.59 | 0.0216 |
| Gemma3-27b | 62.49 | 69.70 | 59.40 | 78.14 | 81.15 | 64.38 | 66.69 | 85.63 | 0.0302 |
| GPT-oss-20b | 65.49 | 71.43 | 60.68 | 80.51 | 81.85 | 67.76 | 70.83 | 86.65 | 0.0093 |
| GPT-oss-120b | 66.67 | 70.97 | 62.90 | 80.62 | 82.08 | 68.58 | 70.40 | 86.69 | 0.0018 |
| Text-Embedding-3-Small | 63.94 | 59.57 | 59.07 | 79.46 | 67.31 | 66.55 | 66.90 | 63.80 | 0.0041 |
| Gemini-Embedding-001 | 64.28 | 60.12 | 60.48 | 80.55 | 67.87 | 66.07 | 68.61 | 64.22 | 0.0023 |

Table 15: Predictor performance on MMLU-constructed dataset evaluated across nine metrics using the strong generator model $f_G$ = GPT-4o. Dataset construction is described in Appendix A.1.

| $f_P$ | AUROC | Accuracy | Balanced Acc. | AUPRC | F1 Score | Sel. Balanced Acc. | Sel. AUROC | Sel. F1 | ECE |
|---|---|---|---|---|---|---|---|---|---|
| Llama3-70b-instruct | 65.86 | 83.85 | 59.45 | 91.13 | 90.97 | 63.63 | 63.60 | 93.88 | 0.8000 |
| Llama3-8b-instruct | 62.46 | 80.82 | 57.21 | 90.42 | 89.06 | 61.86 | 61.44 | 92.93 | 0.7750 |
| Llama3-8b | 62.84 | 83.47 | 59.04 | 90.07 | 90.75 | 61.05 | 60.83 | 93.34 | 0.7635 |
| Llama2-70b | 53.99 | 81.38 | 54.29 | 86.97 | 89.52 | 52.57 | 53.17 | 91.57 | 0.7790 |
| Llama2-7b | 53.30 | 85.72 | 54.46 | 86.81 | 92.31 | 52.49 | 50.87 | 91.53 | 0.7768 |
| Mistral-7b | 57.14 | 85.87 | 56.16 | 88.34 | 92.39 | 58.05 | 58.88 | 92.32 | 0.0798 |
| Mistral-22b | 57.79 | 85.93 | 56.57 | 88.19 | 92.43 | 56.51 | 55.16 | 92.70 | 0.0842 |
| Mistral-24b | 57.78 | 86.00 | 56.87 | 88.86 | 92.47 | 58.53 | 61.51 | 92.73 | 0.0679 |
| Gemma2-27b | 58.91 | 85.82 | 55.51 | 89.61 | 92.37 | 59.37 | 61.80 | 92.82 | 0.0786 |
| Gemma3-12b | 55.78 | 85.97 | 55.98 | 87.91 | 92.45 | 58.08 | 60.13 | 92.16 | 0.0808 |
| Gemma3-27b | 57.58 | 85.87 | 56.16 | 88.62 | 92.39 | 60.30 | 60.95 | 92.46 | 0.0840 |
| GPT-oss-20b | 64.77 | 80.93 | 60.55 | 91.02 | 92.43 | 66.81 | 68.69 | 93.70 | 0.3268 |
| GPT-oss-120b | 63.93 | 85.97 | 60.24 | 90.72 | 92.45 | 66.44 | 68.67 | 93.32 | 0.3031 |
| Text-Embedding-3-Small | 56.53 | 85.93 | 58.53 | 88.55 | 92.43 | 61.32 | 63.33 | 92.78 | 0.0044 |
| Gemini-Embedding-001 | 59.53 | 85.93 | 57.04 | 90.10 | 92.43 | 64.39 | 68.02 | 93.52 | 0.9244 |

Table 16: Predictor performance on MedQA-constructed dataset evaluated across nine metrics using the strong generator model $f_G$ = GPT-3.5-turbo. Dataset construction is described in Appendix A.1.

| $f_P$ | AUROC | Accuracy | Balanced Acc. | AUPRC | F1 Score | Sel. Balanced Acc. | Sel. AUROC | Sel. F1 | ECE |
|---|---|---|---|---|---|---|---|---|---|
| Llama3-70b-instruct | 65.40 | 62.36 | 61.37 | 73.25 | 72.04 | 65.91 | 66.86 | 79.84 | 0.0006 |
| Llama3-8b-instruct | 59.74 | 59.17 | 57.78 | 68.91 | 68.22 | 61.14 | 62.76 | 74.83 | 0.0002 |
| Llama3-8b | 59.96 | 59.09 | 58.07 | 69.04 | 67.90 | 61.33 | 62.76 | 74.74 | 0.0001 |
| Llama2-70b | 56.01 | 59.59 | 54.33 | 65.49 | 73.46 | 56.18 | 56.13 | 76.45 | 0.0001 |
| Llama2-7b | 53.66 | 60.09 | 53.51 | 63.02 | 74.67 | 53.99 | 54.26 | 74.84 | 0.0056 |
| Mistral-7b | 54.13 | 57.57 | 53.33 | 64.24 | 69.44 | 54.55 | 55.05 | 74.46 | 0.0378 |
| Mistral-22b | 56.62 | 59.37 | 56.45 | 65.18 | 70.77 | 57.65 | 57.82 | 74.26 | 0.0008 |
| Mistral-24b | 55.84 | 59.33 | 55.29 | 64.91 | 72.02 | 56.92 | 58.03 | 73.73 | 0.0001 |
| Gemma2-27b | 58.57 | 61.42 | 56.41 | 67.58 | 73.70 | 59.89 | 61.78 | 77.68 | 0.0003 |
| Gemma3-12b | 59.29 | 60.60 | 56.98 | 68.05 | 72.82 | 60.02 | 62.67 | 77.58 | 0.0002 |
| Gemma3-27b | 59.28 | 60.77 | 56.74 | 67.79 | 72.22 | 59.82 | 61.11 | 77.60 | 0.0003 |
| GPT-oss-20b | 61.58 | 60.23 | 58.69 | 70.13 | 72.19 | 62.89 | 66.12 | 77.37 | 0.0001 |
| GPT-oss-120b | 61.67 | 59.77 | 58.76 | 70.40 | 71.52 | 62.37 | 64.93 | 76.33 | 0.0001 |
| Text-Embedding-3-Small | 59.06 | 43.03 | 52.14 | 68.20 | 12.76 | 56.19 | 54.73 | 1.64 | 0.0001 |
| Gemini-Embedding-001 | 61.20 | 41.20 | 50.77 | 69.42 | 5.11 | 58.20 | 55.38 | 0.33 | 0.0000 |

Table 17: Predictor performance on MedQA-constructed dataset evaluated across nine metrics using the strong generator model $f_G$ = GPT-4o. Dataset construction is described in Appendix A.1.

| $f_P$ | AUROC | Accuracy | Balanced Acc. | AUPRC | F1 Score | Sel. Balanced Acc. | Sel. AUROC | Sel. F1 | ECE |
|---|---|---|---|---|---|---|---|---|---|
| Llama3-70b-instruct | 66.15 | 88.10 | 61.54 | 93.08 | 93.67 | 68.66 | 70.79 | 94.89 | 0.0703 |
| Llama3-8b-instruct | 62.87 | 88.10 | 58.70 | 92.01 | 93.67 | 65.37 | 67.89 | 94.64 | 0.0412 |
| Llama3-8b | 62.06 | 88.21 | 57.35 | 91.71 | 93.68 | 63.89 | 67.15 | 94.47 | 0.0343 |
| Llama2-70b | 56.90 | 88.10 | 55.12 | 90.14 | 93.67 | 56.94 | 58.09 | 93.31 | 0.0607 |
| Llama2-7b | 57.82 | 88.10 | 54.49 | 90.39 | 93.67 | 58.45 | 59.37 | 93.67 | 0.0677 |
| Mistral-7b | 53.99 | 88.05 | 51.26 | 89.56 | 93.65 | 53.76 | 54.64 | 93.46 | 0.0960 |
| Mistral-22b | 58.03 | 88.08 | 55.10 | 90.52 | 93.66 | 56.46 | 56.90 | 93.96 | 0.0860 |
| Mistral-24b | 56.14 | 88.10 | 56.49 | 89.78 | 93.67 | 55.96 | 56.57 | 93.57 | 0.0711 |
| Gemma2-27b | 61.01 | 88.10 | 58.72 | 91.06 | 93.67 | 59.73 | 62.51 | 94.09 | 0.0650 |
| Gemma3-12b | 59.92 | 88.10 | 58.10 | 90.95 | 93.67 | 58.56 | 60.03 | 94.13 | 0.0678 |
| Gemma3-27b | 61.35 | 88.25 | 59.07 | 90.95 | 93.74 | 59.57 | 60.04 | 94.03 | 0.0750 |
| GPT-oss-20b | 63.55 | 88.38 | 58.61 | 91.92 | 93.81 | 64.41 | 67.70 | 94.52 | 0.0188 |
| GPT-oss-120b | 63.74 | 88.30 | 59.83 | 92.59 | 93.79 | 64.51 | 68.62 | 94.78 | 0.0479 |
| Text-Embedding-3-Small | 61.45 | 88.35 | 59.04 | 91.56 | 93.79 | 63.38 | 64.34 | 94.34 | 0.0034 |
| Gemini-Embedding-001 | 62.52 | 88.30 | 57.26 | 91.95 | 93.79 | 63.74 | 65.09 | 94.67 | 0.0033 |

Table 18: Predictor performance on SuperGPQA-constructed dataset evaluated across nine metrics using the strong generator model $f_G$ = GPT-3.5-turbo. Dataset construction is described in Appendix A.1.

| $f_P$ | AUROC | Accuracy | Balanced Acc. | AUPRC | F1 Score | Sel. Balanced Acc. | Sel. AUROC | Sel. F1 | ECE |
|---|---|---|---|---|---|---|---|---|---|
| Llama3-70b-instruct | 52.38 | 79.33 | 52.53 | 21.44 | 34.32 | 52.00 | 52.45 | 34.53 | 0.0000 |
| Llama3-8b-instruct | 52.67 | 79.37 | 51.94 | 23.05 | 34.37 | 51.08 | 51.76 | 34.18 | 0.0007 |
| Llama3-8b | 52.33 | 79.42 | 50.85 | 22.56 | 34.45 | 51.62 | 52.76 | 34.74 | 0.0017 |
| Llama2-70b | 53.51 | 79.60 | 51.85 | 23.24 | 14.23 | 52.37 | 52.25 | 3.81 | 0.0000 |
| Llama2-7b | 51.77 | 79.33 | 51.68 | 21.67 | 34.31 | 51.11 | 51.31 | 34.87 | 0.0000 |
| Mistral-7b | 50.54 | 79.32 | 50.52 | 21.60 | 34.28 | 50.64 | 50.13 | 33.74 | 0.0000 |
| Mistral-22b | 51.72 | 79.37 | 51.45 | 22.06 | 34.24 | 52.15 | 51.41 | 34.59 | 0.0003 |
| Mistral-24b | 54.09 | 79.35 | 50.40 | 23.21 | 34.31 | 52.88 | 53.61 | 34.15 | 0.0000 |
| Gemma2-27b | 53.12 | 79.33 | 51.33 | 23.21 | 34.33 | 52.50 | 51.53 | 34.02 | 0.0002 |
| Gemma3-12b | 52.36 | 79.33 | 54.03 | 22.99 | 34.29 | 52.66 | 53.63 | 35.77 | 0.0000 |
| Gemma3-27b | 53.16 | 79.35 | 51.09 | 23.24 | 34.35 | 50.95 | 51.76 | 33.80 | 0.0000 |
| GPT-oss-20b | 63.30 | 69.35 | 57.77 | 46.76 | 38.60 | 59.64 | 59.10 | 36.76 | 0.0003 |
| GPT-oss-120b | 63.95 | 69.33 | 58.16 | 47.01 | 37.66 | 60.85 | 59.54 | 35.90 | 0.0003 |
| Text-Embedding-3-Small | 63.70 | 68.13 | 55.79 | 44.95 | 30.61 | 59.42 | 54.62 | 8.01 | 0.0000 |
| Gemini-Embedding-001 | 63.96 | 68.85 | 56.77 | 46.49 | 32.25 | 59.35 | 55.06 | 8.12 | 0.0000 |

Table 19: Predictor performance on SuperGPQA-constructed dataset evaluated across nine metrics using the strong generator model $f_G$ = GPT-4o. Dataset construction is described in Appendix A.1.

| $f_P$ | AUROC | Accuracy | Balanced Acc. | AUPRC | F1 Score | Sel. Balanced Acc. | Sel. AUROC | Sel. F1 | ECE |
|---|---|---|---|---|---|---|---|---|---|
| Llama3-70b-instruct | 64.85 | 63.42 | 61.31 | 55.17 | 59.50 | 65.35 | 66.17 | 67.75 | 0.0017 |
| Llama3-8b-instruct | 59.70 | 60.15 | 58.03 | 50.17 | 59.00 | 60.09 | 60.16 | 63.57 | 0.0011 |
| Llama3-8b | 59.90 | 60.30 | 57.57 | 50.45 | 59.20 | 60.27 | 59.95 | 63.52 | 0.0018 |
| Llama2-70b | 62.43 | 62.50 | 60.06 | 52.38 | 50.73 | 63.11 | 64.54 | 54.66 | 0.0071 |
| Llama2-7b | 57.27 | 59.10 | 56.77 | 47.94 | 58.81 | 57.56 | 57.58 | 63.05 | 0.0014 |
| Mistral-7b | 57.24 | 59.48 | 55.63 | 48.28 | 46.36 | 57.70 | 58.80 | 49.61 | 0.0005 |
| Mistral-22b | 57.46 | 58.78 | 55.77 | 47.43 | 58.67 | 57.04 | 57.20 | 62.89 | 0.0096 |
| Mistral-24b | 57.88 | 58.53 | 56.27 | 48.04 | 58.81 | 57.51 | 57.63 | 63.43 | 0.0024 |
| Gemma2-27b | 61.45 | 62.70 | 59.70 | 53.94 | 58.85 | 62.87 | 62.73 | 65.60 | 0.0091 |
| Gemma3-12b | 61.85 | 62.42 | 59.22 | 53.71 | 58.80 | 63.28 | 64.00 | 65.77 | 0.0090 |
| Gemma3-27b | 59.54 | 60.15 | 57.05 | 50.46 | 59.22 | 60.30 | 61.19 | 63.63 | 0.0066 |
| GPT-oss-20b | 66.99 | 64.30 | 63.97 | 61.83 | 60.16 | 68.06 | 69.01 | 67.28 | 0.0005 |
| GPT-oss-120b | 67.02 | 64.37 | 63.75 | 61.62 | 59.50 | 68.39 | 69.40 | 67.66 | 0.0004 |
| Text-Embedding-3-Small | 65.65 | 64.00 | 63.38 | 60.25 | 61.47 | 67.38 | 65.86 | 70.88 | 0.0000 |
| Gemini-Embedding-001 | 66.93 | 64.53 | 63.80 | 61.76 | 62.05 | 68.32 | 69.35 | 71.00 | 0.0000 |

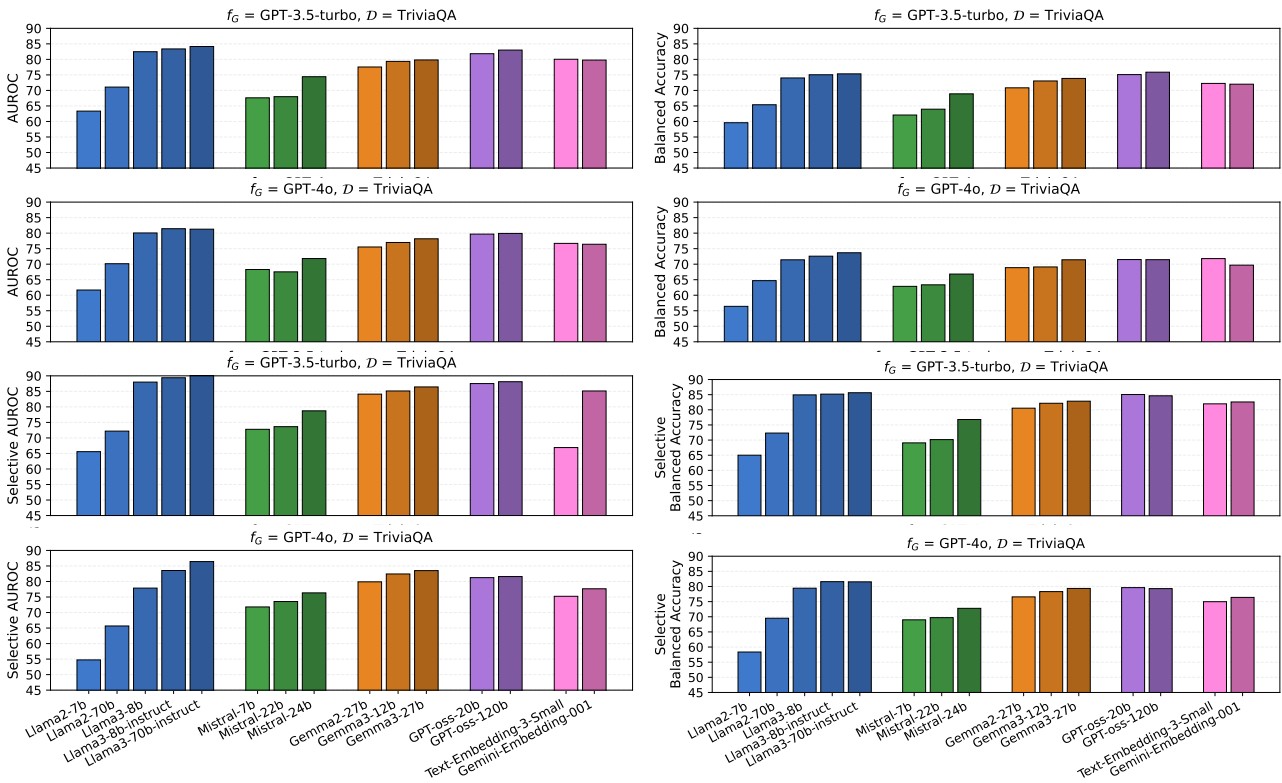

Figure 13: Visualization of predictor performance on TriviaQA in terms of AUROC, balanced accuracy, selective AUROC, and selective balanced accuracy across fifteen backbones. For Llama models (blue), increasing the size of the predictor backbone leads to a significant improvement in all metrics. However, this trend does not generally hold for Mistral or Gemma models, and no consistent scaling pattern is observed across model families.

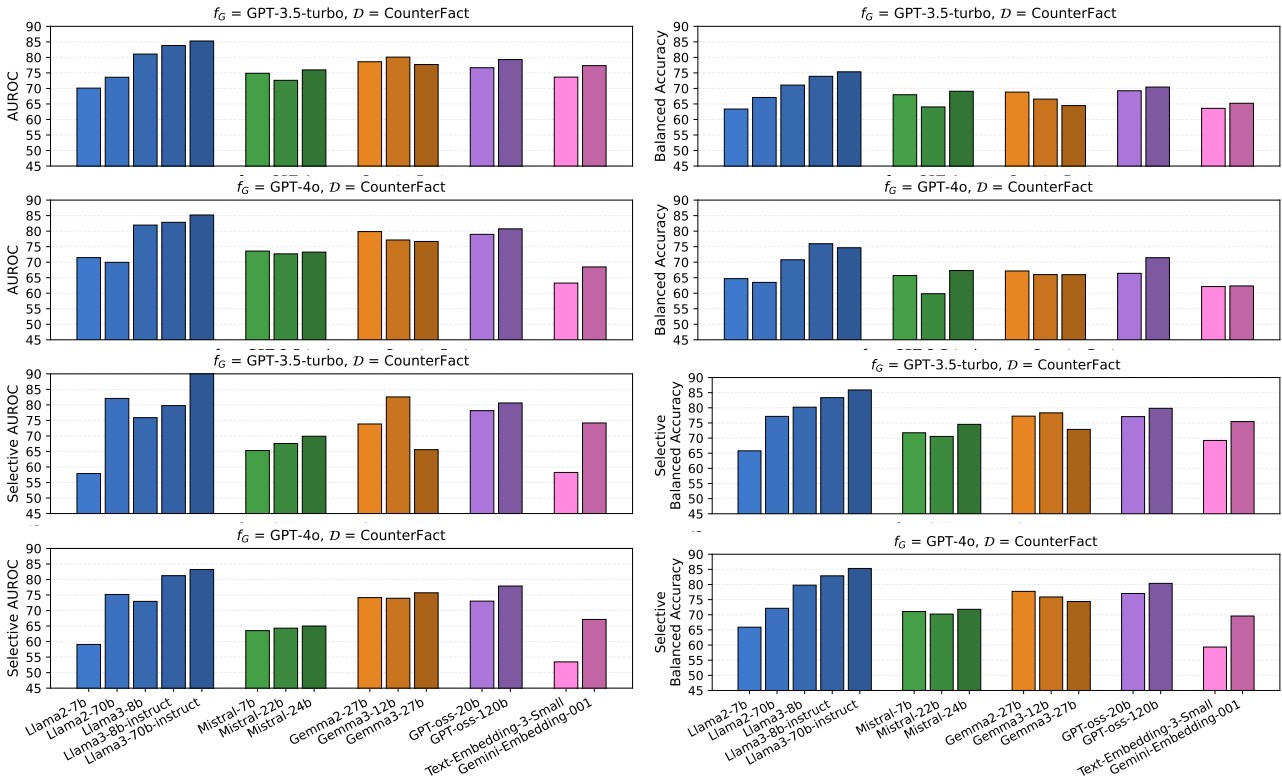

Figure 14: Visualization of predictor performance on CounterFact in terms of AUROC, balanced accuracy, selective AUROC, and selective balanced accuracy across fifteen backbones. For Llama models (blue), increasing the size of the predictor backbone leads to a significant improvement in all metrics. However, this trend does not generally hold for Mistral or Gemma models, and no consistent scaling pattern is observed across model families.

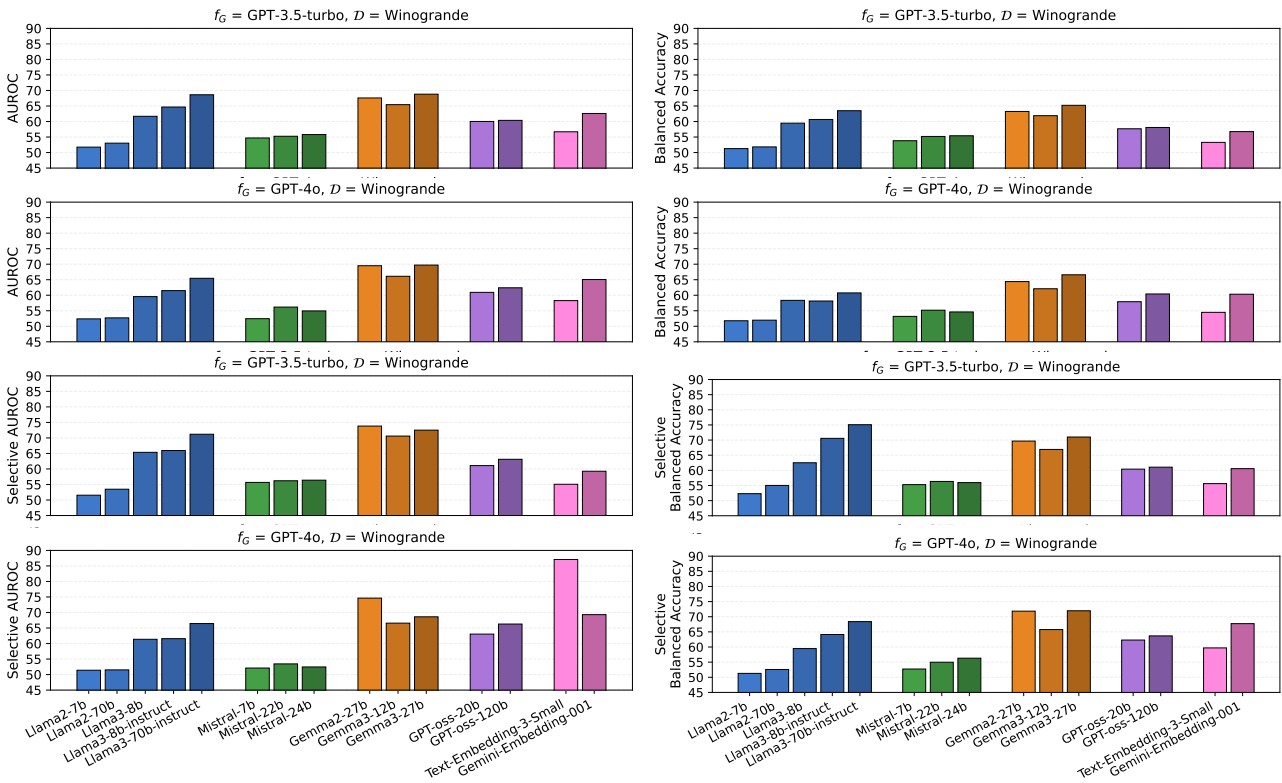

Figure 15: Visualization of predictor performance on Winogrande in terms of AUROC, balanced accuracy, selective AUROC, and selective balanced accuracy across fifteen backbones. For Llama models (blue), increasing the size of the predictor backbone leads to a significant improvement in all metrics. However, this trend does not generally hold for Mistral or Gemma models, and no consistent scaling pattern is observed across model families.

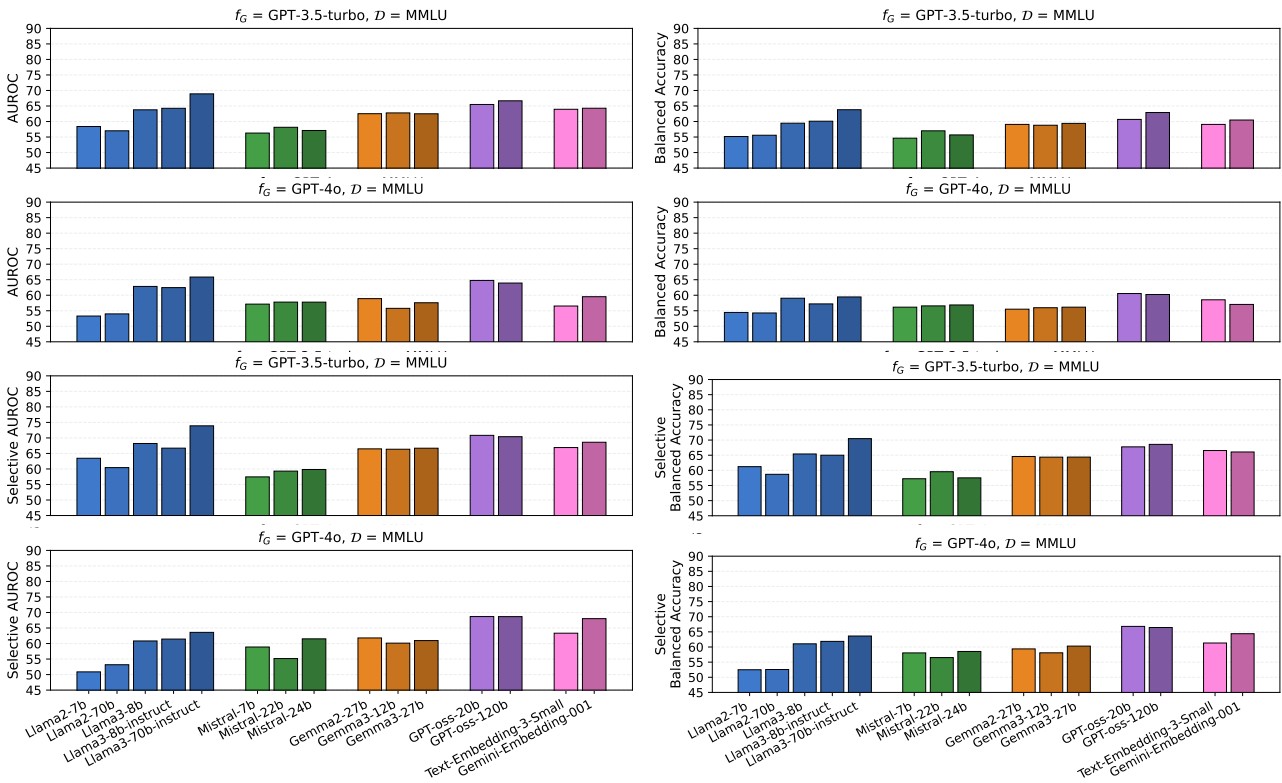

Figure 16: Visualization of predictor performance on MMLU in terms of AUROC, balanced accuracy, selective AUROC, and selective balanced accuracy across fifteen backbones. For Llama models (blue), increasing the size of the predictor backbone leads to a significant improvement in all metrics. However, this trend does not generally hold for Mistral or Gemma models, and no consistent scaling pattern is observed across model families.

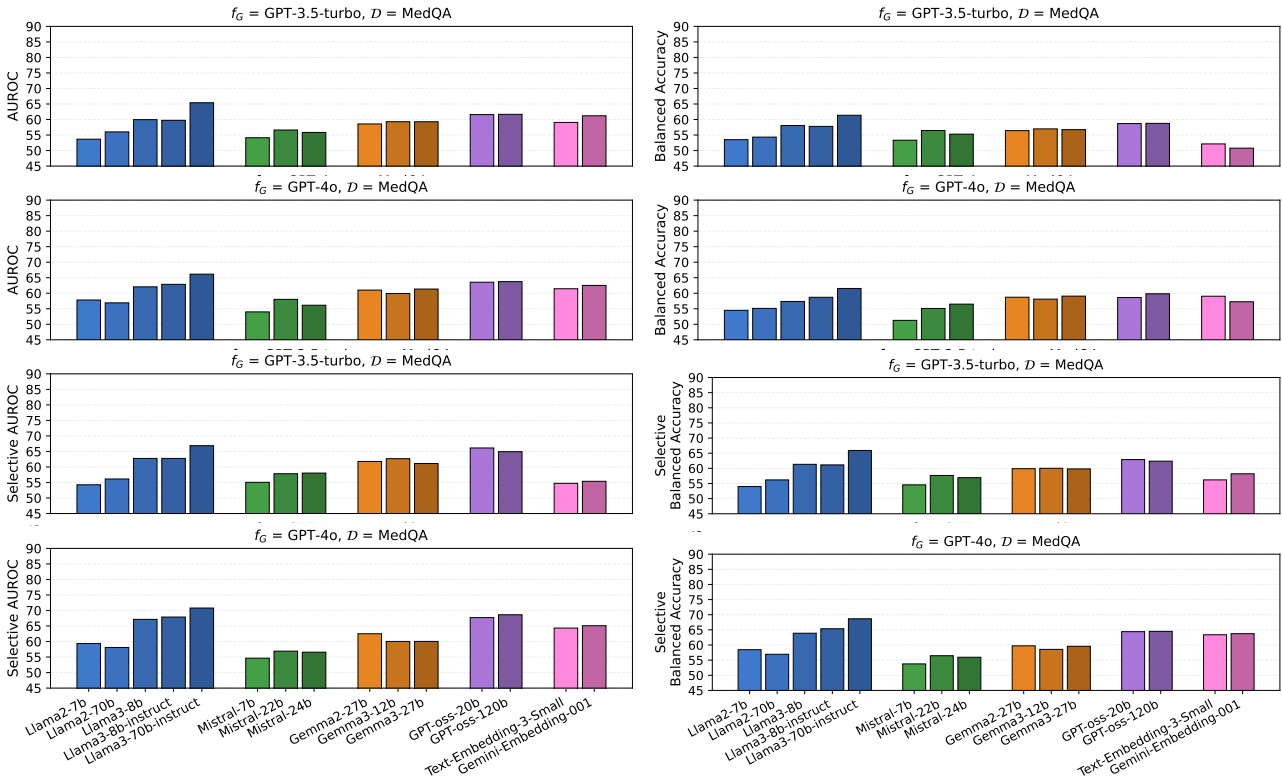

Figure 17: Visualization of predictor performance on MedQA in terms of AUROC, balanced accuracy, selective AUROC, and selective balanced accuracy across fifteen backbones. For Llama models (blue), increasing the size of the predictor backbone leads to a significant improvement in all metrics. However, this trend does not generally hold for Mistral or Gemma models, and no consistent scaling pattern is observed across model families.

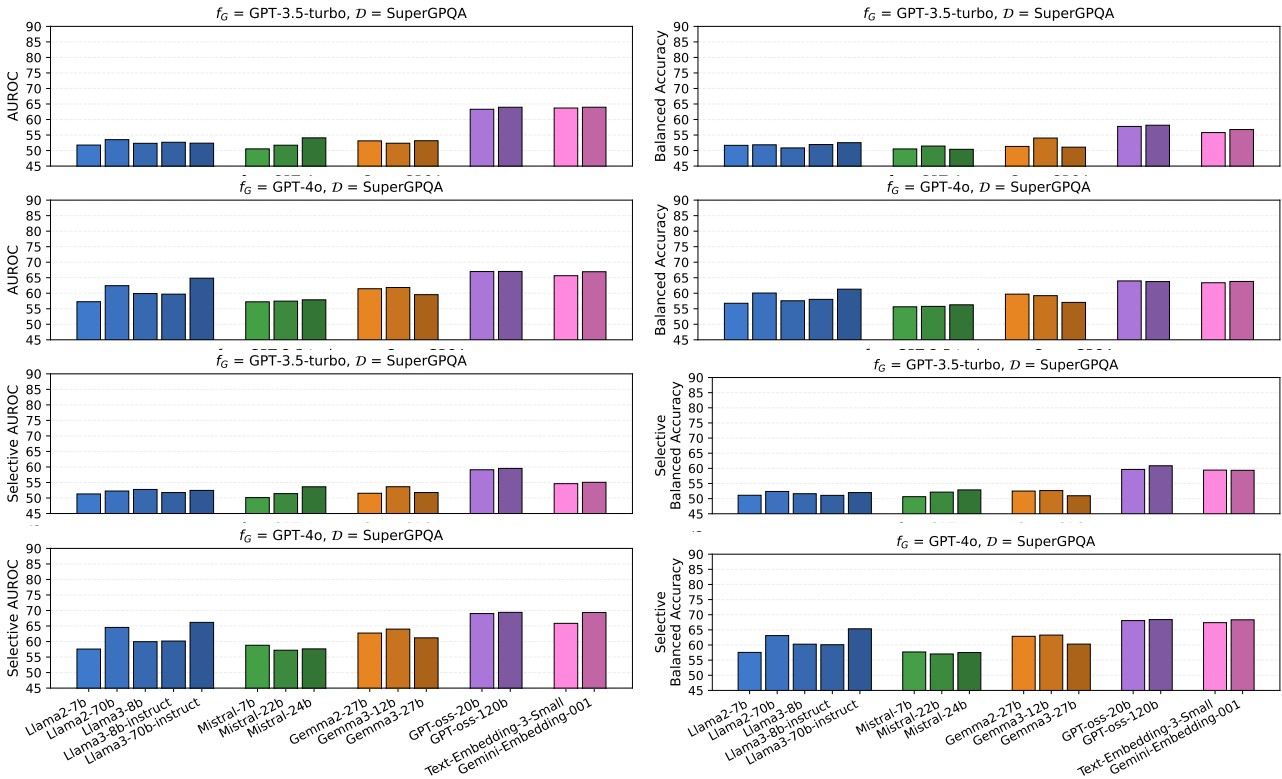

Figure 18: Visualization of predictor performance on SuperGPQA in terms of AUROC, balanced accuracy, selective AUROC, and selective balanced accuracy across fifteen backbones. For Llama models (blue), increasing the size of the predictor backbone leads to a significant improvement in all metrics. However, this trend does not generally hold for Mistral or Gemma models, and no consistent scaling pattern is observed across model families.

## D    Disagreement Entropy Analysis of The Probe Performance Gap on Reasoning Tasks

In the previous sections, probes tend to perform worse on reasoning intensive benchmarks such as MMLU, MedQA, and SuperGPQA. We believe there are several plausible reasons for this. Question-only embeddings may capture topical content and coarse difficulty, but they do not condition on latent intermediate steps that determine correctness in multi step reasoning. In addition, there may be multiple failure modes in the latent reasoning path, increasing difficulty in separability for probes. Finally, latent reasoning chains may lead to more stochastic generations, producing noisier generator answers.

We investigated the stochasticity hypothesis and quantified the effect using empirical answer disagreement entropy computed from 10 sampled generations per question. For each question, we group 10 sampled answers into an empirical distribution over unique answers $\{a_1, ..., a_k\}$ with probabilities $p(a_k) = count(a_k)/k$, where $k = 10$. We compute the disagreement entropy as $H = -\sum_{a \in \mathcal{A}} p(a) \log p(a)$, where higher entropy means more disagreement.

We assessed stochasticity on the five multiple choice datasets, excluding TriviaQA whose free response format makes it difficult for answer grouping. As shown in the table below, among the remaining five datasets, we observe that reasoning-heavy datasets tend to show higher disagreement entropy, indicating higher answer level variability under sampling.

Table 20: Disagreement entropy across six datasets for two generators.

| Dataset | Generator=GPT-3.5-turbo | Generator=GPT-4o |
|---|---|---|
| CounterFact | 9.73 | 9.25 |
| Winogrande | 25.72 | 42.05 |
| MMLU | 28.21 | 44.55 |
| MedQA | 48.59 | 50.31 |
| SuperGPQA | 87.75 | 71.56 |

## E    Full Predictor Performance with Controlling Exact Label Ratios

We provide full experimental results for our analysis in Section 4.3 as follows. In all experiments, the *Original* training and test sets contain 8,000 and 2,000 data points, respectively. For the TriviaQA-based dataset, the *Exact ratio* training set includes 2162 class 0 and 5297 class 1 samples, while the *Exact ratio* test set includes 540 class 0 samples and 1324 class 1 samples. For the MMLU-based dataset, the *Exact ratio* training set contains 1019 class 0 and 5705 class 1 samples, and the *Exact ratio* test set contains 254 class 0 and 1426 class 1 samples.

Table 21: Evaluation AUROC on TriviaQA *Exact ratio* training set. Size of training set = {class 0 = 2162, class 1 = 5297} for both generator models. For $\mathcal{D}_{\text{test}}$ = *Original*, the test size remains 2000; for $\mathcal{D}_{\text{test}}$ = *50/50*, size of test set = {class 0 = 540, class 1 = 540}; for $\mathcal{D}_{\text{test}}$ = *Exact ratio*, size of test set = {class 0 = 540, class 1 = 1324}.

| $f_G$ | GPT-3.5-turbo | | | GPT-4o | | |
|---|---|---|---|---|---|---|
| $\mathcal{D}_{\text{test}}$ | *Original* | *50/50* | *Exact ratio* | *Original* | *50/50* | *Exact ratio* |
| Llama2-7b | 56.90 | 58.13 | 56.64 | 60.60 | 61.70 | 61.05 |
| Llama2-70b | 65.87 | 67.34 | 65.90 | 69.28 | 70.33 | 69.78 |
| Llama3-8b | 80.37 | 81.78 | 80.19 | 79.68 | 80.41 | 79.67 |
| Llama3-8b-instruct | 81.06 | 81.69 | 80.86 | 81.12 | 81.26 | 80.98 |
| Llama3-70b-instruct | 80.89 | 81.64 | 80.61 | 80.79 | 81.70 | 80.68 |

Table 22: Evaluation AUROC on TriviaQA *Original* training set. Test sets remain the same as the table above.

| $f_G$ $\mathcal{D}_{\text{test}}$ | GPT-3.5-turbo | | | GPT-4o | | |
|---|---|---|---|---|---|---|
| | *Original* | *50/50* | *Exact ratio* | *Original* | *50/50* | *Exact ratio* |
| Llama2-7b | 63.34 | 60.77 | 61.37 | 61.68 | 61.06 | 60.69 |
| Llama2-70b | 71.08 | 70.37 | 71.37 | 70.15 | 70.75 | 70.71 |
| Llama3-8b | 82.47 | 82.24 | 82.86 | 80.06 | 80.48 | 80.21 |
| Llama3-8b-instruct | 83.37 | 83.17 | 81.43 | 81.42 | 81.86 | 81.51 |
| Llama3-70b-instruct | 84.15 | 83.75 | 81.28 | 81.36 | 81.15 | 81.15 |

Table 23: Evaluation AUROC on MMLU *Exact ratio* training set. Size of training set = {class 0 = 1019, class 1 = 5705} for both generator models. For $\mathcal{D}_{\text{test}}$ = *Original*, the test size remains 2000; for $\mathcal{D}_{\text{test}}$ = *50/50*, size of test set = {class 0 = 254, class 1 = 254}; for $\mathcal{D}_{\text{test}}$ = *Exact ratio*, size of test set = {class 0 = 254, class 1 = 1426}.

| $f_G$ $\mathcal{D}_{\text{test}}$ | GPT-3.5-turbo | | | GPT-4o | | |
|---|---|---|---|---|---|---|
| | *Original* | *50/50* | *Exact ratio* | *Original* | *50/50* | *Exact ratio* |
| Llama2-7b | 58.10 | 58.38 | 59.63 | 53.10 | 52.18 | 54.01 |
| Llama2-70b | 56.83 | 57.44 | 56.47 | 53.50 | 53.72 | 53.91 |
| Llama3-8b | 63.53 | 64.64 | 64.23 | 61.84 | 60.77 | 61.49 |
| Llama3-8b-instruct | 63.93 | 65.13 | 64.46 | 61.58 | 60.97 | 61.33 |
| Llama3-70b-instruct | 68.12 | 69.00 | 68.79 | 65.09 | 64.22 | 65.64 |

Table 24: Evaluation AUROC on MMLU *Original* training set. Test sets remain the same as the table above.

| $f_G$ $\mathcal{D}_{\text{test}}$ | GPT-3.5-turbo | | | GPT-4o | | |
|---|---|---|---|---|---|---|
| | *Original* | *50/50* | *Exact ratio* | *Original* | *50/50* | *Exact ratio* |
| Llama2-7b | 58.39 | 58.30 | 59.91 | 53.30 | 53.63 | 53.32 |
| Llama2-70b | 57.01 | 57.31 | 58.14 | 53.99 | 55.09 | 53.62 |
| Llama3-8b | 63.77 | 64.40 | 65.76 | 62.84 | 62.29 | 63.13 |
| Llama3-8b-instruct | 64.26 | 64.73 | 66.61 | 62.46 | 63.18 | 63.14 |
| Llama3-70b-instruct | 68.91 | 68.70 | 70.50 | 65.86 | 65.33 | 65.84 |

Table 25: Evaluation accuracy on TriviaQA *Exact ratio* training set. Size of training set = {class 0 = 2162, class 1 = 5297} for both generator models. For $\mathcal{D}_{\text{test}}$ = *Original*, the test size remains 2000; for $\mathcal{D}_{\text{test}}$ = *50/50*, size of test set = {class 0 = 540, class 1 = 540}; for $\mathcal{D}_{\text{test}}$ = *Exact ratio*, size of test set = {class 0 = 540, class 1 = 1324}. Size of training set = {class 0 = 2162, class 1 = 5297}.

| $f_G$ $\mathcal{D}_{\text{test}}$ | GPT-3.5-turbo | | | GPT-4o | | |
|---|---|---|---|---|---|---|
| | *Original* | *50/50* | *Exact ratio* | *Original* | *50/50* | *Exact ratio* |
| Llama2-7b | 74.30 | 52.34 | 72.02 | 81.55 | 54.45 | 78.27 |
| Llama2-70b | 74.25 | 57.72 | 72.58 | 81.38 | 62.13 | 79.13 |
| Llama3-8b | 79.98 | 70.20 | 78.86 | 80.40 | 70.87 | 82.57 |
| Llama3-8b-instruct | 80.97 | 69.09 | 79.93 | 84.73 | 71.40 | 83.01 |
| Llama3-70b-instruct | 80.88 | 66.84 | 79.50 | 84.67 | 71.27 | 82.99 |

Table 26: Evaluation accuracy on TriviaQA *Original* training set. Test sets remain the same as the table above.

| $f_G$ | GPT-3.5-turbo | | | GPT-4o | | |
| $\mathcal{D}_{\text{test}}$ | *Original* | *50/50* | *Exact ratio* | *Original* | *50/50* | *Exact ratio* |
|---|---|---|---|---|---|---|
| Llama2-7b | 71.75 | 56.62 | 70.06 | 81.57 | 56.25 | 77.58 |
| Llama2-70b | 72.58 | 64.65 | 71.70 | 80.30 | 65.78 | 78.42 |
| Llama3-8b | 82.47 | 74.58 | 78.34 | 82.30 | 72.30 | 81.18 |
| Llama3-8b-instruct | 79.88 | 75.97 | 80.08 | 83.72 | 73.02 | 82.24 |
| Llama3-70b-instruct | 80.77 | 75.32 | 79.79 | 83.63 | 71.81 | 82.33 |

Table 27: Evaluation accuracy on MMLU *Exact ratio* training set. Size of training set = {class 0 = 1019, class 1 = 5705} for both generator models. For $\mathcal{D}_{\text{test}}$ = *Original*, the test size remains 2000; for $\mathcal{D}_{\text{test}}$ = *50/50*, size of test set = {class 0 = 254, class 1 = 254}; for $\mathcal{D}_{\text{test}}$ = *Exact ratio*, size of test set = {class 0 = 254, class 1 = 1426}. Size of training set = {class 0 = 1019, class 1 = 5705}.

| $f_G$ | GPT-3.5-turbo | | | GPT-4o | | |
| $\mathcal{D}_{\text{test}}$ | *Original* | *50/50* | *Exact ratio* | *Original* | *50/50* | *Exact ratio* |
|---|---|---|---|---|---|---|
| Llama2-7b | 71.03 | 56.21 | 84.40 | 85.89 | 53.60 | 84.90 |
| Llama2-70b | 69.83 | 54.81 | 82.12 | 83.78 | 54.72 | 83.02 |
| Llama3-8b | 70.68 | 59.80 | 81.27 | 84.65 | 56.55 | 84.04 |
| Llama3-8b-instruct | 69.15 | 61.20 | 78.16 | 81.42 | 56.11 | 80.93 |
| Llama3-70b-instruct | 71.52 | 60.49 | 81.98 | 84.43 | 58.55 | 83.85 |

Table 28: Evaluation accuracy on MMLU *Original* training set. Test sets remain the same as the table above.

| $f_G$ | GPT-3.5-turbo | | | GPT-4o | | |
| $\mathcal{D}_{\text{test}}$ | *Original* | *50/50* | *Exact ratio* | *Original* | *50/50* | *Exact ratio* |
|---|---|---|---|---|---|---|
| Llama2-7b | 55.57 | 55.23 | 56.84 | 85.72 | 55.37 | 84.82 |
| Llama2-70b | 59.30 | 55.67 | 61.38 | 81.38 | 54.96 | 80.77 |
| Llama3-8b | 62.17 | 60.48 | 62.38 | 62.84 | 59.56 | 83.25 |
| Llama3-8b-instruct | 58.08 | 60.59 | 62.46 | 85.36 | 57.73 | 80.52 |
| Llama3-70b-instruct | 66.45 | 63.38 | 65.86 | 85.93 | 59.62 | 83.21 |

Table 29: Evaluation balanced accuracy on TriviaQA *Exact ratio* training set. Size of training set = {class 0 = 2162, class 1 = 5297} for both generator models. For $\mathcal{D}_{\text{test}}$ = *Original*, the test size remains 2000; for $\mathcal{D}_{\text{test}}$ = *50/50*, size of test set = {class 0 = 540, class 1 = 540}; for $\mathcal{D}_{\text{test}}$ = *Exact ratio*, size of test set = {class 0 = 540, class 1 = 1324}. Size of training set = {class 0 = 2162, class 1 = 5297}.

| $f_G$ | GPT-3.5-turbo | | | GPT-4o | | |
| $\mathcal{D}_{\text{test}}$ | *Original* | *50/50* | *Exact ratio* | *Original* | *50/50* | *Exact ratio* |
|---|---|---|---|---|---|---|
| Llama2-7b | 52.23 | 52.03 | 51.94 | 53.66 | 53.84 | 54.11 |
| Llama2-70b | 57.42 | 57.32 | 56.97 | 61.88 | 61.69 | 62.36 |
| Llama3-8b | 68.87 | 70.02 | 69.32 | 70.42 | 70.71 | 70.54 |
| Llama3-8b-instruct | 68.29 | 68.91 | 69.26 | 71.68 | 71.21 | 71.48 |
| Llama3-70b-instruct | 67.15 | 66.66 | 66.63 | 71.18 | 70.99 | 71.62 |

Table 30: Evaluation balanced accuracy on TriviaQA *Original* training set. Test sets remain the same as the table above.

| $f_G$ | GPT-3.5-turbo | | | GPT-4o | | |
| $\mathcal{D}_{\text{test}}$ | *Original* | *50/50* | *Exact ratio* | *Original* | *50/50* | *Exact ratio* |
|---|---|---|---|---|---|---|
| Llama2-7b | 59.60 | 56.58 | 57.36 | 56.42 | 55.97 | 56.86 |
| Llama2-70b | 65.39 | 64.64 | 66.13 | 64.70 | 65.55 | 65.49 |
| Llama3-8b | 74.02 | 74.51 | 75.24 | 71.41 | 72.21 | 71.77 |
| Llama3-8b-instruct | 75.05 | 75.89 | 72.58 | 72.29 | 72.85 | 72.85 |
| Llama3-70b-instruct | 75.34 | 75.36 | 73.68 | 74.12 | 71.67 | 72.27 |

Table 31: Evaluation balanced accuracy on MMLU *Exact ratio* training set. Size of training set = {class $0 = 1019$, class $1 = 5705$} for both generator models. For $\mathcal{D}_{\text{test}} =$ *Original*, the test size remains 2000; for $\mathcal{D}_{\text{test}} =$ *50/50*, size of test set = {class $0 = 254$, class $1 = 254$}; for $\mathcal{D}_{\text{test}} =$ *Exact ratio*, size of test set = {class $0 = 254$, class $1 = 1426$}. Size of training set = {class $0 = 1019$, class $1 = 5705$}.

| $f_G$ | GPT-3.5-turbo | | | GPT-4o | | |
| $\mathcal{D}_{\text{test}}$ | *Original* | *50/50* | *Exact ratio* | *Original* | *50/50* | *Exact ratio* |
|---|---|---|---|---|---|---|
| Llama2-7b | 55.67 | 56.01 | 56.54 | 53.66 | 53.44 | 53.75 |
| Llama2-70b | 54.47 | 54.65 | 56.36 | 53.54 | 54.51 | 54.67 |
| Llama3-8b | 58.52 | 59.68 | 59.65 | 56.56 | 56.46 | 56.05 |
| Llama3-8b-instruct | 60.58 | 61.05 | 60.54 | 56.51 | 56.31 | 55.85 |
| Llama3-70b-instruct | 58.90 | 60.25 | 61.17 | 58.98 | 58.36 | 59.28 |

Table 32: Evaluation balanced accuracy on MMLU *Original* training set. Test sets remain the same as the table above.

| $f_G$ | GPT-3.5-turbo | | | GPT-4o | | |
| $\mathcal{D}_{\text{test}}$ | *Original* | *50/50* | *Exact ratio* | *Original* | *50/50* | *Exact ratio* |
|---|---|---|---|---|---|---|
| Llama2-7b | 55.17 | 55.52 | 56.05 | 54.46 | 55.23 | 53.68 |
| Llama2-70b | 55.58 | 55.67 | 55.48 | 54.29 | 54.76 | 54.08 |
| Llama3-8b | 59.46 | 60.49 | 62.44 | 59.04 | 59.39 | 59.20 |
| Llama3-8b-instruct | 60.10 | 60.63 | 63.14 | 57.21 | 57.49 | 57.50 |
| Llama3-70b-instruct | 63.80 | 63.45 | 65.75 | 59.45 | 59.51 | 59.93 |

Table 33: Evaluation AUPRC on TriviaQA *Exact ratio* training set. Size of training set = {class $0 = 2162$, class $1 = 5297$} for both generator models. For $\mathcal{D}_{\text{test}} =$ *Original*, the test size remains 2000; for $\mathcal{D}_{\text{test}} =$ *50/50*, size of test set = {class $0 = 540$, class $1 = 540$}; for $\mathcal{D}_{\text{test}} =$ *Exact ratio*, size of test set = {class $0 = 540$, class $1 = 1324$}. Size of training set = {class $0 = 2162$, class $1 = 5297$}.

| $f_G$ | GPT-3.5-turbo | | | GPT-4o | | |
| $\mathcal{D}_{\text{test}}$ | *Original* | *50/50* | *Exact ratio* | *Original* | *50/50* | *Exact ratio* |
|---|---|---|---|---|---|---|
| Llama2-7b | 77.99 | 57.34 | 75.74 | 85.40 | 59.66 | 82.75 |
| Llama2-70b | 82.74 | 64.20 | 80.98 | 89.22 | 68.89 | 87.35 |
| Llama3-8b | 90.36 | 79.10 | 89.20 | 93.27 | 78.95 | 91.85 |
| Llama3-8b-instruct | 90.62 | 79.08 | 90.62 | 93.97 | 80.82 | 92.73 |
| Llama3-70b-instruct | 90.95 | 79.30 | 89.80 | 93.85 | 80.88 | 92.63 |

Table 34: Evaluation AUPRC on TriviaQA *Original* training set. Test sets remain the same as the table above.

| $f_G$ $\mathcal{D}_{\text{test}}$ | GPT-3.5-turbo | | | GPT-4o | | |
|---|---|---|---|---|---|---|
| | *Original* | *50/50* | *Exact ratio* | *Original* | *50/50* | *Exact ratio* |
| Llama2-7b | 81.30 | 59.23 | 78.71 | 85.75 | 58.07 | 82.62 |
| Llama2-70b | 85.59 | 68.04 | 84.25 | 89.67 | 67.54 | 87.85 |
| Llama3-8b | 91.89 | 79.91 | 91.04 | 93.47 | 78.21 | 92.17 |
| Llama3-8b-instruct | 92.67 | 82.28 | 90.12 | 94.37 | 80.56 | 93.10 |
| Llama3-70b-instruct | 92.80 | 82.69 | 94.20 | 94.13 | 79.95 | 92.92 |

Table 35: Evaluation AUPRC on MMLU *Exact ratio* training set. Size of training set = {class 0 = 1019, class 1 = 5705} for both generator models. For $\mathcal{D}_{\text{test}}$ = *Original*, the test size remains 2000; for $\mathcal{D}_{\text{test}}$ = *50/50*, size of test set = {class 0 = 254, class 1 = 254}; for $\mathcal{D}_{\text{test}}$ = *Exact ratio*, size of test set = {class 0 = 254, class 1 = 1426}. Size of training set = {class 0 = 1019, class 1 = 5705}.

| $f_G$ $\mathcal{D}_{\text{test}}$ | GPT-3.5-turbo | | | GPT-4o | | |
|---|---|---|---|---|---|---|
| | *Original* | *50/50* | *Exact ratio* | *Original* | *50/50* | *Exact ratio* |
| Llama2-7b | 77.99 | 57.34 | 75.74 | 85.40 | 59.66 | 82.75 |
| Llama2-70b | 82.74 | 64.20 | 80.98 | 89.22 | 68.89 | 87.35 |
| Llama3-8b | 90.36 | 79.10 | 89.20 | 93.27 | 78.95 | 91.85 |
| Llama3-8b-instruct | 90.62 | 79.08 | 90.62 | 93.97 | 80.82 | 92.73 |
| Llama3-70b-instruct | 90.95 | 79.30 | 89.80 | 93.85 | 80.88 | 92.63 |

Table 36: Evaluation AUPRC on MMLU *Original* training set.. Test sets remain the same as the table above.

| $f_G$ $\mathcal{D}_{\text{test}}$ | GPT-3.5-turbo | | | GPT-4o | | |
|---|---|---|---|---|---|---|
| | *Original* | *50/50* | *Exact ratio* | *Original* | *50/50* | *Exact ratio* |
| Llama2-7b | 74.78 | 59.23 | 78.71 | 86.81 | 58.07 | 82.62 |
| Llama2-70b | 73.78 | 68.04 | 84.25 | 86.97 | 67.54 | 87.85 |
| Llama3-8b | 78.82 | 79.91 | 91.04 | 90.07 | 78.21 | 92.17 |
| Llama3-8b-instruct | 78.00 | 82.28 | 91.96 | 90.42 | 80.56 | 93.10 |
| Llama3-70b-instruct | 82.88 | 82.69 | 92.05 | 91.13 | 79.95 | 92.92 |

# F   Predictor Performance on TriviaQA by Difficulty Level

We provide full experimental results in Section 4.3 as follows.

Table 37: Predictor performance on TriviaQA-*Easy* evaluated across eight metrics and Llama backbones using the strong generator model $f_G$ = GPT-3.5-turbo.

| $f_P$ | AUROC | Accuracy | Balanced Acc. | AUPRC | F1 Score | Sel. Balanced Acc. | Sel. AUROC | Sel. F1 |
|---|---|---|---|---|---|---|---|---|
| Llama3-70b-instruct | 78.89 | 74.08 | 71.97 | 87.26 | 80.57 | 85.90 | 81.65 | 88.57 |
| Llama3-8b-instruct | 76.62 | 72.15 | 69.22 | 86.06 | 79.23 | 84.78 | 80.14 | 87.44 |
| Llama3-8b | 76.00 | 72.08 | 69.40 | 85.72 | 78.99 | 81.79 | 78.15 | 86.61 |
| Llama2-70b | 67.00 | 67.37 | 62.34 | 79.09 | 76.73 | 73.49 | 70.11 | 83.27 |
| Llama2-7b | 65.58 | 66.97 | 61.98 | 76.82 | 79.83 | 68.19 | 67.17 | 82.87 |

Table 38: Predictor performance on TriviaQA-*Medium* evaluated across eight metrics and Llama backbones using the strong generator model $f_G$ = GPT-3.5-turbo.

| $f_P$ | AUROC | Accuracy | Balanced Acc. | AUPRC | F1 Score | Sel. Balanced Acc. | Sel. AUROC | Sel. F1 |
|---|---|---|---|---|---|---|---|---|
| Llama3-70b-instruct | 77.55 | 81.00 | 71.83 | 92.34 | 88.37 | 84.35 | 80.67 | 93.68 |
| Llama3-8b-instruct | 75.55 | 80.18 | 69.35 | 91.54 | 88.06 | 82.30 | 78.15 | 93.37 |
| Llama3-8b | 74.85 | 79.83 | 68.89 | 91.31 | 87.76 | 79.26 | 76.37 | 93.45 |
| Llama2-70b | 66.78 | 79.85 | 61.93 | 87.70 | 88.46 | 72.64 | 69.76 | 90.97 |
| Llama2-7b | 64.45 | 79.92 | 62.75 | 85.70 | 88.83 | 65.04 | 65.00 | 90.56 |

Table 39: Predictor performance on TriviaQA-*Hard* evaluated across eight metrics and Llama backbones using the strong generator model $f_G$ = GPT-3.5-turbo.

| $f_P$ | AUROC | Accuracy | Balanced Acc. | AUPRC | F1 Score | Sel. Balanced Acc. | Sel. AUROC | Sel. F1 |
|---|---|---|---|---|---|---|---|---|
| Llama3-70b-instruct | 75.24 | 90.92 | 69.34 | 96.42 | 95.19 | 80.12 | 78.85 | 96.83 |
| Llama3-8b-instruct | 73.83 | 91.42 | 67.07 | 96.34 | 95.52 | 77.86 | 75.74 | 96.57 |
| Llama3-8b | 73.06 | 91.23 | 66.60 | 96.35 | 95.39 | 78.58 | 74.63 | 96.77 |
| Llama2-70b | 64.55 | 91.42 | 61.23 | 94.84 | 95.52 | 68.18 | 67.04 | 95.78 |
| Llama2-7b | 61.82 | 91.43 | 58.94 | 93.75 | 95.52 | 61.03 | 61.26 | 95.34 |

## G    Predictor Performance on MMLU with Different Input Representation Aggregations

We provide full experimental results for our analysis in Section 4.4 as follows.

Table 40: Predictor performance on MMLU using representations from intermediate layers with the strong generator model $f_G$ = GPT-3.5-turbo.

| $f_P$ | AUROC | Accuracy | Balanced Acc. | AUPRC | F1 Score | Sel. Balanced Acc. | Sel. AUROC | Sel. F1 |
|---|---|---|---|---|---|---|---|---|
| Llama3-70b-instruct | 69.86 | 71.37 | 64.42 | 83.92 | 81.84 | 74.86 | 71.86 | 88.39 |
| Llama3-8b-instruct | 66.17 | 70.95 | 62.22 | 80.55 | 82.29 | 70.84 | 68.18 | 86.53 |
| Llama3-8b | 65.50 | 71.05 | 61.86 | 79.72 | 82.20 | 67.29 | 69.17 | 86.40 |
| Llama2-70b | 57.67 | 70.22 | 55.43 | 74.65 | 81.79 | 60.19 | 59.03 | 84.31 |
| Llama2-7b | 57.59 | 70.57 | 55.98 | 75.09 | 82.40 | 62.32 | 60.21 | 84.36 |

Table 41: Predictor performance on MMLU using representations obtained via average pooling with the strong generator model $f_G$ = GPT-3.5-turbo.

| $f_P$ | AUROC | Accuracy | Balanced Acc. | AUPRC | F1 Score | Sel. Balanced Acc. | Sel. AUROC | Sel. F1 |
|---|---|---|---|---|---|---|---|---|
| Llama3-70b-instruct | 68.47 | 71.73 | 62.99 | 82.50 | 81.77 | 70.18 | 73.19 | 87.20 |
| Llama3-8b-instruct | 65.44 | 69.72 | 60.47 | 79.74 | 80.49 | 68.07 | 65.62 | 86.21 |
| Llama3-8b | 65.05 | 69.73 | 60.25 | 79.63 | 80.55 | 65.97 | 68.16 | 86.17 |
| Llama2-70b | 59.16 | 67.68 | 56.70 | 75.87 | 79.31 | 61.15 | 63.31 | 83.91 |
| Llama2-7b | 57.52 | 70.10 | 55.57 | 74.52 | 82.27 | 60.67 | 58.82 | 83.33 |

## H    Ablation study: Baseline comparison with Self-consistency, Disagreement, and Confidence Prompting

This section contains simple black-box baselines that use multiple sampled outputs from the generator: self consistency (SC), disagreement (DA), and confidence prompting (CP). For each question, we normalize the sampled responses and group identical normalized strings as the same answer. SC is the fraction of samples assigned to the majority answer, DA is the negative entropy of the empirical answer distribution, and CP uses a post-hoc True or False self-evaluation from the generator as a confidence score.

Across both generators, the trained probe baselines generally outperform self consistency and disagreement on most datasets, while performance on MMLU and SuperGPQA is comparable, with SC or DA sometimes slightly higher. CP is typically weaker than the probe baselines. TriviaQA is an exception, where SC and DA are unusually strong because naive string matching tends to split incorrect answers into many small groups, which can artificially inflate these metrics. Unlike these black-box baselines, which require $k$ generator calls at inference time, the probe method uses embeddings and does not require sampling at test time.

Table 42: AUROC comparison between weak-to-strong probe predictors and black-box multi-sample baselines, with $f_G$ = GPT-3.5-turbo. SC, DA, and CP require repeated generator queries at inference time, whereas the probe-based predictor uses question embeddings after the probe has been trained. For consistency with Table 2, we report Llama3-70b-instruct as a representative strong Llama-family probe backbone. SC and DA are unusually strong on TriviaQA due to its free-response format, as naive string matching can split incorrect open-ended answers into small separate clusters and inflate agreement-based confidence estimates.

| Baseline Method / $f_p$ | TriviaQA | CounterFact | Winogrande | MMLU | MedQA | SuperGPQA |
|---|---|---|---|---|---|---|
| Self-consistency (SC) | 97.03 | 72.75 | 58.45 | 70.32 | 60.06 | 64.22 |
| Disagreement (DA) | 96.92 | 72.79 | 58.46 | 70.45 | 60.88 | 64.98 |
| Confidence Prompting (CP) | 54.06 | 73.81 | 51.63 | 60.10 | 59.43 | 57.12 |
| Llama3-70b-instruct | 84.15 | 85.28 | 68.62 | 68.91 | 65.40 | 52.38 |

# I   Ablation study: $y_{true}$ label sensitivity

We provide a comprehensive sensitivity analysis of the predictor's AUROC with respect to decoding hyperparameters, including sampling temperature $(0.2, 0.8, 1.0, 1.2)$, the nucleus sampling threshold $top_p$ $(0.9, 1.0)$ [1], and the sampling budget $k$ $(5, 10, 20)$.

We find that small predictor backbones such as Llama3-8b still provide effective weak-to-strong confidence signals, and knowledge retrieval tasks remain easier to predict than reasoning benchmarks. Halving the sample budget from 10 to 5 usually causes a small drop in AUROC, while doubling it from 10 to 20 often yields modest gains, but neither change alters the overall gains from selective prediction or the main conclusions about representational compatibility. Together, these observations provide evidence that successful weak-to-strong confidence prediction is dependent primarily on the underlying representations rather than by a particular choice of decoding hyperparameters or sampling budget.

Table 43: Testing TriviaQA predictor AUROC sensitivity against sampling temperature and topp with $f_G = $ GPT-4o, $k = 10$. Lower temperature and slightly lower top-$p$ generally improve AUROC relative to the baseline across all five predictor families.

| $f_P$ | $temp{=}1.0$, top-$p{=}1.0$ | $temp{=}0.2$, top-$p{=}1.0$ | $temp{=}0.8$, top-$p{=}1.0$ | $temp{=}1.2$, top-$p{=}1.0$ | $temp{=}1.0$, top-$p{=}0.9$ |
|---|---|---|---|---|---|
| Llama3-8b | 80.06 | 83.56 | 83.62 | 83.43 | 83.16 |
| Mistral-22b | 67.53 | 70.76 | 70.44 | 70.13 | 71.42 |
| Gemma3-12b | 76.99 | 77.37 | 77.78 | 77.33 | 77.78 |
| GPT-oss-20b | 79.69 | 83.46 | 83.31 | 83.21 | 83.23 |
| Gemini-Embedding-001 | 76.42 | 80.65 | 80.52 | 79.90 | 80.08 |

Table 44: Testing CounterFact predictor AUROC sensitivity against sampling temperature and topp with $f_G = $ GPT-4o, $k = 10$. Lower randomness hurts the performance worse, as this dataset is highly imbalanced and thus sensitive to more constrained decoding.

| $f_P$ | $temp{=}1.0$, top-$p{=}1.0$ | $temp{=}0.2$, top-$p{=}1.0$ | $temp{=}0.8$, top-$p{=}1.0$ | $temp{=}1.2$, top-$p{=}1.0$ | $temp{=}1.0$, top-$p{=}0.9$ |
|---|---|---|---|---|---|
| Llama3-8b | 81.95 | 76.46 | 77.33 | 78.21 | 75.91 |
| Mistral-22b | 72.68 | 63.27 | 64.20 | 65.23 | 64.73 |
| Gemma3-12b | 77.15 | 67.49 | 67.19 | 65.87 | 64.52 |
| GPT-oss-20b | 78.97 | 74.12 | 76.82 | 77.59 | 75.25 |
| Gemini-Embedding-001 | 68.47 | 64.76 | 65.27 | 67.01 | 63.88 |

Table 45: Testing Winogrande predictor AUROC sensitivity against sampling temperature and topp with $f_G = $ GPT-4o, $k = 10$. AUROC is relatively stable across decoding settings, with modest variation across predictors. Only Gemma-3-12b shows more sensitivity.

| $f_P$ | $temp{=}1.0$, top-$p{=}1.0$ | $temp{=}0.2$, top-$p{=}1.0$ | $temp{=}0.8$, top-$p{=}1.0$ | $temp{=}1.2$, top-$p{=}1.0$ | $temp{=}1.0$, top-$p{=}0.9$ |
|---|---|---|---|---|---|
| Llama3-8b | 59.56 | 57.64 | 58.97 | 58.85 | 58.57 |
| Mistral-22b | 56.19 | 53.27 | 54.82 | 54.03 | 54.32 |
| Gemma3-12b | 66.11 | 58.22 | 58.26 | 58.03 | 58.12 |
| GPT-oss-20b | 60.93 | 59.54 | 59.44 | 60.18 | 58.46 |
| Gemini-Embedding-001 | 65.06 | 66.34 | 65.85 | 66.35 | 65.99 |

Table 46: Testing MMLU predictor AUROC sensitivity against sampling temperature and topp with $f_G = $ GPT-4o, $k = 10$. The level of sensitivity is highly mixed. Some models like Mistral-22b can improve significantly with lower randomness, while other models are quite insensitive to the change of these sampling hyperparameters.

| $f_P$ | $temp{=}1.0$, top-$p{=}1.0$ | $temp{=}0.2$, top-$p{=}1.0$ | $temp{=}0.8$, top-$p{=}1.0$ | $temp{=}1.2$, top-$p{=}1.0$ | $temp{=}1.0$, top-$p{=}0.9$ |
|---|---|---|---|---|---|
| Llama3-8b | 62.84 | 64.23 | 63.98 | 63.60 | 63.93 |
| Mistral-22b | 57.79 | 70.76 | 70.44 | 55.44 | 71.42 |
| Gemma3-12b | 55.78 | 55.93 | 55.79 | 55.48 | 55.90 |
| GPT-oss-20b | 64.77 | 63.22 | 64.26 | 63.88 | 64.06 |
| Gemini-Embedding-001 | 59.53 | 59.49 | 59.59 | 59.30 | 60.04 |

Table 47: Testing MedQA predictor AUROC sensitivity against sampling temperature and topp with $f_G =$ GPT-4o, $k = 10$. AUROC is relatively stable across decoding settings, with modest variation across predictors.

| $f_P$ | $temp=1.0$, top-$p=1.0$ | $temp=0.2$, top-$p=1.0$ | $temp=0.8$, top-$p=1.0$ | $temp=1.2$, top-$p=1.0$ | $temp=1.0$, top-$p=0.9$ |
|---|---|---|---|---|---|
| Llama3-8b | 62.06 | 60.88 | 60.17 | 62.43 | 60.80 |
| Mistral-22b | 58.03 | 57.06 | 56.82 | 56.73 | 57.39 |
| Gemma3-12b | 59.92 | 59.90 | 60.20 | 61.40 | 60.35 |
| GPT-oss-20b | 63.55 | 62.64 | 62.09 | 63.55 | 61.56 |
| Gemini-Embedding-001 | 62.52 | 61.21 | 60.89 | 62.48 | 62.31 |

Table 48: Testing SuperGPQA predictor AUROC sensitivity against sampling temperature and topp with $f_G =$ GPT-4o, $k = 10$. AUROC is relatively stable across decoding settings, with modest variation across predictors.

| $f_P$ | $temp=1.0$, top-$p=1.0$ | $temp=0.2$, top-$p=1.0$ | $temp=0.8$, top-$p=1.0$ | $temp=1.2$, top-$p=1.0$ | $temp=1.0$, top-$p=0.9$ |
|---|---|---|---|---|---|
| Llama3-8b | 59.90 | 65.84 | 66.01 | 67.11 | 67.00 |
| Mistral-22b | 57.46 | 61.47 | 62.78 | 63.72 | 63.48 |
| Gemma3-12b | 61.85 | 64.84 | 64.79 | 66.35 | 66.24 |
| GPT-oss-20b | 66.99 | 65.06 | 65.92 | 67.83 | 66.72 |
| Gemini-Embedding-001 | 66.93 | 64.71 | 65.13 | 66.98 | 66.38 |

Table 49: Testing TriviaQA predictor AUROC sensitivity against different numbers of sample responses $k$ with $f_G =$ GPT-4o. Halving the sample budget from 10 to 5 causes only a small drop in AUROC, while doubling it from 10 to 20 yields modest gains.

| $f_P$ | $K = 5$ | $K = 10$ | $K = 20$ |
|---|---|---|---|
| Llama3-8b | 79.50 | 80.06 | 82.36 |
| Mistral-22b | 67.22 | 67.53 | 69.71 |
| Gemma3-12b | 73.94 | 76.99 | 77.03 |
| GPT-oss-20b | 79.25 | 79.69 | 82.02 |
| Gemini-Embedding-001 | 75.75 | 76.42 | 78.99 |

Table 50: Testing CounterFact predictor AUROC sensitivity against different numbers of sample responses $k$ with $f_G =$ GPT-4o. Halving the sample budget from 10 to 5 causes only a small drop in AUROC, while doubling it from 10 to 20 yields modest gains.

| $f_P$ | $K = 5$ | $K = 10$ | $K = 20$ |
|---|---|---|---|
| Llama3-8b | 78.38 | 81.95 | 80.07 |
| Mistral-22b | 64.34 | 72.68 | 65.78 |
| Gemma3-12b | 69.03 | 77.15 | 66.97 |
| GPT-oss-20b | 78.22 | 78.97 | 79.07 |
| Gemini-Embedding-001 | 63.55 | 68.47 | 70.16 |

Table 51: Testing Winogrande predictor AUROC sensitivity against different numbers of sample responses $k$ with $f_G =$ GPT-4o. Halving the sample budget from 10 to 5 causes only a small drop in AUROC, while doubling it from 10 to 20 generally yields modest gains.

| $f_P$ | $K = 5$ | $K = 10$ | $K = 20$ |
|---|---|---|---|
| Llama3-8b | 55.89 | 59.96 | 57.50 |
| Mistral-22b | 54.00 | 56.19 | 54.34 |
| Gemma3-12b | 56.55 | 66.11 | 59.42 |
| GPT-oss-20b | 60.09 | 60.93 | 61.41 |
| Gemini-Embedding-001 | 61.73 | 65.06 | 67.23 |

Table 52: Testing MMLU predictor AUROC sensitivity against different numbers of sample responses $k$ with $f_G = \text{GPT-4o}$. Halving the sample budget from 10 to 5 causes only a small drop in AUROC, while doubling it from 10 to 20 generally yields modest gains.

| $f_P$ | $K = 5$ | $K = 10$ | $K = 20$ |
|---|---|---|---|
| Llama3-8b | 62.29 | 62.84 | 63.74 |
| Mistral-22b | 55.36 | 57.79 | 55.74 |
| Gemma3-12b | 55.03 | 55.78 | 55.59 |
| GPT-oss-20b | 63.93 | 64.77 | 63.71 |
| Gemini-Embedding-001 | 58.87 | 59.53 | 59.71 |

Table 53: Testing MedQA predictor AUROC sensitivity against different numbers of sample responses $k$ with $f_G = \text{GPT-4o}$. Halving the sample budget from 10 to 5 usually causes only a small drop in AUROC, while doubling it from 10 to 20 generally yields modest gains.

| $f_P$ | $K = 5$ | $K = 10$ | $K = 20$ |
|---|---|---|---|
| Llama3-8b | 61.25 | 62.06 | 61.74 |
| Mistral-22b | 57.42 | 58.03 | 58.54 |
| Gemma3-12b | 61.25 | 59.92 | 61.44 |
| GPT-oss-20b | 63.71 | 63.55 | 63.21 |
| Gemini-Embedding-001 | 60.88 | 62.52 | 64.06 |

Table 54: Testing SuperGPQA predictor AUROC sensitivity against different numbers of sample responses $k$ with $f_G = \text{GPT-4o}$. Halving the sample budget from 10 to 5 sometimes causes only a small drop in AUROC, while doubling it from 10 to 20 often yields modest gains.

| $f_P$ | $K = 5$ | $K = 10$ | $K = 20$ |
|---|---|---|---|
| Llama3-8b | 67.44 | 59.90 | 66.90 |
| Mistral-22b | 63.65 | 57.46 | 63.30 |
| Gemma3-12b | 66.56 | 61.85 | 65.95 |
| GPT-oss-20b | 66.35 | 66.99 | 66.83 |
| Gemini-Embedding-001 | 66.17 | 66.93 | 66.84 |

## J  Ablation study: Out-of-distribution (OOD) Performance

This appendix provides an additional visualization of the cross dataset transfer results discussed in Section 4.6. Figure 19 shows the same source to target transfer pattern as Table 3, with diagonal entries corresponding to in-distribution evaluation and off-diagonal entries corresponding to zero-shot transfer across datasets. The heatmap highlights the same qualitative conclusion: transfer performance is weak across unrelated datasets.

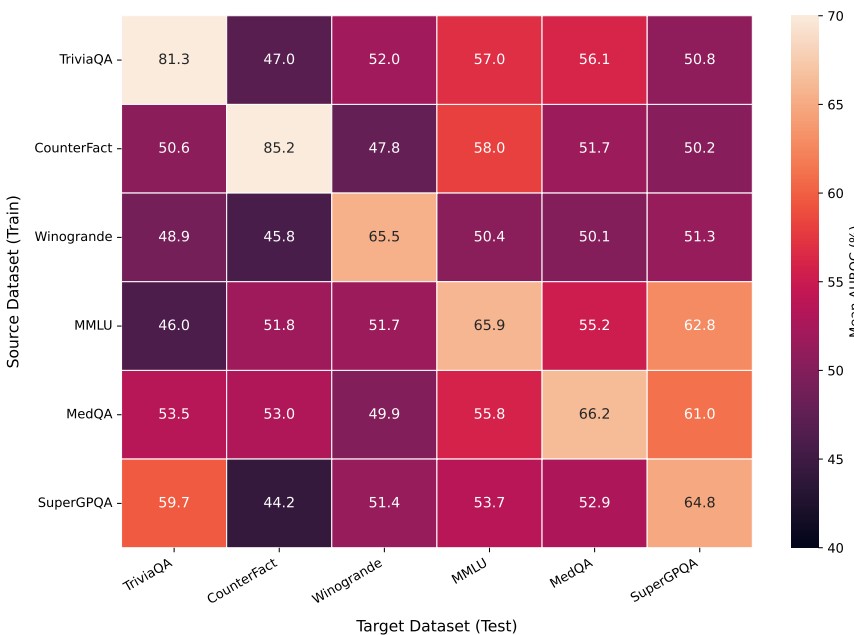

Figure 19: Heatmap visualization of cross-dataset transfer AUROC. Rows denote the source dataset used to train the probe, and columns denote the target dataset used for evaluation. Diagonal entries correspond to in-distribution evaluation, while off-diagonal entries correspond to zero-shot transfer. Results are shown for GPT-4o as the generator and Llama3-70B-Instruct as the predictor backbone.

