# OpenReview forum: "A Systematic Assessment of Weak-to-Strong Confidence Prediction in Large Language Models"
_TMLR — Accepted by TMLR_

### Review · Reviewer_NPrr · 2026-01-12

**Summary Of Contributions:**

This paper presents a systematic evaluation of weak-to-strong confidence prediction, a framework where the correctness of a "strong" black-box generator is predicted by a "weak" open-access model. The authors use simple two-layer MLP probes trained on frozen, question-only embeddings to estimate the probability that the generator will provide a correct answer.

**Summary of Contributions**

- Framework and Systematic Analysis: The authors formalize and evaluate an external, data-driven approach to uncertainty quantification that does not require internal access to a model's log-probabilities or hidden state

- Empirical Performance: Across six benchmarks, the method improves selective prediction accuracy

- "Representational Compatibility" over Scale: A key finding is that the size of the predictor model is not the primary determinant of success.

- Robustness Evaluations: The authors demonstrate that the predictive signal is robust across various label distributions and different embedding aggregation strategies


**Key Strengths**

- Broad Experimental Scope: The study evaluates 15 different predictor models against 2 state-of-the-art generators across 6 diverse datasets spanning knowledge recall and reasoning.


- Practical Utility: The method is computationally efficient and applicable to proprietary "black-box" models where only the final text output is visible.


- Insightful Ablations: The paper goes beyond simple performance metrics to disentangle whether models are truly learning task-specific features or merely mimicking label priors

**Key Weaknesses**

- Performance Bottleneck on Reasoning Tasks: On complex reasoning datasets like SuperGPQA, the absolute accuracy remains low. Even at a 50% rejection rate, the selective accuracy only reaches 57.91%, which is significantly lower than on factual recall tasks

- Limitations of Question-Only Embeddings: As noted in the paper, since the predictor only sees the question, it often fails on reasoning tasks where the correctness signal is hidden in the intermediate steps of the generated answer. Figure 4 suggests that embeddings for logical reasoning tasks are fundamentally less predictive than those for knowledge recall.

- Fairness in Reporting (Table 1): Table 1 reports the "best selective accuracy among all predictors". This may be misleading because, as shown in the scaling analysis, there is no consistent "best" model across all tasks; a larger Gemma model might underperform a smaller one depending on the benchmark

- Missing Baseline Comparisons: The paper lacks a direct comparison with simpler UQ methods, such as direct prompting (asking the model "How confident are you?"). While the authors argue these methods are brittle, they do not provide empirical evidence that their embedding-based probe outperforms them

**Audience:**

Yes

**Audience Explanation:**

The paper addresses several high-interest topics for the TMLR community, particularly those focused on the reliability, safety, and internal representations of Large Language Models

**Claims And Evidence:**

No

**Claims Explanation:**

Weakness in Table 1 and Selective Accuracy Claims
- A primary claim is that the framework improves generator reliability in selective prediction. However, the evidence in Table 1 and the surrounding analysis presents several issues:

- Inconsistent Gains on Reasoning Tasks: On the graduate-level SuperGPQA dataset, the initial accuracy for GPT-4o is 44.98%. Even with a 50% rejection rate, the selective accuracy only reaches 57.91%. This indicates that the method is far less effective for reasoning-intensive tasks than for factual recall (e.g., TriviaQA), where accuracy jumps from 81.47% to 93.77% at the same rejection rate.

- Selective Reporting: Table 1 reports only the "best selective accuracy among all predictors" for each rejection rate. This is a "cherry-picked" upper bound that obscures the fact that performance across predictors is highly inconsistent.

- Lack of Practical Predictive Power: The authors admit that logical reasoning tasks are challenging because the predictor only sees the question and misses the intermediate reasoning steps in the answer. This suggests the method may not be general enough to act as a reliable "oracle" for anything beyond factual retrieval

Missing Baselines and Comparison with Internal UQ

- The paper positions this as a "pragmatic baseline" for black-box uncertainty quantification. However, the evidence is not yet convincing because it lacks comparison to established methods

**Requested Changes:**

Comparison with Introspective Baselines: A primary contribution claim is that this external framework provides a pragmatic alternative to "introspective" signals like self-reported probabilities. However, the paper lacks a direct empirical comparison against a simple zero-shot confidence prompt

Fairer Reporting in Table 1: Table 1 currently reports only the "best selective accuracy among all predictors" for each task. Given that the authors admit scaling trends are not universal and larger models sometimes underperform smaller ones , reporting only the upper bound obscures the variance and reliability of the method.

Analysis of Reasoning Failures: The selective accuracy gains on reasoning-heavy datasets like SuperGPQA are notably lower than on factual recall tasks. A deeper qualitative analysis of why question-only embeddings fail in these cases—beyond the general statement that they miss "internal steps"—would significantly strengthen the "Insights" section.

---

> ### Author Response · Authors · 2026-02-28
> **Response to Reviewer NPrr (1/3)**
>
> Dear Reviewer NPrr,
>
> We sincerely thank you for the careful and detailed review. We are encouraged that you highlighted three key strengths: the broad experimental scope, the practical utility for black-box generators, and the value of our ablation studies.
>
> We address your concerns below.
>
> ---
> > "Comparison with Introspective Baselines: A primary contribution claim is that this external framework provides a pragmatic alternative to \"introspective\" signals like self-reported probabilities. However, the paper lacks a direct empirical comparison against a simple zero-shot confidence prompt"
>
> We appreciate your suggestion to add additional black-box baselines based on multiple sampled outputs. We agree that they would be natural reference points.
>
> For each question, we first normalized each sampled response and then treat identical normalized strings as the same answer.
> Self-consistency (SC) [1] is computed as the fraction of samples that match the majority answer.
> Disagreement (DA) [2] is computed as the negative entropy of the empirical answer distribution across samples, so that higher values indicate higher confidence.
> Confidence prompting (CP) follows Kadavath et al. [3] and performs post hoc self evaluation: we prompt the generator to judge whether a proposed answer is True or False, and use this judgement as a confidence score for predicting answer correctness.
> Note that these baselines all require $k$ generator calls at inference time, while the simple probe-based method considered in our manuscript uses only small-model embeddings and a probe trained on $k$ generator calls obtained upfront, but does not require any samples at test time.
>
> We find that the trained probes outperform both self consistency (SC) and disagreement (DA) on most datasets; on MMLU and SuperGPQA the performance is comparable (but SC and DA require $k$ forward passes through the large generator model), with SC and DA marginally above the probe baseline. TriviaQA shows exceptionally high SC and DA performance because we directly identify correct answers and group the remaining incorrect answers using naive string matching, which often fragments incorrect answers into many small groups and artificially inflates the estimated SC and DA metrics. Confidence prompting is generally less worse than our probes.
> This matches our main finding that question-only embeddings provide strong signals on knowledge recall, while on reasoning-heavy tasks and post hoc self evaluation are often noisier and less calibrated.
>
> **Generator = GPT-3.5-turbo**
> | Metric \ Dataset        | TriviaQA | CounterFact | Winogrande | MMLU  | MedQA | SuperGPQA |
> | ---------------------- | -------- | ----------- | ---------- | ----- | ----- | --------- |
> | Self-Consistency AUROC | 97.03    | 72.75       | 58.45      | 70.32 | 60.06 | 64.22     |
> | Self-Consistency AUPRC | 97.69    | 97.25       | 73.97      | 82.18 | 66.74 | 49.37     |
> | Disagreement AUROC     | 96.92    | 72.79       | 58.46      | 70.45 | 60.88 | 64.98     |
> | Disagreement AUPRC     | 97.74    | 97.26       | 73.98      | 82.27 | 67.11 | 50.07     |
> | Prompt AUROC                | 54.06    | 73.81       | 51.63      | 60.10 | 59.43 | 57.12     |
> | Prompt AUPRC                | 68.11    | 96.56       | 68.48      | 76.93 | 65.00 | 43.43     |
> | Llama3-8b AUROC (Probe)     | 82.47    | 81.07       | 61.68     | 63.77 | 59.96 | 52.33     |
> | Llama3-8b AUPRC (Probe)     | 91.89    | 98.38       | 87.80      | 78.82 | 69.04 | 22.56     |
>
> **Generator = GPT-4o**
> | Metric                 | TriviaQA | CounterFact | Winogrande | MMLU  | MedQA | SuperGPQA |
> | ---------------------- | -------- | ----------- | ---------- | ----- | ----- | --------- |
> | Self-Consistency AUROC | 92.42    | 83.50       | 60.29      | 71.95 | 62.54 | 67.52     |
> | Self-Consistency AUPRC | 95.23    | 98.64       | 90.34      | 93.15 | 91.71 | 65.53     |
> | Disagreement AUROC     | 92.37    | 83.46       | 61.48      | 71.76 | 64.51 | 68.62     |
> | Disagreement AUPRC     | 95.26    | 98.64       | 90.55      | 93.15 | 92.06 | 66.32     |
> | Prompt AUROC           | 56.12    | 68.03       | 64.41      | 57.95 | 55.89 | 55.16     |
> | Prompt AUPRC           | 75.50    | 96.93       | 90.17      | 89.15 | 89.50 | 59.06     |
> | Llama3-8b AUROC (Probe)     | 80.06    | 81.95       | 59.56      | 62.84 | 62.06 | 59.90     |
> | Llama3-8b AUPRC (Probe)     | 93.47   | 98.43       | 94.45      | 90.07 | 91.71 | 50.45     |

---

> ### Author Response · Authors · 2026-02-28
> **Response to Reviewer NPrr (2/3)**
>
> > "Analysis of Reasoning Failures: The selective accuracy gains on reasoning-heavy datasets like SuperGPQA are notably lower than on factual recall tasks. A deeper qualitative analysis of why question-only embeddings fail in these cases—beyond the general statement that they miss \"internal steps\"—would significantly strengthen the \"Insights\" section."
>
> We agree that the current explanation for the performance gap on reasoning-focuses tasks could be further strengthened. We believe there could be a few additional plausible reasons, such as: question-only embeddings may only capture topical content and coarse difficulty, but they do not condition on latent intermediate steps that determine correctness in multi step reasoning; multiple failure modes exist in latent reasoning paths [4,5,6], increasing difficulty in separability for probes; latent reasoning chains could result in more stochastic sample generations, producing noisier generator answers.
>
> We investigated the reasoning stochasticity hypothesis and quantified the effect using empirical answer disagreement entropy computed from 10 sampled generations per question. For each question, we group 10 sampled answers into an empirical distribution over unique answers $\{a_1,...,a_k\}$ with probabilities $p(a_k)=count(a_k)/k$, where $k=10$. We compute the disagreement entropy as $H=-\sum_{a\in\mathcal{A}} p(a)\log p(a)$, where higher entropy means more disagreement.
>
> We assessed stochasticity on the five multiple choice datasets, excluding TriviaQA whose free response format makes it difficult for answer grouping. As shown in the table below, among the remaining five datasets, we observe that reasoning-heavy datasets tend to show higher disagreement entropy, indicating higher answer level variability under sampling.
>
>
> | Dataset | Disagreement Entropy with Generator=GPT-3.5-turbo | Disagreement Entropy with Generator=GPT-4o |
> | ----------- | ---- | ---- |
> | CounterFact | 9.73 |9.25 |
> | Winogrande | 25.72 | 42.05 |
> | MMLU | 28.21 | 44.55 |
> | MedQA | 48.59 | 50.31 |
> | SuperGPQA | 87.75 | 71.56 |
>
> In the manuscript, we included this disagreement entropy analysis to demonstrate that reasoning benchmarks are more stochastic and less identifiable from question-only features, helping clarify why performance is comparatively weaker in these settings.

---

> ### Author Response · Authors · 2026-02-28
> **Response to Reviewer NPrr (3/3)**
>
> > "Fairer Reporting in Table 1: Table 1 currently reports only the \"best selective accuracy among all predictors\" for each task. Given that the authors admit scaling trends are not universal and larger models sometimes underperform smaller ones, reporting only the upper bound obscures the variance and reliability of the method."
>
> We agree that Table 1 could benefit from reporting both mean and standard deviation of the selected predictor family. Following your suggestion, we revised Table 1 to report selective accuracy using predictions from a Llama3-8B backbone and also reported the distribution mean and standard deviation across five Llama backbones, as shown in the table below. We included this additional table in Appendix B *Selective Generator Accuracy* in the manuscript.
>
> To ensure transparency, we also added a note to the manuscript to explain that the selective generator accuracy reported in Table 1 is computed from predictions obtained with a Llama3-8b backbone. We selected this backbone because it offers strong open-weight performance and is reasonably easy to run on compute infrastructure available to us.
>
> ---
> | Dataset     | GPT 3.5 turbo Acc. | GPT 3.5 turbo Sel. Acc. (10%) | GPT 3.5 turbo Sel. Acc. (30%) | GPT 3.5 turbo Sel. Acc. (50%) | GPT 4o Acc. | GPT 4o Sel. Acc. (10%) | GPT 4o Sel. Acc. (30%) | GPT 4o Sel. Acc. (50%) |
> | :---------- | -----------------: | ---------------------------: | ---------------------------: | ---------------------------: | ----------: | --------------------: | --------------------: | --------------------: |
> | TriviaQA    |              74.33 |             79.27 $\pm$ 1.40 |             85.19 $\pm$ 2.99 |             88.77 $\pm$ 3.68 |       81.47 |      85.55 $\pm$ 1.15 |      89.26 $\pm$ 2.14 |      91.41 $\pm$ 2.73 |
> | CounterFact |              94.42 |             96.28 $\pm$ 0.36 |             97.33 $\pm$ 0.70 |             97.95 $\pm$ 0.84 |       95.68 |      97.16 $\pm$ 0.31 |      97.90 $\pm$ 0.49 |      98.24 $\pm$ 0.69 |
> | Winogrande  |              72.70 |             73.95 $\pm$ 0.73 |             75.24 $\pm$ 1.73 |             77.06 $\pm$ 3.56 |       90.00 |      90.57 $\pm$ 0.36 |      90.99 $\pm$ 0.91 |      91.24 $\pm$ 1.44 |
> | MMLU        |              69.78 |             71.96 $\pm$ 0.56 |             74.40 $\pm$ 1.55 |             75.87 $\pm$ 2.63 |       85.93 |      87.24 $\pm$ 0.53 |      88.35 $\pm$ 0.95 |      89.00 $\pm$ 1.90 |
> | MedQA       |              59.95 |             61.21 $\pm$ 0.43 |             63.82 $\pm$ 1.62 |             66.29 $\pm$ 2.33 |       88.30 |      89.24 $\pm$ 0.45 |      89.94 $\pm$ 0.85 |      90.48 $\pm$ 0.76 |
> | SuperGPQA   |              32.37 |             33.20 $\pm$ 0.42 |             35.58 $\pm$ 0.63 |             38.71 $\pm$ 1.36 |       44.98 |      46.60 $\pm$ 0.28 |      50.46 $\pm$ 0.89 |      54.98 $\pm$ 1.94 |
>
> We believe the additional experiments and explanations provided in this response meaningfully strengthen our manuscript, and we hope you find these clarifications and the added results and analysis insightful.
> We would be grateful for your support and hope you will view our response and the revisions in our manuscript favorably.
>
> ---
> ### References
>
> [1] Wang X, Wei J, Schuurmans D, Le Q, Chi E, Narang S, Chowdhery A, Zhou D. Self-Consistency Improves Chain of Thought Reasoning in Language Models. In International Conference on Learning Representations (ICLR) 2023. arXiv preprint arXiv:2203.11171.
>
> [2] Kuhn L, Gal Y, Farquhar S. Semantic Uncertainty: Linguistic Invariances for Uncertainty Estimation in Natural Language Generation. arXiv preprint arXiv:2302.09664.
>
> [3] Kadavath S, Conerly T, Askell A, et al. Language Models (Mostly) Know What They Know. arXiv preprint arXiv:2207.05221.
>
> [4] Song, P., Han, P., and Goodman, N. Large Language Model Reasoning Failures. Transactions on Machine Learning Research (TMLR), January 2026.
>
> [5] Yang S, Gribovskaya E, Kassner N, Geva M, Riedel S. Do large language models latently perform multi-hop reasoning?. In Proceedings of the 62nd Annual Meeting of the Association for Computational Linguistics (Volume 1: Long Papers) 2024 .
>
> [6] Li X, Wang W, Li M, Guo J, Zhang Y, Feng F. Evaluating mathematical reasoning of large language models: A focus on error identification and correction. In Findings of the Association for Computational Linguistics: ACL 2024 .

---

### Review · Reviewer_uoEB · 2026-01-14

**Summary Of Contributions:**

- This paper systematically studies weak-to-strong confidence prediction. Specifically, it uses embeddings from a smaller, open-access predictor model and a simple probe to estimate the probability that a stronger, black-box LLM generator (e.g., GPT-3.5-turbo or GPT-4o) will answer a given question correctly. The authors construct supervision by sampling multiple generator responses per query, computing a per-question correctness as a soft label, and training a lightweight MLP probe on frozen question-only embeddings. They show that these predictors can rank generator reliability and improve selective prediction performance by abstaining on low-confidence queries. Through controlled experiments, they find that predictor performance does not universally scale with predictor model size and is relatively robust to label imbalance and embedding aggregation strategies.
- Strengths
  - The paper proposes a lightweight approach to estimate a black-box generator’s likelihood of being correct without requiring access to the generator’s logits or hidden states.
  - The authors provide the experimental results over multiple benchmark datasets and a large set of predictor backbones.
  - The paper includes controlled experiments on label imbalance and embedding aggregation methods. These ablations provide actionable guidance on when the method is likely to work well.
- Weaknesses
  - The results are obtained by splitting train and test within the same benchmark dataset, which can overstate how well the predictor would transfer to unseen datasets. Since the authors claim that the proposed approach is broadly applicable in practice, stronger evidence would require evaluating true transfer settings (e.g., training on one benchmark and testing on a different benchmark), rather than primarily in-dataset splits.
  - Because $y_{true}$ is constructed from multiple sampled generator outputs, its value can change with decoding strategies, sampling budget $k$, etc. The paper would be more convincing with a systematic sensitivity analysis showing how robust the conclusions are to these sampling variations.
  - The proposed method should be compared with simple baselines. For example, computing self-consistency or disagreement based on multiple generated outputs can be a straightforward simple baseline that also works in a black-box setting.

**Audience:**

No

**Audience Explanation:**

The approach of estimating a black-box generator’s confidence using a weak predictor is interesting. However, the current evidence is limited because it is unclear whether the approach generalizes across different datasets, domains, or decoding settings. As a results, the direction is interesting, but the findings in the current form are not especially compelling.

**Claims And Evidence:**

No

**Claims Explanation:**

I do not find the paper’s claims fully supported by the evidence as presented. While the experiments show that a simple probe on weak-model question embeddings can predict a black-box generator’s correctness, these results are obtained under a fairly constrained setup. Therefore, it remains unclear whether the method captures a general, transferable notion of confidence or mostly learns dataset-specific question difficulty.

**Requested Changes:**

The paper would benefit from revisions that strengthen the main claims by addressing the weaknesses discussed above.

---

> ### Author Response · Authors · 2026-02-28
> **Response to Reviewer uoEB (1/3)**
>
> Dear Reviewer uoEB,
>
> We sincerely thank you for taking the time to write such a detailed and comprehensive review.
> We appreciate that you highlighted three strengths you found valuable: the lightweight black-box approach, the breadth of datasets and predictor backbones, and the controlled ablations on label imbalance and embedding aggregation.
>
> We address your concerns below.
>
> ---
>
> > "stronger evidence would require evaluating cross benchmark transfer, rather than primarily in-dataset splits."
>
> We appreciate your question of cross-domain generalization. Since we’re considering a simple supervised learning problem with data from narrow domains, we would not expect to see any meaningful cross-dataset generalization unless datasets are meaningfully related, but to investigate whether the learned representations capture a transferrable uncertainty direction, we conducted out-of-distribution (OOD) transfer evaluation as below. Specifically, we used probes trained on one dataset and evaluated their AUROC performance on the remaining datasets without any further fine-tuning. We ​​used Llama3-70B-Instruct as the predictor backbone and GPT-4o as the generator for illustration.
>
> | Source \ Target | TriviaQA | CounterFact | Winogrande | MMLU | MedQA | SuperGPQA |
> | ----------- | ---- | ----- | ---- | ---- | ---- | ---- |
> | TriviaQA    | 81.3 | 47.0  | 52.0 | 57.0 | 56.1 | 50.8 |
> | CounterFact | 50.6 | 85.2  | 47.8 | 58.0 | 51.7 | 50.2 |
> | Winogrande  | 48.9 | 45.8  | 65.5 | 50.4 | 50.1 | 51.3 |
> | MMLU        | 46.0 | 51.8  | 51.7 | 65.9 | 55.2 | 62.8 |
> | MedQA       | 53.5 | 53.0  | 49.9 | 55.8 | 66.2 | 61.0 |
> | SuperGPQA   | 59.7 | 44.2  | 51.4 | 53.7 | 52.9 | 64.8 |
>
> Probes trained on more difficult reasoning tasks exhibit slightly better OOD generalization performance than those trained on knowledge retrieval tasks, but overall there is no consistently transferable uncertainty direction across these domains.
> In fact, most AUROC scores are around 50%, which aligns with random guessing.
>
> Limited zero shot transfer is expected in our setting because the probe only sees question embeddings (Section 3.2) and the datasets differ substantially in domain, style, and difficulty (Section 3.1).
> However, zero shot transfer is not required for practical usefulness: the probe is lightweight and inexpensive to train per target distribution once generator answers can be sampled, and we envision this simple approach can be particularly useful in narrowly defined settings (e.g., in medicine).
>
> We added a dedicated subsection in the appendix discussing these OOD findings, as we believe this dataset-dependency is a valuable insight for the community working on LLM probing.
> We also updated the Introduction to mention the practical meaning of such probing.
>
> ---
> > "Because y_true is constructed from multiple sampled generator outputs, its value can change with decoding strategies, sampling budget k, etc.
> The paper would be more convincing with a systematic sensitivity analysis showing how robust the conclusions are to these sampling variations."
>
> Thank you for the suggestion.
> In response to your comment, we performed a targeted sensitivity analysis that varies sampling temperature $(0.2, 0.8, 1.0, 1.2)$, the nucleus sampling threshold $top_p$ $(0.9, 1.0)$ [1], and the sampling budget $k$ $(5, 10, 20)$.
>
> We evaluated across six datasets and all fifteen representative predictors and found that our main qualitative conclusions are generally stable and do not heavily depend on the sampling hyperparameter choices.
> Here in the rebuttal, we present Llama3-8b and GPT-oss-20b as a representative subset of small and mid scale predictors from different model families; the same qualitative conclusions hold across all fifteen predictors, and the full results are included in Appendix H in the manuscript for completeness.

---

> ### Author Response · Authors · 2026-02-28
> **Response to Reviewer uoEB (2/3)**
>
> We find that small predictor backbones such as Llama3-8b still provide effective weak-to-strong confidence signals, and knowledge retrieval tasks remain easier to predict than reasoning benchmarks.
> Halving the sample budget from $10$ to $5$ usually causes a small drop in AUROC, while doubling it from $10$ to $20$ often yields modest gains, but neither change alters the overall gains from selective prediction or the main conclusions about representational compatibility.
> Together, these observations provide evidence that successful weak-to-strong confidence prediction is dependent primarily on the underlying representations rather than by a particular choice of decoding hyperparameters or sampling budget.
>
>
> **TriviaQA**
>
> | LM          | temp=1.0, $top_p$=1.0 | temp=0.2, $top_p$=1.0 | temp=1.2, $top_p$=1.0 | temp=1.0, $top_p$=0.9 | k=5   | k=20  |
> | ----------- | ------------------ | ------------------ | ------------------ | ------------------ | ----- | ----- |
> | AUGAC - Llama3-8b   | 92.10     | 90.63  | 90.62   | 90.52              | 91.83 | 91.22 |
> | AUGAC - GPT-oss-20b | 91.56   | 90.70    | 76.09              | 76.73              | 90.99 | 78.84 |
> | AUROC - Llama3-8b   |   80.06   |   83.56   | 77.33        |       77.78       | 79.50  | 82.36 |
> | AUROC - GPT-oss-20b |     79.69   |  83.46   | 83.21        |      83.23        | 79.25  | 82.02  |
>
> **CounterFact**
>
> | LM          | temp=1.0, $top_p$=1.0 | temp=0.2, $top_p$=1.0 | temp=1.2, $top_p$=1.0 | temp=1.0, $top_p$=0.9 | k=5   | k=20  |
> | ----------- | ------------------ | ------------------ | ------------------ | ------------------ | ----- | ----- |
> | AUGAC - Llama3-8b   | 98.27              | 97.31              | 97.48              | 97.31              | 97.60 | 97.72 |
> | AUGAC - GPT-oss-20b | 98.46              | 97.26              | 94.89              | 94.58              | 97.54 | 95.13 |
> | AUROC - Llama3-8b   |   81.95           |      76.46        |     78.21       |       75.91      | 78.38  | 80.07 |
> | AUROC - GPT-oss-20b |     78.97         |     74.12         |      77.59       |      75.25      | 78.22  | 79.07  |
>
> **Winogrande**
>
> | LM          | temp=1.0, $top_p$=1.0 | temp=0.2, $top_p$=1.0 | temp=1.2, $top_p$=1.0 | temp=1.0, $top_p$=0.9 | k=5   | k=20  |
> | ----------- | ------------------ | ------------------ | ------------------ | ------------------ | ----- | ----- |
> | AUGAC - Llama3-8b   | 91.57              | 90.52              | 91.33              | 90.79              | 90.48 | 90.44 |
> | AUGAC - GPT-oss-20b | 92.70              | 91.64              | 88.89              | 89.17              | 91.91 | 89.21 |
> | AUROC - Llama3-8b   | 59.56              | 57.64             | 58.85             | 58.57             | 55.89 | 57.50 |
> | AUROC - GPT-oss-20b | 60.93              | 59.54              | 60.18              | 58.46              | 60.09 | 61.41 |
>
> **MMLU**
>
> | LM          | temp=1.0, $top_p$=1.0 | temp=0.2, $top_p$=1.0 | temp=1.2, $top_p$=1.0 | temp=1.0, $top_p$=0.9 | k=5   | k=20  |
> | ----------- | ------------------ | ------------------ | ------------------ | ------------------ | ----- | ----- |
> | AUGAC - Llama3-8b   | 89.93              | 89.25              | 88.90              | 89.18              | 88.69 | 88.67 |
> | AUGAC - GPT-oss-20b | 90.58              | 89.09              | 85.19              | 85.02              | 89.42 | 85.48 |
> | AUROC - Llama3-8b   | 62.84             | 64.23             | 63.60              | 63.93              | 62.29 | 63.94 |
> | AUROC - GPT-oss-20b | 64.77  | 63.22              | 63.88              | 64.06              | 63.93 | 63.71 |
>
> **MedQA**
>
> | LM          | temp=1.0, $top_p$=1.0 | temp=0.2, $top_p$=1.0 | temp=1.2, $top_p$=1.0 | temp=1.0, $top_p$=0.9 | k=5   | k=20  |
> | ----------- | ------------------ | ------------------ | ------------------ | ------------------ | ----- | ----- |
> | AUGAC - Llama3-8b   | 90.51  | 89.25  | 90.24 | 89.91 | 89.70 | 89.85 |
> | AUGAC - GPT-oss-20b | 91.85  | 90.84 | 87.45 | 87.19  | 90.73 | 86.93 |
> | AUROC - Llama3-8b   | 62.06 | 60.88 | 62.43 | 60.80 | 61.25 | 61.74 |
> | AUORC - GPT-oss-20b | 63.55  | 62.64   | 63.55  | 61.56  | 63.71 | 63.21 |
>
> **SuperGPQA**
>
> | LM          | temp=1.0, $top_p$=1.0 | temp=0.2, $top_p$=1.0 | temp=1.2, $top_p$=1.0 | temp=1.0, $top_p$=0.9 | k=5   | k=20  |
> | ----------- | -------------- | ------- | -------- | ------- | ------- | ------ |
> | AUGAC - Llama3-8b     | 57.14   | 58.47  | 44.86  | 45.90  | 45.62 | 46.09 |
> | AUGAC - GPT-oss-20b | 57.71   | 58.50  | 47.32  | 47.55  | 57.80 | 47.63 |
> | AUROC - Llama3-8b     | 59.90   | 65.84  | 67.11  | 67.00  | 67.44 | 66.90 |
> | AUROC - GPT-oss-20b | 66.99   | 65.06  | 67.83  | 66.72  | 66.35 | 66.83 |

---

> ### Author Response · Authors · 2026-02-28
> **Response to Reviewer uoEB (3/3)**
>
> > "Add simple black-box baselines such as self-consistency or disagreement based on multiple generated outputs"
>
> We appreciate your suggestion to add additional black-box baselines based on multiple sampled outputs. We agree that they would be natural reference points.
>
> For each question, we first normalized each sampled response and then treated identical normalized strings as the same answer.
> Self-consistency (SC) [2] is computed as the fraction of samples that match the majority answer.
> Disagreement (DA) [3] is computed as the negative entropy of the empirical answer distribution across samples, so that higher values indicate higher confidence.
> Confidence prompting (CP) follows Kadavath et al. [4] and performs post hoc self evaluation: we prompt the generator to judge whether a proposed answer is True or False, and use this judgement as a confidence score for predicting answer correctness.
> Note that these baselines all require $k$ generator calls at inference time, while the simple probe-based method considered in our manuscript uses only small-model embeddings and a probe trained on $k$ generator calls obtained upfront, but does not require any samples at test time.
>
> We find that the trained probes outperform both self consistency (SC) and disagreement (DA) on most datasets; on MMLU and SuperGPQA the performance is comparable (but SC and DA require $k$ forward passes through the large generator model), with SC and DA marginally above the probe baseline. TriviaQA shows exceptionally high SC and DA performance because we directly identify correct answers and group the remaining incorrect answers using naive string matching, which often fragments incorrect answers into many small groups and artificially inflates the estimated SC and DA metrics. Confidence prompting is generally less worse than our probes.
> This matches our main finding that question-only embeddings provide strong signals on knowledge recall, while on reasoning-heavy tasks and post hoc self evaluation are often noisier and less calibrated.
>
> **Generator = GPT-3.5-turbo**
> | Metric \ Dataset  | TriviaQA | CounterFact | Winogrande | MMLU  | MedQA | SuperGPQA |
> | ---------------------- | -------- | ----------- | ---------- | ----- | ----- | --------- |
> | Self-Consistency AUROC | 97.03    | 72.75       | 58.45      | 70.32 | 60.06 | 64.22     |
> | Self-Consistency AUPRC | 97.69    | 97.25       | 73.97      | 82.18 | 66.74 | 49.37     |
> | Disagreement AUROC     | 96.92    | 72.79       | 58.46      | 70.45 | 60.88 | 64.98     |
> | Disagreement AUPRC     | 97.74    | 97.26       | 73.98      | 82.27 | 67.11 | 50.07     |
> | Prompt AUROC  | 54.06    | 73.81       | 51.63      | 60.10 | 59.43 | 57.12     |
> | Prompt AUPRC  | 68.11    | 96.56       | 68.48      | 76.93 | 65.00 | 43.43     |
> | Llama3-8b AUROC (Probe)     | 82.47    | 81.07       | 61.68     | 63.77 | 59.96 | 52.33     |
> | Llama3-8b AUPRC (Probe)     | 91.89    | 98.38       | 87.80      | 78.82 | 69.04 | 22.56     |
>
> **Generator = GPT-4o**
> | Metric   | TriviaQA | CounterFact | Winogrande | MMLU  | MedQA | SuperGPQA |
> | ---------------------- | -------- | ----------- | ---------- | ----- | ----- | --------- |
> | Self-Consistency AUROC | 92.42    | 83.50  | 60.29      | 71.95 | 62.54 | 67.52     |
> | Self-Consistency AUPRC | 95.23    | 98.64   | 90.34      | 93.15 | 91.71 | 65.53     |
> | Disagreement AUROC     | 92.37    | 83.46  | 61.48      | 71.76 | 64.51 | 68.62     |
> | Disagreement AUPRC     | 95.26    | 98.64  | 90.55      | 93.15 | 92.06 | 66.32     |
> | Prompt AUROC   | 56.12    | 68.03   | 64.41   | 57.95 | 55.89 | 55.16     |
> | Prompt AUPRC    | 75.50    | 96.93  | 90.17   | 89.15 | 89.50 | 59.06     |
> | Llama3-8b AUROC (Probe)   | 80.06    | 81.95   | 59.56      | 62.84 | 62.06 | 59.90     |
> | Llama3-8b AUPRC (Probe)  | 93.47   | 98.43   | 94.45      | 90.07 | 91.71 | 50.45     |
>
> We believe the additional experiments and explanations provided in this response meaningfully strengthen our manuscript, and we hope you find these clarifications and the added results and analysis insightful.
> We would be grateful for your support and hope you will view our response and the revisions in our manuscript favorably.
>
> ---
> ### References
>
> [1] Holtzman A, Buys J, Du L, Forbes M, Choi Y. The Curious Case of Neural Text Degeneration. In International Conference on Learning Representations (ICLR) 2020. arXiv preprint arXiv:1904.09751.
>
> [2] Wang X, Wei J, Schuurmans D, Le Q, Chi E, Narang S, Chowdhery A, Zhou D. Self-Consistency Improves Chain of Thought Reasoning in Language Models. In International Conference on Learning Representations (ICLR) 2023. arXiv preprint arXiv:2203.11171.
>
> [3] Kuhn L, Gal Y, Farquhar S. Semantic Uncertainty: Linguistic Invariances for Uncertainty Estimation in Natural Language Generation. arXiv preprint arXiv:2302.09664.
>
> [4] Kadavath S, Conerly T, Askell A, et al. Language Models (Mostly) Know What They Know. arXiv preprint arXiv:2207.05221.

---

### Review · Reviewer_aXK4 · 2026-02-06

**Summary Of Contributions:**

The paper studies weak-to-strong confidence prediction in large language models, showing that embeddings from smaller, open-access models combined with simple probes can reliably predict whether stronger, black-box models will answer questions correctly.

1. Along six benchmarks and multiple predictor backbones, the approach improves selective prediction accuracy and demonstrates robustness to label imbalance and embedding choices.

2. A key finding is that representation alignment matters more than model size.

**Audience:**

Yes

**Audience Explanation:**

The work is relevant to researchers interested in uncertainty quantification, model evaluation, and AI safety. Its focus on external confidence estimation for closed or frontier models addresses a practical and timely problem.

**Claims And Evidence:**

Yes

**Claims Explanation:**

The claims are supported by extensive experiments across multiple datasets, generator models, and predictor backbones. Results are evaluated using appropriate metrics, including AUROC and selective accuracy, and are reinforced by thorough ablation studies.

**Requested Changes:**

1. Provide additional insight into why performance is weaker on reasoning-heavy tasks.
2. Expand discussion on cross-dataset or transfer generalization.

---

> ### Author Response · Authors · 2026-02-28
> **Response to Reviewer aXK4 (1/2)**
>
> Dear Reviewer aXK4,
>
> We sincerely thank you for the thoughtful and constructive review. We are especially grateful that you highlighted several strengths of the paper, noting the selective prediction gains across benchmarks, the robustness analysis on label imbalance and embedding aggregation, and the central insight that representational alignment matters more than model size with clear relevance to uncertainty quantification and safety.
>
> We address your questions and comments below.
>
> ---
>
> > "Provide additional insight into why performance is weaker on reasoning-heavy tasks."
>
> We agree that the current explanation for the performance gap on reasoning heavy tasks can be strengthened beyond the high-level intuition that question-only embeddings may not reflect intermediate reasoning steps. We expanded our discussion with the following plausible reasons: question-only embeddings capture topical content and coarse difficulty, but they do not condition on latent intermediate steps that determine correctness in multi step reasoning; multiple failure modes exist in latent reasoning path [1,2,3], increasing difficulty in separability for probes; latent reasoning chains could result in more stochastic sample generations, producing noisier generator answers.
>
> We investigated the stochasticity hypothesis and quantified the effect using empirical answer disagreement entropy computed from 10 sampled generations per question. For each question, we group 10 sampled answers into an empirical distribution over unique answers $\{a_1,...,a_k\}$ with probabilities $p(a_k)=count(a_k)/k$, where $k=10$. We compute the disagreement entropy as $H=-\sum_{a\in\mathcal{A}} p(a)\log p(a)$, where higher entropy means more disagreement.
>
> We evaluate on the five multiple choice datasets, excluding TriviaQA whose free response format makes it difficult for answer grouping. As shown in the table below, among the remaining five datasets, we observe that reasoning-heavy datasets tend to show higher disagreement entropy, indicating higher answer level variability under sampling.
>
> | Dataset | Disagreement Entropy with Generator=GPT-3.5-turbo | Disagreement Entropy with Generator=GPT-4o |
> | ----------- | ---- | ---- |
> | CounterFact | 9.73 |9.25 |
> | Winogrande | 25.72 | 42.05 |
> | MMLU | 28.21 | 44.55 |
> | MedQA | 48.59 | 50.31 |
> | SuperGPQA | 87.75 | 71.56 |
>
> In the manuscript, we included this disagreement entropy analysis to demonstrate that reasoning benchmarks are more stochastic and less identifiable from question-only features, helping clarify why performance is comparatively weaker in these settings.

---

> ### Author Response · Authors · 2026-02-28
> **Response to Reviewer aXK4 (2/2)**
>
> > "Expand discussion on cross-dataset or transfer generalization."
>
> We appreciate your question of cross-domain generalization. Since we’re considering a simple supervised learning problem with data from narrow domains, we would not expect to see any meaningful cross-dataset generalization unless datasets are meaningfully related, but to investigate whether the learned representations capture a transferrable uncertainty direction, we conducted out-of-distribution (OOD) transfer evaluation as below. Specifically, we used probes trained on one dataset and evaluated their AUROC performance on the remaining datasets without any further fine-tuning. We ​​used Llama3-70B-Instruct as the predictor backbone and GPT-4o as the generator for illustration.
>
> | Source \ Target | TriviaQA | CounterFact | Winogrande | MMLU | MedQA | SuperGPQA |
> | ----------- | ---- | ----- | ---- | ---- | ---- | ---- |
> | TriviaQA    | 81.3 | 47.0  | 52.0 | 57.0 | 56.1 | 50.8 |
> | CounterFact | 50.6 | 85.2  | 47.8 | 58.0 | 51.7 | 50.2 |
> | Winogrande  | 48.9 | 45.8  | 65.5 | 50.4 | 50.1 | 51.3 |
> | MMLU        | 46.0 | 51.8  | 51.7 | 65.9 | 55.2 | 62.8 |
> | MedQA       | 53.5 | 53.0  | 49.9 | 55.8 | 66.2 | 61.0 |
> | SuperGPQA   | 59.7 | 44.2  | 51.4 | 53.7 | 52.9 | 64.8 |
>
> Probes trained on more difficult reasoning tasks exhibit slightly better OOD generalization performance than those trained on knowledge retrieval tasks, but overall there is no consistently transferable uncertainty direction across these domains.
> In fact, most AUROC scores are around 50%, which aligns with random guessing.
> Limited zero shot transfer is expected in our setting because the probe only sees question embeddings (Section 3.2) and the datasets differ substantially in domain, style, and difficulty (Section 3.1).
> However, zero shot transfer is not required for practical usefulness: the probe is lightweight and inexpensive to train per target distribution once generator answers can be sampled, and we envision this simple approach can be particularly useful in narrowly defined settings (e.g., in medicine).
>
> We added a dedicated subsection in the appendix discussing these OOD findings, as we believe this dataset-dependency is a valuable insight for the community working on LLM probing.
> We also updated the Introduction to mention the practical meaning of such probing.
>
> We believe the additional experiments and explanations provided in this response meaningfully strengthen our manuscript, and we hope you find these clarifications and the added results and analysis insightful.
> We would be grateful for your support and hope you will view our response and the revisions in our manuscript favorably.
>
> ---
>
> ### References
>
> [1] Song, P., Han, P., and Goodman, N. Large Language Model Reasoning Failures. Transactions on Machine Learning Research (TMLR), January 2026.
>
> [2] Yang S, Gribovskaya E, Kassner N, Geva M, Riedel S. Do large language models latently perform multi-hop reasoning?. In Proceedings of the 62nd Annual Meeting of the Association for Computational Linguistics (Volume 1: Long Papers) 2024 .
>
> [3] Li X, Wang W, Li M, Guo J, Zhang Y, Feng F. Evaluating mathematical reasoning of large language models: A focus on error identification and correction. In Findings of the Association for Computational Linguistics: ACL 2024.

---

### Author Response · Authors · 2026-05-10
**Camera Ready Revision**

We thank the reviewers and Action Editor for their helpful feedback and thoughtful discussion, which helped us improve the presentation of the paper. We have incorporated all requested revisions and have also submitted the camera ready version of the manuscript.

---

### Decision · Action_Editor_sgFL · 2026-04-07

**Recommendation:** Accept with minor revision

**Additional Comments:**

Please add discussion to this point which I can easily verify.

Here are two quotes from the borderline reviewers that the authors cannot see:
> Failure of OOD Transfer: Evaluations show out-of-distribution AUROC scores near 50%, suggesting the probes are highly specialized to their training distributions and do not capture a transferable uncertainty direction.
> Reasoning Bottleneck: The method scales poorly on complex benchmarks like SuperGPQA, where selective accuracy for GPT-4o only reaches 57.91% even at a 50% rejection rate. This stems from a fundamental limitation: question-only embeddings miss the latent intermediate steps critical for reasoning

and

> If question embeddings already contain enough dataset-specific information about difficulty, then it is not particularly surprising that a small probe can predict within-dataset correctness. The added cross-dataset results further suggest that the learned signal is not transferable, but largely tied to benchmark-specific regularities. In addition, the practical advantage over simple black-box baselines remains unclear, and the efficiency comparison is somewhat misleading because the proposed method also relies on repeated generator calls during training.

**Audience:**

Yes

**Audience Explanation:**

Yes, I think the topic of this paper is definitely of interest to TMLR readers.

**Claims And Evidence:**

Yes

**Claims Explanation:**

Two of three reviews agreed on this point.  They all appreciated the authors extensive and careful reply to their responses.  While the method provides more-efficient-than-sampling uncertainty quantification, there was concern that it did not generalize out of distribution -- indicating it may not be as robust a measure as desired, and not transferable.
The authors should carefully discuss this concern in the final version.

Overall, the majority of the reviewers found the work contained interesting findings despite these OOD-generalization concerns.  The concepts and experiments underlying the paper were of interest even if the results did not pan out to be quite as useful as hoped.

---

> ### Author Response · Authors · 2026-05-10
> **Response to Action Editor sgFL**
>
> Dear Action Editor sgFL,
>
> Thank you for the decision and for the clear guidance. We appreciate the reviewers’ careful reading and are glad that the paper’s findings were seen as interesting and relevant to the TMLR audience.
>
> Regarding the first quote,
>  - We agree with the reviewer that the probe shows weak OOD transferability, with AUROC near chance in several settings. We have revised the manuscript to narrow our claims and position the method primarily as an in-distribution uncertainty estimator.
>  - Performance also degrades on difficult reasoning benchmarks such as SuperGPQA, which suggests that question-only embeddings do not fully capture the information needed for these tasks. We have revised the discussion to state this limitation more directly.
>
> Regarding the second quote,
>  - We agree with the reviewer that question embeddings can encode dataset specific regularities, including signals correlated with difficulty. We have revised the manuscript to make clear that our claim is not that question-only embeddings provide a broadly transferable uncertainty signal, but rather that the representation of weak models contain useful signals for uncertainty prediction and the probe can recover it, with limited transfer across benchmarks.
>  - We also agree that the cross dataset results indicate that much of the learned signal is benchmark specific rather than broadly transferrable. We view these results primarily as a limitation, and we have revised the manuscript to make that limitation explicit and to avoid overstating generalization.
>  - We have also reframed the comparison with black box baselines more carefully. In the revised Table 2 and accompanying discussion, we show that stronger probe backbones are competitive with the baselines on knowledge retrieval datasets, but they are not uniformly better, particularly on reasoning datasets.
>  - Finally, we have clarified the efficiency discussion by distinguishing training cost from inference cost. The intended advantage of our method compared to baselines is in inference: once the probe is trained, new questions can be scored without repeated sampling, which may be beneficial under repeated use on a stable distribution. We have revised the discussion to avoid implying an end-to-end efficiency advantage over black-box methods when training data collection costs are included.
>
> Thank you again for the helpful feedback.